# Post-Training LLMs as Better Decision-Making Agents: A Regret-Minimization Approach

Chanwoo Park [1]   Ziyang Chen [2]   Asuman Ozdaglar [1]   Kaiqing Zhang [2]

## Abstract

Large language models (LLMs) are increasingly deployed as agents for decision-making (DM) in interactive and dynamic environments. However, since they are not originally designed for DM, recent studies show that LLMs struggle in basic online DM settings. We introduce ITERATIVE REGRET-MINIMIZATION FINE-TUNING (ITERATIVE RMFT), a post-training procedure that repeatedly distills low-regret decision trajectories into the base model. Unlike prior methods that rely on distilling known algorithms or enforcing manually designed reasoning formats, our approach leverages regret as a training signal to elicit improved decision-making behavior while incorporating model-generated reasoning in natural language. Empirically, ITERATIVE RMFT improves DM performance across models, including numerical Transformers, lightweight open-weight LLMs, and the closed-weight model GPT-4o mini, while exhibiting generalization across varying horizons, action spaces, reward processes, and natural-language-described DM scenarios. Overall, we position our approach as an initial exploration, calling for more principled and novel post-training paradigms for LLMs when it comes to addressing DM tasks.

## 1. Introduction

Many real-world decision-making (DM) tasks are grounded in *natural language*, where both the task descriptions and the input/output of the decision-maker are expressed in language, beyond the classical DM formulations with symbolic and numeric quantities (Lattimore &

Szepesvári, 2020; Hazan, 2016). This makes LLMs, with their prominent language interface, a natural candidate as a *DM agent* for language-grounded DM tasks, with many empirical successes (see §B for more examples). However, since LLMs are not trained as decision-makers, it remains unclear why they would excel in DM. In fact, recent studies have shown that LLMs, when used *off-the-shelf* (*i.e.*, without fine-tuning or strong inference-time intervention), may perform poorly even in simple and canonical online DM tasks, as exhibited through their inadequate exploration and exploitation behavior (Krishnamurthy et al., 2024; Park et al., 2025b; Xia et al., 2025; Zhang et al., 2025b). Hence, it is important to systematically enhance the DM ability of LLMs within their design, before blindly deploying them as DM agents at scale.

In this work, we provide a new framework for post-training LLMs for language-grounded DM, based on the common metric of *regret* in online DM (Shalev-Shwartz, 2012; Hazan, 2016). Regret measures how much worse an agent's sequence of decisions is compared to the best possible decision (sequence) in hindsight. Minimizing regret typically requires a careful balance between exploration and exploitation, as well as robustness to arbitrary, or sometimes even adversarial, environments. Moreover, regret also offers a way of modeling and analyzing strategic behaviors in *multi-agent interactive* environments, with experimental evidence (Erev & Roth, 1998; Nekipelov et al., 2015). Indeed, regret has served as a *unified* metric to be optimized across various (numerical) online DM environments. *Can we leverage regret-minimization as a principle to post-train LLMs as better decision-makers?*

Specifically, we develop a self-improving post-training approach for eliciting LLMs' online DM abilities, ITERATIVE REGRET-MINIMIZATION FINE-TUNING (ITERATIVE RMFT), by iteratively performing supervised fine-tuning (SFT) on self-generated decision-making and reasoning trajectories with *low regret*. ITERATIVE RMFT is flexible enough to apply across various DM environments, thanks to the universality of regret as a metric for online DM. Because ITERATIVE RMFT naturally operates in the language space: it samples scenarios expressed in natural language, selects trajectories based on enhanced reasoning, ranks them by regret, and directly

---

[1]Massachusetts Institute of Technology, Massachusetts, USA [2]University of Maryland, College Park, Maryland, USA. Correspondence to: Chanwoo Park <cpark97@mit.edu>.

*Proceedings of the 43$^{rd}$ International Conference on Machine Learning*, Seoul, South Korea. PMLR 306, 2026. Copyright 2026 by the author(s).

fine-tunes on those trajectories, the approach does not need to be translated to a *pre-specified* numeric DM environment, *e.g.,* the action space size, reward generation processes, and time horizon, as in the recent studies Nie et al. (2025); Schmied et al. (2026). More importantly, the iterative SFT process naturally leverages and further enhances the chain-of-thought (CoT) reasoning rationales generated by LLMs, for improved online DM performance. Our SFT-based paradigm is also compatible with all existing post-training interfaces, even including the training API for proprietary *closed-weight* models such as the GPT-series. We evaluate the effectiveness of **ITERATIVE RMFT** on the canonical DM environments of *full-information online learning (FOL)*, *multi-armed bandits*, and *non-stationary multi-armed bandits (NS-MABs)*, when they are grounded in language. Beyond lowering regret values, **ITERATIVE RMFT** also automatically elicits enhanced online DM behavior from LLMs (*e.g.,* improved *exploration-exploitation (E-E) tradeoff*), despite not being explicitly guided by or distilled from existing expert algorithms from online DM. Without relying on pre-defined expert algorithms, **ITERATIVE RMFT** may be viewed as a way to empower LLMs to *autonomously discover algorithms* for DM, as incentivized by the regret metric. In a simplified single-layer attention setting, this imitation-based process again converges to FTRL, suggesting that no-regret behaviors may naturally emerge by this iterative self-imitation process. We defer a detailed literature review to Appendix B.

## 2. Preliminaries

### 2.1. Decision-Making Environments

We focus on the following canonical online DM environments: *full-information online learning*, *multi-armed bandits*, and *non-stationary multi-armed bandits*. Since these are standard DM environments, we defer a detailed introduction of them to Appendix D.1, and summarize the specifications of these environments and notation used throughout in Table 6.

**Language-Grounded Decision-Making.** These classical online DM environments are specified in a *non-linguistic* form: the actions $a \in \mathcal{A}$ are given as symbolic or numeric values, and the reward functions $r$ take numeric values. More importantly, both the *input* and *output* of the decision-making agent are *numeric*. In contrast, thanks to their language ability, LLM agents may address online DM tasks beyond these classical, purely numeric ones, where the task description, input, and output can all be language-grounded. We refer to each *language-based description* of a task as a *scenario* (see Sections 5.1 and 6.1 and Appendix I, J for more examples). The LLM agent is then asked to make a decision based on the language-based description of the interaction history so far in a *dialogue form*,

receives feedback from the environment, and repeats. The decision output from the LLM agent is also in the natural language format. Additionally, in contrast to the numeric online DM algorithms that *directly* map numeric inputs to the output policies, LLM agents may also *reason*, in the language format, about the decision before outputting the policies. We refer to such an interaction protocol between the LLM agent and the environment as *language-grounded DM*, and our *dialogue-form* interaction protocols resemble how LLMs are used as agents in real-world applications, in which the reasoning rationales of the agents can also be naturally integrated.

### 2.2. Performance Metric: Regret

In online DM, a fundamental measure of performance is the *regret* (Hazan, 2016; Szepesvári, 2022; Cesa-Bianchi & Lugosi, 2006). Given an algorithm $\mathscr{A}$ for such environments, the regret metric quantifies how much worse the agent's decisions are, during the course of learning, compared to those of the best possible policy. We denote this metric by $\text{Regret}_{\mathscr{A}}$ (or $\text{Regret}_{(\pi_{\mathscr{A},t})_{t \in [T]}}$), and define regret for the MAB environment as below. We defer the regret definitions for other online DM environments to Appendix D.2 due to space constraints. We also follow the regression-based procedure in Park et al. (2025b) to validate the no-regret behaviors (see Appendix D.3 for more details).

**Multi-Armed Bandits.** In the MAB environment, the *expected regret* of algorithm $\mathscr{A}$, which generates an action sequence $(a_{\mathscr{A},t})_{t=1}^{T}$ of length $T$, is defined as $\text{Regret}_{\mathscr{A}}(r, T) := \mathbb{E}\left[T \cdot \max_{a \in \mathcal{A}} r(a) - \sum_{t=1}^{T} r(a_{\mathscr{A},t})\right]$. The regret notion quantifies the difference between the cumulative expected reward that would have been obtained by selecting the best arm, and the expected reward accumulated by $\mathscr{A}$ through its chosen actions.

**Metrics for Exploration Efficiency.** To be qualified as a good decision-maker, the agent needs to balance *exploitation* with *exploration* in learning. To evaluate the *exploration efficiency* of LLM agents in MABs, other than regret, we also adopt the metrics proposed by Krishnamurthy et al. (2024): `SuffFailFreq`$(t)$ and `MinFrac`$(t)$. Specifically, `SuffFailFreq`$(t)$ measures the proportion of runs in which the best action with the highest expected reward is never selected from round $t$ to $T$; `MinFrac`$(t)$ captures how uniformly the available actions are explored. We provide more details of these metrics in Appendix D.6.

Other than these metrics, we will also compare the performance of our trained models with that of *known no-regret* learning algorithms in online DM as baselines.

# 3. Meta Algorithm: ITERATIVE REGRET-MINIMIZATION FINE-TUNING

Our goal is to develop a post-training paradigm for the decision-making ability of LLMs that is compatible with the interaction protocol with language description of the tasks, as well as language-grounded input/output. To this end, we propose to fine-tune LLMs based on *self-generated* language data. To promote the decision-making capability, we then propose to leverage the regret notions introduced in Section 2.2 as the criterion to select the self-generated trajectory data for training, which will then be used to fine-tune the model via supervised fine-tuning, reinforcing the low-regret behavior. Thanks to the universal applicability of the regret metric in online DM, this process can be viewed as a meta-algorithm generally applicable to environments considered in Table 6. We term such a meta-algorithm ITERATIVE REGRET-MINIMIZATION FINE-TUNING (ITERATIVE RMFT), as tabulated in Algorithm 1.

---

**Algorithm 1** Meta Algorithm: ITERATIVE REGRET-MINIMIZATION FINE-TUNING

---

1: **Input:** A DM environment (*e.g.*, FOL, MAB, NS-MAB); an initial model ($\mathscr{A}$) for DM
2: **for** iteration = $0, 1, 2, \ldots$ **do**
3:     $\mathcal{D} = \emptyset$
4:     **for** scenario index $i = 1, 2, \ldots, M$ **do**
5:         Sample $L$ trajectories $C_1, \ldots, C_L$ from the model under scenario$_i$
6:         Compute the regret (Section 2.2) of each trajectory and select the $k$ trajectories with the lowest regret: $C_{(1),i}, \ldots, C_{(k),i}$
7:         Update the training dataset: $\mathcal{D} = \mathcal{D} \cup \{\{\text{scenario}_i, C_{(1),i}, \ldots, C_{(k),i}\}\}$
8:     **end for**
9:     Fine-tune the model on the dataset $\mathcal{D}$ via supervised fine-tuning
10: **end for**

---

Specifically, in each iteration of ITERATIVE RMFT, we iterate over $M$ different *scenarios*. Here, a *scenario* refers to a specifically generated instance of a DM environment grounded in language. Then, for each scenario, $L$ trajectories are sampled from the current model $\mathscr{A}$. These $L$ trajectories are then evaluated based on the regret metric. Note that the regret can be computed *during training*, since the environments (*e.g.,* rewards and transitions) are generated by and thus known to the trainer, and the max operator in the regret definition can thus also be evaluated. Importantly, we highlight that, such privileged information is not required *during inference time*, and does not contradict our overall goal: the post-training procedure is not to teach LLM agents to solve DM tasks tied to a *particular* reward (seen during training), but instead to elicit their *DM capa-*

*bility* when facing novel DM tasks at inference time. See Remark 1 for a more detailed discussion. The top-$k$ trajectories with the *lowest regret* are then selected and added to the dataset $\mathcal{D}$. At the end of each iteration, the model is fine-tuned using the trajectories in $\mathcal{D}$ via supervised fine-tuning. This self-improving loop allows the model to iteratively refine its decision-making capability by learning from its own best-performing behaviors so far, in terms of the regret metric.

In the next sections, we will instantiate the meta-algorithm, Algorithm 1, in several online DM tasks with different environments and input-output modalities, as summarized in Table 1. More details on the definitions of these tasks are provided in each section separately.

# 4. Training Transformers with ITERATIVE RMFT

In this section, we instantiate the ITERATIVE REGRET-MINIMIZATION FINE-TUNING framework for training Transformer models to solve numerical online DM tasks. We will focus on the most basic online DM environments–FOL and MAB–throughout this section. Our goal is to assess the feasibility of our new post-training paradigm, by focusing on Transformers—the architecture behind most LLMs. The analyses in this section aim to provide a quantitative and more controlled understanding of the potential of ITERATIVE RMFT.

## 4.1. Experimental Setup

**Setup.** For the FOL environment, at each round $t$, the input to the Transformer is the history of numerical rewards $(R_1, R_2, \ldots, R_{t-1})$, and the output is a policy $\pi_t \in \Pi$ for the next round, where $\Pi = \Delta(\mathcal{A})$ or $\Pi = B(\mathbf{0}_d, R_\Pi, \|\cdot\|_2)$. For the MAB environment, at each round $t$, the input is the history of the observed rewards $(\widehat{R}_1, \ldots, \widehat{R}_{t-1})$, where $\widehat{R}_\tau \in \mathbb{R}^{|\mathcal{A}|}$ with $\widehat{R}_\tau(a) := \mathbf{1}(a = a_\tau)R_\tau(a)$, and $a_\tau$ is the action selected at round $\tau$. The Transformer then outputs a policy $\pi_t \in \Pi = \Delta(\mathcal{A})$, from which the next action $a_t \sim \pi_t$ is sampled. We provide more details on the reward generation processes used for training and evaluation in this section in Appendix G.

**Model.** We use a single-layer linear attention architecture with an output operator mapping to the policy space $\Pi$. This represents one of the simplest forms of Transformer architectures, and has been studied in the literature for the theoretical understanding of Transformers (Ahn et al., 2023; Mahankali et al., 2023; Park et al., 2025b). The model output at round $t$ is as follows:

| | (Section 4) **Transformers with Numerical Input/Output** | (Section 5) **Open-Weight LLMs** | (Section 6) **Closed-Weight LLMs** |
|---|---|---|---|
| **DM Environments** | FOL & MAB | FOL & MAB | FOL & MAB & NS-MAB |
| **Category of DM Tasks** | Numerical DM | Language-Grounded Numerical DM | Language-Grounded DM with Real-world Contexts |
| **Decision Output** | $\Delta(\mathcal{A})$ or $B(\mathbf{0}_d, R_\Pi, \|\cdot\|_2)$ | Extract $\begin{array}{l}\pi_t \in \Delta(\mathcal{A}) \text{ for FOL} \\ a_t \in \mathcal{A} \text{ for MAB}\end{array}$ from the output | Extract $\pi_t \in \Delta(\mathcal{A})$ from the output |
| **Sampling** | Gaussian Noise Perturbation | Stochastic Decoding | Stochastic Decoding |

*Table 1.* Summary of how our meta-algorithm is instantiated for different tasks studied in this paper.

$$g((R_1,\ldots,R_{t-1},\mathbf{1}_d); V, K, Q, v_c, k_c, q_c) = \qquad (1)$$
$$\texttt{Operator}\left(\sum_{\tau=1}^{t-1}(VR_\tau + v_c)((KR_\tau + k_c)^{\intercal}(Q\mathbf{1}_d + q_c))\right),$$

which is parameterized by $V, K, Q \in \mathbb{R}^{d \times d}$ and $v_c, k_c, q_c \in \mathbb{R}^d$, corresponding to the *value*, *key*, and *query* matrices and their respective bias vectors, respectively. We denote the full set of parameters as $\theta = (V, K, Q, v_c, k_c, q_c)$. The attention scores are computed over the reward history, and the resulting vector is then passed through an $\texttt{Operator}$. The choice of $\texttt{Operator}$ depends on the policy space $\Pi$: for the Experts Problem where $\Pi$ is the probability simplex (*i.e.*, $\Pi = \Delta(\mathcal{A})$), $\texttt{Operator}$ corresponds to the $\texttt{Softmax}$ operator; for the case where $\Pi$ is an $\ell_2$-ball (*i.e.*, $\Pi = B(\mathbf{0}_d, R_\Pi, \|\cdot\|_2)$), $\texttt{Operator}$ corresponds to the projection operation onto the $\ell_2$-ball.

## 4.2. Instantiating ITERATIVE RMFT for Numerical Decision-Making with Transformers

We instantiate the meta algorithm of Algorithm 1 for numerical DM tasks (cf. Section 4.1) as Algorithm 2. In Line 5 of Algorithm 1, we need to sample multiple trajectories per scenario. However, in this numerical setting, a single Transformer model $\mathscr{A}$ *deterministically* maps numerical reward histories to policies, without randomness nor sampling. This deterministic nature of the output differs from the sampling process of actual LLMs, to which feeding the same reward history yields different trajectories due to the stochastic nature of token sampling. In particular, the decision-making policy from LLMs is implicitly defined through the autoregressive generation process, where each sampled token affects subsequent ones, and the final output text encodes the policy.

To fill this gap and generate diverse trajectories from the deterministic Transformer model $\mathscr{A}$, we introduce stochastic perturbations to its output at each round in Algorithm 2. For each scenario$_i$, we sample $L$ trajectories $C_{1,i}, \ldots, C_{L,i}$, where each trajectory $C_{\ell,i} = (\pi_{\ell,i,t})_{t \in [T]}$ is a sequence of *perturbed* policies. Specifically, at each round $t$, the model

outputs the following policy:

$$\pi_{\ell,i,t} = \texttt{Operator}\left(\pi_\theta(\text{reward history}_{<t} \text{ of scenario}_i) + \epsilon_{\ell,i,t}\right), \quad (2)$$

where $\epsilon_{\ell,i,t} \sim \mathcal{N}(\mathbf{0}_d, \sigma^2 I)$ is a Gaussian noise. This procedure induces stochasticity while ensuring that each resulting policy $\pi_{\ell,i,t}$ is a valid policy.

After perturbation, $k$ trajectories with the *lowest regret* are selected and stored in the dataset $\mathcal{D}$ alongside their corresponding scenarios. The model is then updated by minimizing the following loss with respect to $\theta$:

$$\sum \sum_{t=1}^{T} \texttt{dist}\left(\pi_\theta(\text{reward history}_{<t} \text{ of scenario}_i), \pi_{\ell,i,t}\right), \quad (3)$$

where the outer sum is taken over $(\text{scenario}_i, C_{\ell,i} = (\pi_{\ell,i,t})_{t=1}^{T}) \in \mathcal{D}$. Here, $\texttt{dist}$ can be any suitable divergence metric (*e.g.*, the $\ell_2$ distance, the cross-entropy loss, or the KL divergence).

## 4.3. Experimental Results

We consider two policy spaces for training the Transformer: (**Env I**) the $\ell_2$-ball with $\Pi = B(\mathbf{0}_d, R_\Pi, \|\cdot\|_2)$ with radius $R_\Pi = 1$; (**Env II**) the probability simplex with $\Pi = \Delta(\mathcal{A})$ (where actions are sampled by applying the softmax function to the Transformer's output vector). During training, we adopt the $\ell_2$-distance as a $\texttt{dist}$ function in Equation (3). The Transformer is trained with $d = 3$ and time horizon $T = 25$, using reward vectors generated from the Gaussian reward (see Appendix G for more details). Full details on the training hyperparameters are provided in Appendix H. In the main body, we provide only results for the FOL environment. We provide the results of the MAB environment in Appendix K.2.

Interestingly, as illustrated below, for both **Env I** and **Env II** of FOL, the Transformer's output empirically converges to that of the known online learning algorithm, Follow-the-Regularized-Leader (FTRL), with various regularizers (see Appendix E for a formal introduction). Specifically, to analyze this phenomenon, we first express the linear attention architecture from Equation (1) as follows:

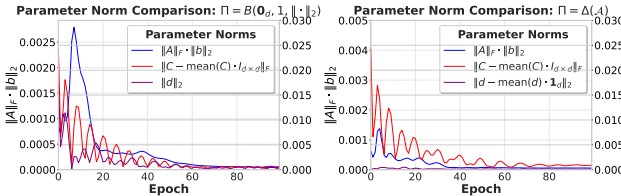

*Figure 1.* Evolution of the Transformer parameters with different policy spaces. **Left:** $\ell_2$-ball policy space $\Pi = B(\mathbf{0}_d, 1, \|\cdot\|_2)$ (**Env I**). **Right:** Simplex policy space $\Pi = \Delta(\mathcal{A})$ (**Env II**). The right y-axis is $\|\mathbf{C} - \text{mean}(\mathbf{C}) \cdot I_{d \times d}\|_F$, $\|\mathbf{d}\|_2$, and $\|\mathbf{d} - \text{mean}(\mathbf{d}) \cdot \mathbf{1}_d\|_2$ for **Env I** and **Env II**, respectively.

$$g((R_1, \ldots, R_{t-1}, \mathbf{1}_d)) = \texttt{Operator}\left(\sum_{\tau=1}^{t-1} \mathbf{A} R_\tau R_\tau^\mathsf{T} \mathbf{b} + \mathbf{C} R_\tau + \mathbf{d}\right),$$

where $\mathbf{A} := V$, $\mathbf{b} := K^\mathsf{T}(Q\mathbf{1}_d + q_c)$, $\mathbf{C} := k_c^\mathsf{T}(Q\mathbf{1}_d + q_c)V + v_c \mathbf{b}^\mathsf{T}$, and $\mathbf{d} := k_c^\mathsf{T}(Q\mathbf{1}_d + q_c)v_c$. We then track the evolution of the following quantities during training:

$$\|\mathbf{A}\|_F \|\mathbf{b}\|_2, \|\mathbf{C} - \text{mean}(\mathbf{C}) \cdot I_{d \times d}\|_F, \quad \|\mathbf{d}\|_2 \text{ (for \textbf{Env I}) or}$$
$$\|\mathbf{d} - \text{mean}(\mathbf{d}) \cdot \mathbf{1}_d\|_2 \text{ (for \textbf{Env II}),}$$

if these values converge to 0, then the architecture converges to known online learning algorithms. Specifically (using Equation (6)):

- When the `Operator` is *projection onto the $\ell_2$-ball*, it recovers **FTRL with $\ell_2$-regularization**,

- When the `Operator` is `Softmax` and $c = 0$, the architecture recovers the **Hedge algorithm** (*i.e.*, **FTRL with entropy regularization**),

since $\sum_{\tau=1}^{t-1}\left(\mathbf{A} R_\tau R_\tau^\mathsf{T} \mathbf{b} + \mathbf{C} R_\tau + \mathbf{d}\right) = c' \sum_{\tau=1}^{t-1} R_\tau$.

We visualize the convergence behaviors of these quantities in Figure 1 along the training iterations of our **ITERATIVE RMFT**. For **Env I**, we observe that $\|\mathbf{A}\|_F \|\mathbf{b}\|_2$, $\|\mathbf{C} - \text{mean}(\mathbf{C}) \cdot I_{d \times d}\|_F$, and $\|\mathbf{d}\|_2$ all converge to zero. This indicates that the Transformer's output asymptotically approximates that of FTRL with $\ell_2$-regularization. For **Env II**, we similarly observe that $\|\mathbf{A}\|_F \|\mathbf{b}\|_2$, $\|\mathbf{C} - \text{mean}(\mathbf{C}) \cdot I_{d \times d}\|_F$, and $\|\mathbf{d} - \text{mean}(\mathbf{d}) \cdot \mathbf{1}_d\|_2$ all approach approximately zero, suggesting that the Transformer's output behavior emerges to approximate that of the Hedge algorithm.

Our trained Transformer exhibits sublinear regret across unseen reward distributions (`Reward Generalization`), and over extended horizons up to $T = 100$ despite being trained only at $T = 25$ (`Horizon Generalization`). We defer a detailed analysis to Section K.

### 4.4. Theoretical Insight: From An Imitator to A Decision-Maker

Motivated by the empirical findings in Section 4.3, where we observe that the single-layer linear attention architecture empirically recovers known online learning algorithms (*e.g.*, Hedge, FTRL) under our post-training paradigm, we now provide a formal analysis to better understand this emergent behavior. We show that in a simplified setting of **ITERATIVE RMFT**, our training paradigm can indeed provably recover FTRL. This result offers theoretical support for **ITERATIVE RMFT**, by showing that the *imitation* of low-regret behaviors may yield a no-regret learning *algorithm*, which justifies the improved and generalizable no-regret behaviors observed empirically in Section 4.3.

For analytical convenience, we study a simple yet canonical subclass of Equation (1) obtained by removing `Operator`. We aim to learn model parameters that minimize the expected squared distance between the model's output and the *best-in-hindsight* policy:

$$\mathbb{E}\left[\sum_{t=1}^T \left\|g((R_1, \ldots, R_{t-1}, \mathbf{1}_d)) - \pi^\star(R_1, \ldots, R_T)\right\|_2^2\right], \quad (4)$$

where the expectation is taken over the randomness of $R_i$, and $\pi^\star(R_1, \ldots, R_t) \in \arg\max_{\pi \in \Pi}\left\langle \pi, \sum_{i=1}^t R_i \right\rangle$, which is a policy that maximizes the cumulative reward in hindsight.

Note that, minimizing the above loss function corresponds to an idealized regime of Algorithm 1, when an infinite number of trajectories $L \to \infty$ are sampled per scenario, and only the *lowest-regret* trajectories are selected. In this limit, the dataset used for supervised fine-tuning consists entirely of trajectories that are globally optimal (in terms of regret). We show that post-training the model on such a best-in-hindsight policy leads the model parameters to converge to those that implement FTRL. Thus, **ITERATIVE RMFT** in this regime does not merely imitate a good policy—it *provably emerges* as an *algorithm* for online DM. This result highlights the insight that **ITERATIVE RMFT**, when applied with sufficient samples and a well-structured inductive bias, may go beyond imitation learning, and recover principled algorithms for online DM. We defer a formal statement/proof of Theorem 1 to Appendix N.

**Theorem 1** ((Informal)). *Plugging in any global minimizer of Equation* (4) *within single-layer linear attention architecture and projecting the resulting output using* $\texttt{Proj}_{\Pi, \|\cdot\|}$ *yield an output from running FTRL with an $\ell_2$-regularizer and a stepsize of order $\Theta(1/\sqrt{Td})$.*

## 5. Training Open-Weight LLMs with ITERATIVE RMFT

We apply our **ITERATIVE REGRET-MINIMIZATION FINE-TUNING** framework to train open-weight LLMs in

tasks with both FOL and MAB environments. Compared to Section 4, which focused on Transformers with numerical input/output, we now explore more language-grounded numerical DM tasks, where we provide a language-based prompt to describe the numerical DM environment, and both input and output of the model are also language-based. Similar settings have also been considered recently in Park et al. (2025b); Krishnamurthy et al. (2024); Nie et al. (2025), to understand the in-context decision-making capabilities of LLM agents. We here focus on smaller open-weight LLMs to examine their *trainability*—whether they can be effectively trained under our ITERATIVE RMFT paradigm to function as decision-makers. To do so, we begin with language-grounded numerical DM examples that capture essential decision-making structures, before contrasting these results with those in more complex, real-world contexts later in Section 6. The instantiation of Algorithm 1 for training such models in language-grounded numerical DM tasks is presented in Algorithm 3, with a detailed explanation in Section 5.2.

## 5.1. Experimental Setup

At each round $t$, the model receives a natural language description of a DM problem together with the reward history $(R_1, R_2, \ldots, R_{t-1})$ or the history of the observed rewards $(\widehat{R}_1, \widehat{R}_2, \ldots, \widehat{R}_{t-1})$. The model is then expected to produce a natural language response specifying the policy $\pi_t$ to be applied at round $t$. We refer to this task as *language-grounded numerical DM*. We use the same numerical reward generation processes as described in Section 4.1. We defer the prompt design and its ablation to Appendix I. Each experiment trains the model for $T = 25$ rounds with $d = 3$ actions. For FOL environments, the training reward generation process is defined as a mixture of the Gaussian, Uniform, and Sine-trend rewards, whereas for MAB environments, we use the Gaussian reward. We train three open-weight LLMs – Phi-3.5-mini-instruct, Gemma-2-9b-it, and Qwen3-8B. Although these models are smaller and generally weaker than the proprietary closed-weight models, they remain among the most capable in their size category, exhibiting competitive performance in language understanding. All models are pre-trained and instruction-tuned, and are expected to effectively interpret natural language descriptions of the DM environments.

## 5.2. Instantiating ITERATIVE RMFT for Language-Grounded DM

We sample the output from the model using temperature $\tau = 1.0$ to encourage the diversity of the outputs, which will be used as training data. Note that this diversity affects the exploratory property of the training data, and is distinct from the exploration to be performed by the agent at inference time. The complete instantiation of Algorithm 1 is detailed in Algorithm 3. Unlike the numerical setting

described in Section 4.1, where a Transformer directly outputs numerical policies, the language-grounded trajectories consist of text outputs. These outputs implicitly define a policy, which is then parsed and executed in the environment to produce feedback. Post-training is then performed via supervised fine-tuning: minimizing the cross-entropy loss between the model's predicted token distributions and those from the top-$k$ low-regret trajectories. This training objective encourages the model to imitate high-performing behaviors and reasoning rationales (in terms of regret) encoded in the natural language form.

## 5.3. Experimental Results

Across all reported results in this section and its corresponding appendices, we assess performance using 50 trajectory samples for open-weight LLMs. We report the maximum and average values of the cumulative regret at round $T$, denoted by *max(LR)* and *avg(LR)*, as well as $\widehat{\beta}$, the estimate of the *growth rate* $\beta$ of the regret over time in Section 2.2, to validate the regret performance of the trained models/DM algorithms. The highlighted (yellow) cells indicate the lowest values of max(LR), avg(LR), and $\widehat{\beta}$ of the base model and the trained model. We also provide figures of the regret over time, and the final regret distributions from the trained and base models, with the Kolmogorov–Smirnov (KS) test results to compare them. A detailed explanation of the KS test is provided in Section D.4. A lower $p$-value of the KS test indicates stronger statistical evidence that the trained model yields lower regret. We further report 3-replicate robustness checks with per-setting error bars in Section H.2. For the MAB environment, we additionally provide the exploration metric SuffFailFreq($t$) before and after our training. As we adopt *action-based outputs* throughout this section, we argue that the exploitation metric MinFrac($t$) is already adequately low for the base model (see Figure 23), in line with the observations of Krishnamurthy et al. (2024), and therefore we omit this result here.

As baselines, we also include the performance of FTRL for the FOL environment and UCB for the MAB environment. Importantly, note that this comparison is in favor of these baseline algorithms, as they avoid language understanding and directly take the *numeric* descriptions of the DM environments (*i.e.,* reward vectors/values) as input, for which they are known to be asymptotically optimal in terms of regret minimization. Hence, these algorithms should be viewed as *reference points* when the numeric descriptions of the problem are perfectly accessible, without distraction or ambiguity due to language descriptions, instead of some baselines to be *beaten* by our language-grounded, post-trained LLM agents. Finally, in terms of the generalizability of the trained LLM-agents, our evaluation will focus on two axes: Reward Generalization and Horizon

`Generalization.`

**Full-Information Online Learning.** We evaluate the `Horizon Generalization` and `Reward Generalization` performance of our trained model with $d = 3$ actions and a time horizon of $T = 50$, across multiple different reward generation processes on Gemma-2-9b-it. We first provide the result using the same reward generation process as the training one in Table 2 (which corresponds to `Horizon Generalization`). The regret over time and the final regret distribution can also be found in Figure 17. The trained model exhibits lower regrets than the base model in all three reward generation processes, which demonstrates the effectiveness of **ITERATIVE RMFT**. We also evaluate the trained Gemma-2-9b-it on the Alternating, Bernoulli, and Noisy Alternating rewards in Figures 18 to 20 (which corresponds to both `Horizon Generalization` and `Reward Generalization`). We observe lower regret values for the trained model for all the reward generation processes.

**Multi-Armed Bandits.** Similar to the FOL environment, we evaluate both `Horizon Generalization` and `Reward Generalization` [Gaussian → Gamma] performance with $d = 3$ actions on both Gemma-2-9b-it and Qwen3-8B models. Because the two LLMs differ in their context length limits, we set the time horizon to $T = 50$ for Gemma-2-9b-it and $T = 100$ for Qwen3-8B. The Gemma-2-9b-it model is limited to a context length of 8192 tokens; therefore, we cannot run experiments at $T = 100$. We provide the experimental results of the Gamma reward in Figure 2. We find that the model achieves lower regret values and slower regret growth rates and also exhibits improved exploration, as evidenced by the reduced `SuffFailFreq`$(t)$ at smaller time steps $t$, and also exhibits good `Horizon Generalization` performance. We also provide results for the Gaussian reward on both Gemma-2-9b-it and Qwen3-8B models in Figures 21 and 22. We provide detailed examples and quantitative analysis on the reasoning rationales in Appendix L.5.

**AD and GRPO Baselines.** We further compare **ITERATIVE RMFT** against algorithm distillation (AD) (Laskin et al., 2023; Nie et al., 2025) and GRPO-style baselines (Shao et al., 2024; Guo et al., 2025) on the Qwen3-8B MAB experiment with a UCB teacher (Auer et al., 2002a) in Table 3, where all methods are trained with horizon $T = 25$ and evaluated under `Horizon Generalization` at $T = 100$. Let $I$ denote the number of iterations, $M$ the number of instances per iteration, $L$ the number of samples per instance, and $k$ the number of selected trajectories. Thus, **ITERATIVE RMFT** uses $IMk$ trajectories for SFT after generating $IML$ trajectories.

This comparison suggests that the gains of **ITERATIVE RMFT** are not explained solely by the generation budget: although **ITERATIVE RMFT** generates more trajectories than $AD_{IMk}$ in order to perform selection, it outperforms $AD_{IML}$ with a matched generation budget and $AD_{2IML}$ with twice the generation budget. The efficiency likely comes from two factors: selective SFT on low-regret trajectories induces minimal policy drift, while AD trained to imitate UCB at $T = 25$ suffers from a horizon mismatch when evaluated at $T = 100$, since UCB's exploration schedule is coupled with the history length. We provide additional parameter-sensitivity ablations in Section H.5.

## 6. Training the Closed-Weight LLM with ITERATIVE RMFT

In this section, we apply the **ITERATIVE REGRET-MINIMIZATION FINE-TUNING** framework to post-train the closed-weight LLM GPT-4o mini, which does not yet expose its full model weights but supports *supervised fine-tuning* via API access. Notably, our **ITERATIVE RMFT** is inherently compatible with such a supervised fine-tuning paradigm (see Line 9 in Algorithm 1), making it possible for us to post-train such proprietary, but arguably stronger, closed-weight models for DM. Unlike the training for open-weight LLMs in Section 5, which focused on *language-grounded numerical DM* (Section 5.1), thanks to the stronger language ability of the closed-weight LLMs, we now consider more complex and context-rich DM scenarios expressed in natural language, which we term *language-grounded DM with real-world contexts*.

Although closed-weight LLMs have demonstrated strong general-purpose language capabilities, prior studies Park et al. (2025b); Krishnamurthy et al. (2024); Nie et al. (2025) have shown that such models may struggle even in simple language-grounded DM tasks with specific contexts. Indeed, these studies have largely relied on simple and *fixed* examples of contexts (*e.g.*, button-clicking (Krishnamurthy et al., 2024)) that lack the diversity and complexity needed for the generalization to more real-world DM tasks. In contrast, our goal is to fine-tune these models, via our **ITERATIVE RMFT**, to be more generalizable to a wider range of language-based DM tasks with diverse real-world contexts. To this end, we first need to curate new language-grounded DM datasets, the systematic generation of which may be a contribution of independent interest.

### 6.1. Experimental Setup: Language-grounded DM with Real-World Contexts

To systematically create diverse language-grounded DM tasks with real-world contexts, we generate reward and linguistic context independently.

We adopt a unified procedure to construct linguistic contexts for both the full-information online learning (FOL)

| | **Gaussian** | | | **Uniform** | | | **Sine-trend** | | |
|---|---|---|---|---|---|---|---|---|---|
| | max(LR) | avg(LR) | $\widehat{\beta}$ | max(LR) | avg(LR) | $\widehat{\beta}$ | max(LR) | avg(LR) | $\widehat{\beta}$ |
| FTRL | 49.70 | 27.99 | 0.81 | 55.04 | 38.90 | 0.75 | 54.33 | 12.32 | 0.31 |
| Gemma-2-9b-it | 137.19 | 20.62 | 0.87 | 227.48 | 28.47 | 0.81 | 186.81 | 18.00 | 0.48 |
| Trained Gemma-2-9b-it | 93.08 | 20.45 | 0.80 | 122.45 | 12.05 | 0.57 | 139.79 | 13.28 | 0.38 |

*Table 2.* **Summary of the regret values for the FOL environment evaluated with `Horizon Generalization` [$T = 25 \rightarrow T = 50$] on the Gemma-2-9b-it model**, trained and tested on a mixture of the Gaussian, Uniform and Sine-trend rewards. The results with the Uniform and Sine-trend rewards both demonstrate a lower regret value and sublinear regret growth after **ITERATIVE RMFT**, while that of the Gaussian reward shows a better maximum regret value and sublinear regret growth, but does not clearly lower the average regret value.

| Method | Base | $\text{AD}_{IMk}$ | $\text{AD}_{IML}$ | $\text{AD}_{2IML}$ | $\text{GRPO}_{\text{step}}$ | $\text{GRPO}_{\text{regret}}$ | Ours |
|---|---|---|---|---|---|---|---|
| Avg Regret ($T = 100$) | 52.10 | 58.72 | 33.44 | 32.97 | 35.88 | 40.09 | **22.37** |

*Table 3.* **Comparison with algorithm distillation and GRPO-style baselines on Qwen3-8B in the MAB environment.** All methods are trained with $T = 25$ and evaluated at $T = 100$ using the UCB teacher setup. **ITERATIVE RMFT** substantially outperforms AD with the matched selected-data budget ($IMk$), AD with the matched generation budget ($IML$), AD with twice the generated budget ($2IML$), and both GRPO variants. The $\text{GRPO}_{\text{regret}}$ run uses the $IMk$ budget due to time constraints.

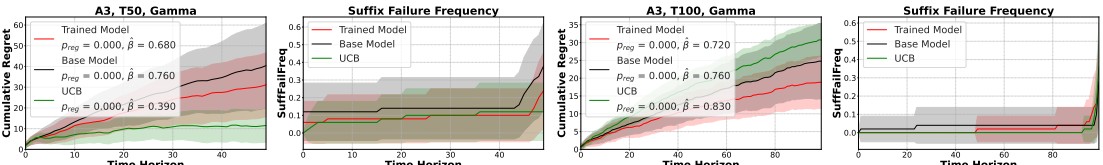

*Figure 2.* **The regret over time, and the exploration metric using `SuffFailFreq`($t$) for the MAB environment:** The left two subplots are under both `Horizon Generalization`[$T = 25 \rightarrow T = 50$] and `Reward Generalization`[Gaussian $\rightarrow$ Gamma] on Gemma-2-9b-it, while the right two subplots are under both `Horizon Generalization`[$T = 25 \rightarrow T = 100$] and `Reward Generalization`[Gaussian $\rightarrow$ Gamma] on Qwen3-8B. Both models show a lower regret value, sublinear regret, and improved *E-E tradeoff* after **ITERATIVE RMFT**.

| | **Gaussian** | | | **Uniform** | | | **Sine-trend** | | |
|---|---|---|---|---|---|---|---|---|---|
| | max(LR) | avg(LR) | $\widehat{\beta}$ | max(LR) | avg(LR) | $\widehat{\beta}$ | max(LR) | avg(LR) | $\widehat{\beta}$ |
| FTRL | 39.59 | 27.21 | 0.64 | 39.15 | 24.16 | 0.75 | 38.09 | 11.32 | 0.43 |
| GPT-4o mini | 57.36 | 27.42 | 0.67 | 70.82 | 22.28 | 0.74 | 40.62 | 11.24 | 0.43 |
| Trained GPT-4o mini | 53.29 | 25.63 | 0.62 | 39.65 | 17.09 | 0.65 | 39.64 | 9.83 | 0.38 |

*Table 4.* **Summary of the regret value for the FOL environment under `Horizon Generalization`[$T = 15 \rightarrow T = 25$] on GPT-4o mini,** trained and evaluated on a mixture of the Gaussian, Uniform, and Sine-trend rewards. The trained model consistently shows improved regret behavior in a longer horizon.

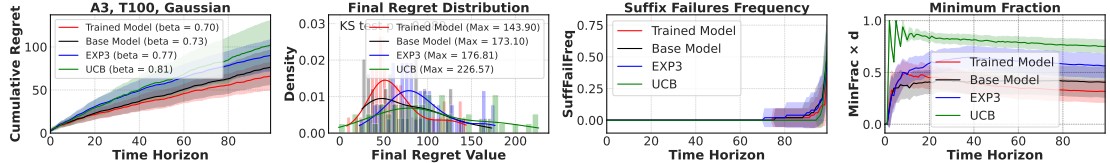

*Figure 3.* **The regret over time, the final regret distribution, and the exploration and exploitation metrics for tasks with the MAB environment under `Horizon Generalization`[$T = 25 \rightarrow T = 100$] on GPT-4o mini**, trained and evaluated with the Gaussian reward, which shows a lower regret and sublinear regret behavior for the trained model. The trained GPT-4o mini model sustains effective exploration and adapts its exploitation strategy over time.

| | **LC=Gemini, T=50, A=3** | | | **LC=GPT, T=100, A=3** | | | **LC=GPT, T=100, A=5** | | |
|---|---|---|---|---|---|---|---|---|---|
| | max(LR) | $\widehat{\beta}$ | SuffFailFreq | max(LR) | $\widehat{\beta}$ | SuffFailFreq | max(LR) | $\widehat{\beta}$ | SuffFailFreq |
| Rexp3 | 214.10 | 0.89 | 0.32 | 403.45 | 0.86 | 0.26 | 560.95 | 0.88 | 0.46 |
| GPT-4o mini | 277.74 | 0.88 | 0.22 | 427.91 | 0.96 | 0.18 | 453.72 | 0.90 | 0.32 |
| Trained GPT-4o mini | 204.74 | 0.86 | 0.22 | 370.02 | 0.91 | 0.22 | 418.87 | 0.89 | 0.18 |

*Table 5.* **Summary of max(LR), $\widehat{\beta}$, and `SuffFailFreq`($0.98T$) for the NS-MAB environment,** trained on the Gradual Variation reward with horizon $T = 25$, evaluated on various settings with different linguistic contexts (*LC*), time horizon (*T*), and action space size (*A*). *LC=Gemini* means the linguistic context is generated by Gemini 2.0-Flash; *LC=GPT* means the linguistic context is generated by GPT-4o mini. The trained model improves max(LR) and $\widehat{\beta}$ in all three settings; for `SuffFailFreq`, it improves in one (*LC=GPT, T=100, A=5*), ties in one (*LC=Gemini, T=50, A=3*), and is slightly higher than the base model in one (*LC=GPT, T=100, A=3*).

and (non-)stationary multi-armed bandit (MAB/NS-MAB) environments. Using the GPT-4o mini API, we synthesize scenario descriptions that specify the common action set shared by the environments. Manually designed prompt templates span domains such as healthcare, business strategy, and science fiction, and they explicitly instruct GPT-4o mini to describe multiple plausible actions so that no candidate is obviously optimal. We further randomize the narrative style—ranging from concise reports to analytical dilemmas—to encourage linguistic and structural diversity. These prompts apply verbatim across FOL and MAB/NS-MAB tasks; the only difference is how the generated scenarios are paired with subsequent reward signals. Complete prompt templates appear in Section J, and we validate that this richer prompt induces substantially more diverse linguistic contexts than a simple prompt baseline in Section J.1.

We provide illustrative examples of the language-grounded DM scenarios used in our experiments in Figure 4.

### 6.2. Experimental Results

Similar to Section 5.3, we report *max(LR)*, *avg(LR)*, the *growth rate estimate* of regret $\widehat{\beta}$, and for tasks with MAB environments, we additionally report $\texttt{SuffFailFreq}(t)$ and $\texttt{MinFrac}(t)$ (Section 2.2). Further training details are provided in Appendix H.6. The evaluation focuses on: (a) In-Distribution performance, to confirm whether effective training has occurred, and (b) four axes of generalization: Reward Generalization and Horizon Generalization as in Section 5, as well as Linguistic Context Generalization (*i.e.,* testing under diverse real-world description distributions), and Action Space Size Generalization. Notably, the last two generalization dimensions are unique to language-grounded DM with real-world contexts and do not typically arise in numerical DM settings. Our intention is not to claim superior performance of the trained model over the numerical algorithms, as the classical algorithms directly take *numeric* values as input and do not need to parse language descriptions. The classical algorithms only serve as references. We also provide examples of the reasoning rationales in Appendix M.4.

**Full-Information Online Learning.** The model is trained with $d = 3$ actions and a time horizon of $T = 15$ using a mixture of the Gaussian, Uniform, and Sine-trend rewards, with policy space $\Pi = \Delta(\mathcal{A})$. We evaluate Reward Generalization by altering the reward generation processes to the Bernoulli and Alternating rewards, evaluate Horizon Generalization by extending the time horizon to 25, evaluate Linguistic Context Generalization by generating linguistic contexts by Gemini 2.0-Flash rather than GPT-4o mini, and evaluate Action Space Size Generalization by in-

creasing the number of actions to 4. Overall generalization results are summarized in Tables 4 and 14 to 16. Across these diverse and unseen conditions, the trained model consistently demonstrates competitive performance.

**Multi-Armed Bandits.** We train our model using $d = 3$ actions and a short horizon $T = 25$ using the Gaussian reward. First, we evaluate the Horizon Generalization[$T = 25 \rightarrow T = 100$] performance and report the results in Figure 3, which presents $\texttt{SuffFailFreq}(t)$ and $\texttt{MinFrac}(t)$. In particular, $\texttt{MinFrac}(t)$ reveals a key difference in exploitation strategies: the base GPT-4o mini model exhibits *uniform-like failures* (Krishnamurthy et al., 2024), *i.e.,* the agent continues to select all arms at approximately equal rates without effectively eliminating the suboptimal choices. In contrast, the trained GPT-4o mini model initially explores broadly (higher $\texttt{MinFrac}(t)$) but later concentrates its selections, as indicated by the decline in $\texttt{MinFrac}(t)$, reflecting a transition from exploration to exploitation, similar to the behaviors of known online DM algorithms of EXP3 and UCB. In addition, as shown in Figure 3, the trained GPT-4o mini model continues to exhibit favorable regret performance, achieving a lower maximum regret. Then, we evaluate Reward Generalization by altering the reward generation processes to the Bernoulli one, and evaluate Linguistic Context Generalization by generating linguistic contexts with Gemini 2.0-Flash. We also increase the action space to 4 and 5 with longer horizons to further evaluate Action Space Size Generalization (see Figures 26 to 28).

**Non-Stationary Multi-Armed Bandits.** We train our model using $d = 3$ actions and a short horizon $T = 25$ with the Gradual Variation reward generation process. The trained GPT-4o mini model sustains effective exploration and adapts its exploitation strategy over time. Our result shows an improved regret growth rate (from 0.96 to 0.91) compared to the base model with Horizon Generalization. Note that the Gradual Variation reward used here yields a regret lower bound of $\Omega(T^{5/6})$, obtained by substituting $V_T = \Theta(T^{1/2})$ (see Appendix G) into the lower bound in Besbes et al. (2014). As such, the trained model sustains competitive performance under this non-stationary environment. Furthermore, we report $\texttt{SuffFailFreq}(0.98T)$ values, indicating that the model more reliably identifies the optimal arm near the end of the horizon compared to the base model. The quantitative results are summarized in Table 5, and for completeness, corresponding plots are provided in Figures 29 to 31.

## Acknowledgment

The authors would like to thank Jisu Jang for the feedback on the figures. C.P. acknowledges the support from the Amazon AI PhD Fellowship and Korea Foundation for Advanced Studies Scholarship. A.O. acknowledges this work was supported by the Department of the Navy, Office of Naval Research, under grant 036388-00002. Z.C. and K.Z. acknowledge the support from the Army Research Office (ARO) grant W911NF-24-1-0085, the NSF CAREER Award 2443704, a Coefficient Giving AI Safety Research Award, and the University of Maryland supercomputing resources (http://hpcc.umd.edu) made available for conducting the research reported in this paper. K.Z. additionally acknowledges the support from the AFOSR YIP Award FA9550-25-1-0258, a Cisco Faculty Research Award, and a JP Morgan Faculty Research Award.

## Impact Statement

This work studies how LLMs can be post-trained to act as better decision-making agents, by leveraging regret minimization as a training objective. As LLM-powered agents are increasingly deployed in interactive and sequential settings, including recommendation systems, automated assistants, resource allocation, and decision support tools, their ability to make principled decisions over time has direct societal implications.

**Potential Benefits.** A primary positive impact of this work is the development of a more *theoretically grounded* approach to improving LLM decision-making behavior. By using regret—a well-established metric in online learning and decision theory—our method promotes behaviors that balance exploration and exploitation, adapt to feedback, and generalize across changing environments. Such properties are desirable in real-world systems where decisions unfold sequentially and feedback is partial or delayed. In safety-critical or high-stakes domains, regret-aware decision-making may help reduce persistent suboptimal behavior, improve robustness to distribution shifts, and mitigate brittle or overly greedy policies that can arise from reward-only optimization.

Additionally, our framework is model-agnostic and compatible with both open-weight and closed-weight LLMs, making it broadly applicable without requiring hand-designed algorithms or domain-specific heuristics. This lowers the barrier for incorporating principled decision-making behavior into deployed systems and may help align LLM agents more closely with long-term performance objectives rather than short-term gains.

**Risks and Limitations.** Despite these benefits, improving the decision-making capabilities of LLM agents also carries risks. More capable decision-makers may amplify existing harms if deployed in inappropriate contexts, such as manipulative recommendation systems, strategic persuasion, or adversarial economic settings. While regret minimization encourages long-term performance, it does not by itself encode ethical constraints, fairness objectives, or alignment with human values. As a result, a regret-minimizing agent could still learn to exploit loopholes or pursue undesirable strategies if the underlying environment or reward structure is poorly specified.

Furthermore, our experiments primarily focus on synthetic or procedurally generated environments, in order to conduct *controlled* experiments. Although these settings are useful for isolating decision-making behavior, they do not fully capture the complexity of real human-in-the-loop systems. Overgeneralizing the empirical results without careful domain-specific evaluation could lead to misplaced trust in the robustness of trained agents.

**Mitigation and Responsible Use.** We view this work as a foundational step toward principled post-training for LLM decision-making, rather than a turnkey solution for deployment. Regret-based training should be combined with complementary safeguards, such as explicit safety constraints, human oversight, and alignment-oriented objectives, especially in socially sensitive applications. We encourage future work to study how regret minimization interacts with fairness, interpretability, and value alignment, as well as to evaluate such agents in realistic, human-centered decision environments.

Overall, we believe this work contributes positively by clarifying how established decision-theoretic principles can be integrated into LLM post-training, while highlighting the importance of cautious and responsible deployment.

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

# A. Deferred Conclusion, Limitations, and Future Directions

In this paper, we introduced ITERATIVE REGRET-MINIMIZATION FINE-TUNING, a regret-driven post-training paradigm that turns self-generated trajectories into a supervised fine-tuning signal for training better decision-making LLM agents. Empirically, ITERATIVE RMFT consistently lowered the regret values across three model classes: Transformers with numerical input/output, open-weight LLMs (Qwen3-8B and Gemma-2-9b-it) operating in language-grounded numerical tasks, and a closed-weight LLM (GPT-4o mini) fine-tuned in linguistically rich, real-world DM scenarios. Across these settings, we observed that the post-trained LLMs can achieve lower regret growth over time, a better exploration-exploitation tradeoff, and generalization to longer horizons, new reward distributions, and novel language descriptions.

This study was constrained by several practical considerations. The training was limited to relatively short horizons (although the trained model, as an *algorithm*, exhibited good performance on longer horizons at inference time), due to API budget and context-length limits. Moreover, our tasks remained mostly synthetic or procedurally generated, leaving the performance evaluation on real human-in-the-loop applications as important future work. Other future directions include training ITERATIVE RMFT on genuinely long-horizon interactions, generalizing to other DM environments with richer structures such as contextual bandits and Markov decision processes, augmenting the ITERATIVE RMFT trained models with agentic workflows and other inference-time interventions to further enhance their DM ability, and evaluating the trained LLM agents on more real-world DM applications such as tool-use, web browsing, and software engineering. Overall, we position our approach as an initial exploration, calling for more principled and novel post-training paradigms for LLM agents when it comes to addressing decision-making tasks.

# B. Additional Related Work

**LLM-Agents for Real-World Decision-Making.** The reasoning capabilities of LLMs have seen significant improvement due to the recent progress in pre-training (OpenAI, 2023; Bubeck et al., 2023), post-training (Guo et al., 2025; Park et al., 2025a), and prompting methods (Wei et al., 2022; Yao et al., 2023a). Leveraging these substantial developments, there has been an emerging trend to position LLMs as the *central controller* of autonomous agents for *decision-making* (Yao et al., 2023b; Shinn et al., 2024; Sumers et al., 2024). One major benefit of utilizing LLMs as controllers is their capacity to parse arbitrary language inputs and generate natural-language-based outputs, allowing for a highly flexible and adaptable interaction interface for real-world applications. As an example, a notable strand of research investigated LLM agents through the lens of planning (Hao et al., 2023; Valmeekam et al., 2023; Huang et al., 2022; Shen et al., 2023), an approach that has also been adopted in embodied-AI and robotic systems (ichter et al., 2023; Wang et al., 2023; Significant Gravitas, 2023; Wang et al., 2024). In parallel, another line of work concentrated on developing LLM agents tailored to specific domains, such as software engineering (Yang et al., 2024; Wang et al., 2025), enterprise operations (Drouin et al., 2024; Boisvert et al., 2024), healthcare (Kim et al., 2024), and cybersecurity (Zhang et al., 2025a).

**Understanding LLM-Agents in Canonical Settings.** The aforementioned progress on LLM-agents for decision-making primarily focused on tailoring LLMs to solve complex but specific real-world tasks. However, it remains unclear *if* and *why* LLMs are good decision-makers, especially given that they were not designed for DM. The high complexity in real-world tasks makes it challenging to diagnose these questions, inspiring a series of works on understanding LLMs in basic, canonical DM settings. Wu et al. (2024); Krishnamurthy et al. (2024); Park et al. (2025b) first carefully studied the in-context online DM ability of LLMs, with Wu et al. (2024); Krishnamurthy et al. (2024) focusing on the exploration ability in stochastic MABs, and Park et al. (2025b) focusing on the online adversarial and game-theoretic settings. Subsequently, Nie et al. (2025) expanded the bandit settings considered in Krishnamurthy et al. (2024), deepening the understanding of LLMs' (in)ability for efficient exploration. Further, Zhang et al. (2025b) examined both the MAB and the NS-MAB environments, focusing on the comparison with human behaviors; Xia et al. (2025) studied the dueling bandit setting when explicit numerical rewards are not observable. Other works that also examined LLMs in canonical bandit environments are Felicioni et al. (2024); Rahn et al. (2024); Monea et al. (2025). In particular, Nie et al. (2025); Rahn et al. (2024); Monea et al. (2025); Sun et al. (2025) also developed *training-free* interventions to enhance the DM ability of LLMs.

**Post-Training LLMs for Decision-Making.** In light of the observations above, several studies have also proposed *(post-)training* methods to explicitly enhance LLMs' DM abilities. Nie et al. (2025); Schmied et al. (2026) represent the recent efforts in post-training/fine-tuning LLMs as better decision-makers, aligning with the motivation of the present paper. Specifically, Nie et al. (2025) developed a fine-tuning approach based on *algorithm distillation* (Laskin et al., 2023), which requires *pre-defined* input/output formats, *e.g.*, the action space size/reward vector dimension and time horizon, and

thus has limited generalizability to DM tasks with rich scenarios, real-world contexts, and variable problem structures. Moreover, it may not be applicable to scenarios when the optimal/expert algorithms are unknown/unclear, as is common in real-world applications. Finally, the approach did not incorporate nor enhance the *CoT reasoning rationales* of the LLMs in DM. Very recently, Schmied et al. (2026) introduced reinforcement learning fine-tuning (RLFT) on self-generated CoT paths and further compared it against algorithm distillation. RLFT is closely related to ITERATIVE RMFT in that both can incorporate self-generated reasoning rationales in the training data. In comparison, ITERATIVE RMFT uses the cumulative regret of the entire trajectory as the training signal, while RLFT uses the rewards per se. As (cumulative) reward maximization is not identical to regret minimization in general, it is unclear if such an approach applies to the online learning (with adversaries) or non-stationary bandit settings we consider. The tasks focused on are also different: RLFT addressed MABs with two specific DM scenarios from Nie et al. (2025), also with extensions to contextual bandits (CBs) and tic-tac-toe; ITERATIVE RMFT focused on tasks with the MAB environment with more generic DM scenarios, as well as those with online learning and non-stationary bandit environments. Thanks to the flexibility and unification offered by the regret metric, ITERATIVE RMFT trained LLMs also exhibit generalizability in various problem structures. Finally, Schmied et al. (2026) also demonstrated the promise of SFT-based training on CoT rationales (as ITERATIVE RMFT does), although the rationales were manually designed to mimic the UCB algorithm (Auer et al., 2002a), rather than being organically and automatically generated by the LLMs.

**Post-Training LLMs for *Decision-Making* vs. for *Reward Maximization*.** Our focus in this paper shall not be confused with the recent line of *RL-based post-training* paradigms for LLMs, for tasks such as mathematical reasoning (Jaech et al., 2024; Guo et al., 2025; Shao et al., 2024) and alignment (Ouyang et al., 2022; Rafailov et al., 2023). In those paradigms, LLMs are trained to *maximize certain reward functions* (under regularization) associated with the specific task, which reflect the correctness and/or human-preference of the responses, using RL methods. Notably, the reward maximization objective per se cannot elicit *efficient exploration* automatically, which is precisely the inspiration of many variants that explicitly incorporated exploration mechanisms in such RL-based *training process*, see *e.g.,* Xie et al. (2025); Du et al. (2024); Dwaracherla et al. (2024). In contrast, our focus is on post-training LLMs *for decision-making* as an *ability*, so that they may be able to address novel rewards/tasks at *inference time* efficiently. To this end, the post-training should provide signals to elicit LLMs to *learn to explore* at inference time, ideally with *generalizability* across task specifications like the action (sets), rewards, horizons, etc. Our approach resorts to the regret metric as a possible source of such signals.

# C. Additional Notation

We use $\mathbb{R}$ to denote the set of real numbers, $\mathbb{R}^+$ to denote the set of non-negative real numbers, $\mathbb{N}$ for the set of non-negative integers, and $\mathbb{N}^+$ for the set of positive integers. For any finite set $\mathcal{S}$, let $\Delta(\mathcal{S})$ denote the probability simplex over $\mathcal{S}$. Given a positive integer $d \in \mathbb{N}^+$, we define the shorthand notation $[d] := \{1, 2, \ldots, d\}$. For vectors $x, y \in \mathbb{R}^d$, we write $\langle x, y \rangle$ to denote their standard Euclidean inner product. Let $\mathbf{0}_d$ and $\mathbf{1}_d$ represent the $d$-dimensional all-zero and all-one vectors, respectively. We also let $O_{d \times d}$ and $I_{d \times d}$ denote the $d \times d$ zero matrix and identity matrix, respectively. When the dimension $d$ is clear from context, we omit the subscript. The vector $e_i$ denotes the $i$-th standard basis (unit) vector in $\mathbb{R}^d$. Given a vector $p \in \mathbb{R}^d$, a radius $R > 0$, and a norm $\|\cdot\|$, the closed ball centered at $p$ with radius $R$ is denoted by $B(p, R, \|\cdot\|) := \{x \in \mathbb{R}^d \mid \|x - p\| \le R\}$. For a convex set $C \subseteq \mathbb{R}^d$, we define the projection of $p$ onto $C$ under norm $\|\cdot\|$ as $\texttt{Proj}_{C, \|\cdot\|}(p) := \arg\min_{x \in C} \|x - p\|$, which is well-defined due to the convexity of $C$. The softmax function is defined by $\texttt{Softmax}(x) := \left( \frac{e^{x_i}}{\sum_{j \in [d]} e^{x_j}} \right)_{i \in [d]}$. For a matrix $A \in \mathbb{R}^{m \times n}$ with columns $A_i \in \mathbb{R}^m$, we define the Frobenius norm $\|A\|_F := \sqrt{\text{Tr}(A^\intercal A)}$. We use $A_{-1} := A_n$ to denote the last column of $A$. For any set $\Pi \subseteq \mathbb{R}^d$, we define its diameter under a norm $\|\cdot\|$ as $\text{diam}(\Pi, \|\cdot\|) := \sup_{\pi_1, \pi_2 \in \Pi} \|\pi_1 - \pi_2\|$. The indicator function $\mathbb{1}(\mathcal{E})$ equals 1 if the event $\mathcal{E}$ is true, and 0 otherwise. For functions $f, g : \mathbb{R} \to \mathbb{R}$, we write $g(x) = \mathcal{O}(f(x))$ if there exist constants $x_0 \in \mathbb{R}$ and $M < \infty$ such that $|g(x)| \le M|f(x)|$ for all $x > x_0$. For a sequence $(\ell_t)_{t \in [T]}$ with $T \in \mathbb{N}^+$, we define the sub-sequence $\ell_{a:b} := (\ell_a, \ldots, \ell_b)$ for indices $1 \le a \le b \le T$. Let $\mathbb{O}(d)$ denote the orthogonal group of $d \times d$ real matrices, *i.e.*, $\mathbb{O}(d) = \{Q \in \mathbb{R}^{d \times d} \mid Q^\intercal Q = I_d\}$. We use $n!$ to denote the factorial of $n$, and $n!!$ to denote the double factorial, *i.e.*, the product of all integers from $n$ down to 1 that have the same parity as $n$. The Gamma function $\Gamma(z)$ generalizes the factorial and is defined for complex $z$ with positive real part as $\Gamma(z) = \int_0^\infty t^{z-1} e^{-t} dt$.

# D. Deferred Preliminaries

## D.1. Decision-Making Environments

**Full-Information Online Learning.** In the FOL environment, an agent interacts with the environment over $T$ rounds by sequentially making decisions based on the feedback received from prior rounds. At each round $t \in [T]$, the agent selects a decision policy $\pi_t \in \Pi$, where $\Pi$ denotes a bounded decision policy space. After the agent commits to $\pi_t$, the environment reveals a bounded reward function $f_t : \Pi \to [-B, B]$ for some constant $B > 0$, possibly chosen in an adversarial manner. The agent then receives some feedback about $f_t(\pi_t)$ (*e.g.*, $\nabla f_t(\pi_t)$), and updates the policy to $\pi_{t+1}$ based on the observed feedback so far (Cover, 1966; Vovk, 1990; Littlestone & Warmuth, 1994; Hazan, 2016). A common instantiation of this framework is when $\Pi = \Delta(\mathcal{A})$, the probability simplex over a finite action set $\mathcal{A}$, and the reward function is linear in the policy: $f_t(\pi_t) = \langle R_t, \pi_t \rangle$, where $R_t \in \mathbb{R}^d$ is a reward vector. This case is also referred to as the *Experts Problem* (Littlestone & Warmuth, 1994), which may also be used to model agents' learning in repeated games (Cesa-Bianchi & Lugosi, 2006). Alternatively, $\Pi$ may be an $\ell_2$-Euclidean-ball $B(\mathbf{0}_d, R_\Pi, \|\cdot\|_2)$ for some $R_\Pi > 0$, and the reward takes the same linear form. In either case, we assume the agent receives the full reward vector $R_t$ at each round $t$.

**Multi-Armed Bandits.** In the MAB environment, an agent interacts with the environment over $T$ rounds, sequentially selecting actions and updating its strategy based on partial feedback (Lattimore & Szepesvári, 2020). The agent has access to an *action set* $\mathcal{A}$ with $d := |\mathcal{A}|$, where each element corresponds to a distinct "arm." At each round $t \in [T]$, a reward vector $R_t \in \mathbb{R}^d$ is sampled from some underlying reward generation distribution with mean $r \in [-B, B]^d$. The agent chooses an action $a_t \in \mathcal{A}$, and upon committing to this action, the environment reveals only the sampled reward associated with that arm, $R_t(a_t)$. Thus, unlike the FOL environment, the agent receives only *bandit feedback*—a single scalar reward $R_t(a_t)$—rather than the entire reward vector $R_t$.

**Non-Stationary Multi-Armed Bandits.** In the NS-MAB environment, an agent operates with the same action set $\mathcal{A}$, but the mean reward of each arm is allowed to evolve over time. Formally, at each round $t \in [T]$, the environment generates a reward vector $R_t$ with mean $r_t \in [-B, B]^d$, where the sequence $(r_t)_{t \in [T]}$ may drift. A common assumption is that this drift is controlled by a variation budget $V_T := \sum_{t=2}^{T} \|r_t - r_{t-1}\|_\infty$, which captures the total amount of non-stationarity (Besbes et al., 2014). The agent still chooses an action $a_t \in \mathcal{A}$ and only observes the realized reward $R_t(a_t)$. The objective is therefore to adapt to the changing reward landscape while learning from bandit feedback.

Both MABs and NS-MABs belong to, and will be referred to as, online DM problems with *bandit/partial* feedback, in contrast to the *full-information* feedback in the FOL environment.

## D.2. Performance Metric: Regret

**Full-Information Online Learning.** In the FOL environment with reward vectors $(R_t)_{t \in [T]}$ and decision space $\Pi$, the regret of a DM algorithm $\mathscr{A}$—which generates a sequence of policies $\pi_{\mathscr{A},t} \in \Pi$—is defined as

$$\text{Regret}_{\mathscr{A}}\big((R_t)_{t \in [T]}, T\big) := \max_{\pi \in \Pi} \sum_{t=1}^{T} \langle \pi, R_t \rangle - \sum_{t=1}^{T} \langle \pi_{\mathscr{A},t}, R_t \rangle.$$

This expression compares the cumulative reward of the best fixed decision in hindsight against that of the algorithm's chosen decisions over time.

**Multi-Armed Bandits.** In the MAB environment, the *expected regret* of algorithm $\mathscr{A}$, which generates an action sequence $(a_{\mathscr{A},t})_{t=1}^{T}$ of length $T$, is defined as $\text{Regret}_{\mathscr{A}}(r, T) := \mathbb{E}\big[T \cdot \max_{a \in \mathcal{A}} r(a) - \sum_{t=1}^{T} r(a_{\mathscr{A},t})\big]$. The regret notion quantifies the difference between the cumulative expected reward that would have been obtained by selecting the best arm, and the expected reward accumulated by $\mathscr{A}$ through its chosen actions.

**Non-Stationary Multi-Armed Bandits.** In the NS-MAB environment, the common metric is *dynamic regret* (Besbes et al., 2014), which, for an algorithm $\mathscr{A}$ that generates an action sequence $(a_{\mathscr{A},t})_{t=1}^{T}$, is defined as

$$\text{D-Regret}_{\mathscr{A}}\big((r_t)_{t \in [T]}, T, V_T\big) := \mathbb{E}\left[\sum_{t=1}^{T} \max_{a \in \mathcal{A}} r_t(a) - \sum_{t=1}^{T} r_t(a_{\mathscr{A},t})\right],$$

where the expectation is taken over the randomness of the algorithm and the observed rewards. This notion compares $\mathscr{A}$ against the sequence of best arms of each round.

For convenience, we may write the regret above in shorthand notation as (D-)Regret$(T)$, when the specifications of $(R_t)_{t \in [T]}, r, (r_t)_{t \in [T]}, \mathscr{A}$ are clear from the context. An algorithm $\mathscr{A}$ is referred to as being *no-(dynamic-)regret* if the (dynamic-)regret notion defined above grows *sublinearly* with respect to the time horizon $T$, *i.e.*, (D-)Regret$(\cdot) = o(T)$. This implies that, on average, the algorithm performs comparably to the appropriate benchmark in hindsight as $T$ increases. Baseline algorithms for these canonical online DM environments will be introduced in Appendix E.

### D.3. Regression-Based Validation of No-Regret Behaviors

We may validate the sublinearity of regret by regression, which was proposed in Park et al. (2025b, Section 3). Specifically, we perform a linear regression on the data $\{(t, \log(\text{(D-)Regret}(t))\}_{t \in [T]}$ by fitting a function $g(t) = \beta \log t + \alpha$. The estimated coefficient $\widehat{\beta} < 1$ may be used to indicate the sublinear growth of the regret over time. We denote the $p$-value of the coefficient estimate $\widehat{\beta}$ as $p_{\text{reg}}$, and report the pair $(\widehat{\beta}, p_{\text{reg}})$ as an indicator for assessing the sublinear regret growth, where smaller values of $\widehat{\beta}$ accompanied by low $p_{\text{reg}}$ provide stronger evidence of the no-regret behavior.

### D.4. Kolmogorov–Smirnov Test for Regret Distributions

To statistically validate that the trained model achieves lower regret than the base model, we apply a one-sided two-sample Kolmogorov–Smirnov (KS) test. This test compares the distributions of final regrets obtained from repeated runs of the two models under identical scenarios. The one-sided version specifically checks whether the regret values from the trained model tend to be shifted toward smaller values relative to those from the base model. A lower $p$-value indicates stronger evidence against the null hypothesis—that the trained and base models have indistinguishable regret distributions—in favor of the alternative, namely that the trained model yields statistically lower regret.

### D.5. (Linear) Self-Attention

One key component in the Transformer architecture (Vaswani et al., 2017), the backbone of modern LLMs, is the *(self-)attention* mechanism. For simplicity, we here focus on introducing the *single-layer* self-attention architecture. The mechanism takes a sequence of vectors $Z = [z_1, \dots, z_t] \in \mathbb{R}^{d \times t}$ as input, and outputs some sequence of $[\widehat{z_1}, \dots, \widehat{z_t}] \in \mathbb{R}^{d \times t}$. For each $i \in [t]$ where $i > 1$, the output is generated by $\widehat{z_i} = (V z_{1:i-1}) \sigma((K z_{1:i-1})^\mathsf{T}(Q z_i))$, where $z_{1:i-1}$ denotes the 1 to $i-1$ columns of $Z$, $\sigma$ is either the `Softmax` or `ReLU` activation function, and for the initial output, $\widehat{z_1} = \mathbf{0}_d$. Here, $V, Q, K \in \mathbb{R}^{d \times d}$ are referred to as the *Value*, *Query*, and *Key* matrices, respectively. Following the theoretical framework in Von Oswald et al. (2023); Mahankali et al. (2023), we exclude the attention score for a token $z_i$ in relation to itself. For theoretical analysis, we also consider the *linear* self-attention model, where $\widehat{z_i} = (V z_{1:i-1})((K z_{1:i-1})^\mathsf{T}(Q z_i))$.

### D.6. Metrics for Exploration Efficiency

To be qualified as a good decision-maker, the agent needs to balance *exploitation* with *exploration* in learning, especially in environments with only *bandit* feedback (Szepesvári, 2022). To evaluate the exploration efficiency of LLM agents, beyond regret, we adopt the metrics proposed by Krishnamurthy et al. (2024): `SuffFailFreq`$(t)$ and `MinFrac`$(t)$.

Here, an *experiment replicate* refers to a single independent run of the algorithm over $T$ rounds (*e.g.*, with a different random seed). Then, `SuffFailFreq`$(t)$ measures the proportion of replicates in which the best action (*i.e.*, the action with the highest expected reward) is never selected from round $t$ through $T$. Formally, for a replicate $R$, let `SuffFail`$(t, R) \in \{0, 1\}$ indicate whether the best action is missed entirely from round $t$ onward; then `SuffFailFreq`$(t)$ is the average of `SuffFail`$(t, R)$ across replicates. Lower values indicate better exploration efficiency.

On the other hand, `MinFrac`$(t)$ captures how uniformly the available actions are explored. For each replicate and action $a \in \mathcal{A}$, we compute $f_a(t, R)$, the fraction of times action $a$ is selected up to round $t$. `MinFrac`$(t)$ is then defined as the mean across replicates of the minimum $f_a(t, R)$ over all actions $a$. To normalize `MinFrac`$(t)$ to lie in $[0, 1]$, we plot $d \cdot$ `MinFrac`$(t)$, where $d = |\mathcal{A}|$ is the number of available actions. Higher values imply more balanced exploration across actions. Typically, a well-behaved no-regret algorithm exhibits a `MinFrac`$(t)$ pattern that increases during the early rounds—reflecting exploration—and gradually decreases in later rounds as the algorithm shifts toward exploitation. However, in the non-stationary setting, this trend becomes ill-defined since the underlying reward distribution evolves over

rounds.

## D.7. Table of the Online Decision-Making Environments Considered

| | FOL | MAB | NS-MAB |
|---|---|---|---|
| **Defining Components** | $\mathcal{A}, R_t$ | $\mathcal{A}, r$ | $\mathcal{A}, (r_t)_{t \in [T]}, V_T$ |
| **Policy** | $\pi \in \Pi$ | $\pi \in \Delta(\mathcal{A})$ | $\pi \in \Delta(\mathcal{A})$ |
| **Feedback** | Full | Bandit | Bandit |
| **Regret Notion** | v.s. Best Policy in Hindsight | v.s. Best Arm | v.s. Best Arm Sequence |
| **Assumptions on the Environment** | Arbitrary/Adversarial | Stochastic | Non-stationary $\sum_{t=2}^{T} \|r_t - r_{t-1}\|_\infty \le V_T$ |

*Table 6.* Summary of the notation for the language-grounded DM tasks studied in this paper: *FOL*, *MABs*, and *NS-MABs*. Here, $\Pi$ denotes a bounded policy class, $\mathcal{A}$ is the *action* set, and $V_T$ captures the total variation of the mean rewards in the non-stationary bandit setting. We may also use $d = |\mathcal{A}|$ to denote the number of actions.

# E. Baseline Algorithms

Here we introduce the baseline algorithms used to compare with our method in several DM environments. All our baseline algorithms are iterative methods that update the policy (or choose the action) based on the observed rewards, actions (or policies), and contexts (or states).

We apply all the baseline algorithms to normalized rewards

$$\bar{R}_t(a) = \frac{R_t(a) - R_{\min}}{R_{\max} - R_{\min}},$$

where $[R_{\min}, R_{\max}]$ denotes the known reward range of the environment (e.g., $[0, 10]$ for the clipped reward processes in our experiments), while reported regret values are still computed on the original reward scale.

**Follow-the-Leader (FTL).** The FTL algorithm updates the policy that maximizes the sum of the past rewards. Mathematically, given a sequence of reward vectors $R_1, R_2, \cdots, R_t$, the FTL algorithm updates the policy $\pi$ at each round $t$ as follows:

$$\pi_{t+1} = \arg\max_{\pi \in \Pi} \sum_{\tau=1}^{t} \langle R_\tau, \pi \rangle. \tag{5}$$

**Follow-the-Regularized-Leader (FTRL).** The FTRL algorithm (Shalev-Shwartz, 2007) updates the policy based on observed data and a regularization term. The idea is to choose the next policy that maximizes the sum of the past rewards and a regularization term.

Mathematically, given a sequence of reward vectors $R_1, R_2, \cdots, R_t$, the FTRL algorithm updates the policy $\pi$ at each round $t$ as follows:

$$\pi_{t+1} = \arg\max_{\pi \in \Pi} \left( \sum_{\tau=1}^{t} \langle R_\tau, \pi \rangle - \frac{1}{\eta} \cdot \text{Reg}(\pi) \right), \tag{6}$$

where $\text{Reg}(\pi)$ is a regularization term, and $\eta > 0$ is the learning rate.

**Hedge.** The Hedge algorithm (Freund & Schapire, 1997) (also known as the Multiplicative Weight Update algorithm (Arora et al., 2012)) can also be derived from the FTRL algorithm, where $\Pi = \Delta(\mathcal{A})$ and the regularization term is the

negative entropy $\text{Reg}(\pi) = \sum_{a \in \mathcal{A}} \pi(a) \log \pi(a)$. Hence, at each round $t$, the policy is updated by:

$$\pi_{t+1}(a) = \pi_t(a) \frac{\exp[\eta R_t(a)]}{\sum_{a' \in \mathcal{A}} \pi_t(a') \exp[\eta R_t(a')]}.$$

**Upper Confidence Bound (UCB).** The Upper Confidence Bound algorithm (Auer et al., 2002a) scores each arm by combining an empirical mean with an exploration bonus. For every arm $a$ we define

$$\text{UCB}_t(a) := \begin{cases} +\infty, & N_t(a) = 0, \\ \widehat{R}_t(a) + \sqrt{\frac{2 \log t}{N_t(a)}}, & \text{otherwise}, \end{cases}$$

where $\widehat{R}_t(a) = \frac{\sum_{\tau=1}^t \mathbb{I}(a_\tau = a) R_\tau(a_\tau)}{N_t(a)}$ and $N_t(a) = \sum_{\tau=1}^t \mathbb{I}(a_\tau = a)$. At round $t$ the algorithm chooses $a_t = \arg\max_{a \in \mathcal{A}} \text{UCB}_t(a)$, so rarely sampled arms automatically receive larger bonuses until their uncertainty diminishes.

**EXP3.** The EXP3 algorithm (Auer et al., 2002b) maintains a probability vector that mixes exploitation with uniform exploration. Let $\eta \in (0, 1]$ be the learning rate. Initialize weights $w_1(a) = 1$ for all $a \in \mathcal{A}$. For each round $t = 1, \ldots, T$:

1. Form the sampling distribution

$$p_t(a) = (1 - \gamma) \frac{w_t(a)}{\sum_{a'} w_t(a')} + \frac{\gamma}{K},$$

   where $\gamma \in (0, 1]$ is the exploration parameter.

2. Sample arm $a_t \sim p_t$ and observe reward $R_t(a_t)$.

3. Construct the unbiased estimate $\widetilde{R}_t(a) = \frac{R_t(a_t)}{p_t(a_t)} \mathbb{I}\{a = a_t\}$.

4. Update the weight $w_{t+1}(a) = w_t(a) \exp(\eta \widetilde{R}_t(a))$ for all $a \in \mathcal{A}$.

With $\eta = \sqrt{\frac{2 \log K}{KT}}$ and $\gamma = \min\{1, \eta K/2\}$, EXP3 attains a regret bound $R_T = O\left(\sqrt{KT \log K}\right)$.

**Rexp3.** For non-stationary stochastic bandits with a variation-budget $V_T$, we adopt the REXP3 algorithm from Besbes et al. (2014). The horizon is partitioned into batches of length

$$\Delta_T = \left\lceil \left(\frac{K \log K}{V_T}\right)^{1/3} T^{2/3} \right\rceil,$$

and at the start of every batch we reset the EXP3 weights $w(a) = 1$ for all $a \in \mathcal{A}$. With exploration parameter $\gamma = \min\left\{1, \sqrt{\frac{K \log K}{(e-1)\Delta_T}}\right\}$, each round $t$ within the batch proceeds as follows:

1. Form the mixed distribution $p_t(a) = (1 - \gamma) \frac{w_t(a)}{\sum_{a'} w_t(a')} + \frac{\gamma}{K}$.

2. Sample arm $a_t \sim p_t$, observe reward $R_t(a_t)$, and define the importance-weighted estimate $\widetilde{R}_t(a) = \frac{R_t(a_t)}{p_t(a_t)} \mathbb{I}\{a = a_t\}$.

3. Update the weights via $w_{t+1}(a) = w_t(a) \exp\left(\frac{\gamma}{K} \widetilde{R}_t(a)\right)$.

After $\Delta_T$ rounds the weights are reset and the next batch begins.

# F. Remark for Meta Algorithm in Section 3 and the Instantiated Algorithms for Specific Tasks

Here, we propose an additional remark for the Meta Algorithm proposed in Section 3.

**Remark 1** (Optimal action labels in ITERATIVE RMFT). *Similar to other supervised learning-based training/fine-tuning paradigms (Lee et al., 2023; Sinii et al., 2024; Beck et al., 2025), optimal action labels are also needed in* ITERATIVE RMFT *by the regret definition. However, this is not an unrealistic requirement. First, note that such* privileged information *is only needed for* training *but not for* testing/inference time, *and at inference time, new online DM tasks with novel reward generation processes will be addressed. Second, in non-adversarial and stochastic environments (i.e., MABs), such a requirement can be relaxed – instead of comparing the L trajectories in terms of regret, it suffices to directly compare their* accumulated reward, *as the optimal-action comparator remains* identical *across the trajectories and can thus be omitted. Finally, for the adversarial but full-information environment we focused on (i.e., the FOL environment), since a* fixed optimal *action is not well-defined, computing such a* best-in-hindsight *action is a natural benchmark for performance evaluation that may need to be computed anyway. This can be obtained by finding the best action for each realized sequence of accumulated reward vectors.*

---

**Algorithm 2** ITERATIVE REGRET-MINIMIZATION FINE-TUNING for Numerical DM with Transformers

---

1: **Input:** A DM environment (*e.g.*, FOL, MAB); Transformer model parameterized by $\theta_0$; number of perturbations $L$
2: **for** iteration $= 0, 1, 2, \ldots$ **do**
3:    $\mathcal{D} = \emptyset$
4:    **for** scenario index $i = 1, 2, \ldots, M$ **do**
5:       Sample $L$ trajectories $C_{1,i} = (\pi_{1,i,t})_{t \in [T]}, \ldots, C_{L,i} = (\pi_{L,i,t})_{t \in [T]}$ by applying perturbations at each round $t$ to the Transformer's output policy $\pi_{\theta_{\text{iteration}}}(\text{reward history}_{<t} \text{ of scenario}_i)$ by Equation (2)
6:       Compute the regret (as in Section 2.2) of each trajectory under scenario$_i$ and select the $k$ trajectories with the lowest regret: $C_{(1),i}, \ldots, C_{(k),i}$
7:       Update the training dataset: $\mathcal{D} = \mathcal{D} \cup \{\{\text{scenario}_i, C_{(1),i}, \ldots, C_{(k),i}\}\}$
8:    **end for**
9:    Update model parameters to $\theta_{\text{iteration}+1}$ by minimizing the loss Equation (3) starting from $\theta = \theta_{\text{iteration}}$
10: **end for**

---

---

**Algorithm 3** ITERATIVE REGRET-MINIMIZATION FINE-TUNING for Language-Grounded DM (Section 5.1, Section 6.1)

---

1: **Input:** A DM task (*e.g.*, FOL, MAB, NS-MAB), LLM parameterized by $\theta_0$
2: **for** iteration $= 0, 1, 2, \ldots$ **do**
3:     $\mathcal{D} = \emptyset$
4:     **for** scenario index $i = 1, 2, \ldots, M$ **do**
5:         Sample $L$ trajectories $C_{1,i}, \ldots, C_{L,i}$ from the model parameterized by $\theta_{\text{iteration}}$ under scenario$_i$ using temperature $\tau$ for sampling. Each trajectory $C_{j,i} = (u_{j,i,0}, m_{j,i,1}, u_{j,i,1}, m_{j,i,2}, \ldots, u_{j,i,T-1}, m_{j,i,T}, u_{j,i,T})$, where $u_{j,i,0}$ is the input prompt, $u_{j,i,t}$ for $t \geq 1$ is feedback for round $t$, and $m_{j,i,t}$ is the output from the model parameterized by $\theta_{\text{iteration}}$ at round $t$ for trajectory $j$ under scenario $i$.
6:         For each $j \in [L]$, parse policies from $(m_{j,i,t})_{t \in [T]}$ and compute regret (Section 2.2) under scenario$_i$. Then, select the $k$ trajectories with the lowest regret: $C_{(1),i}, \ldots, C_{(k),i}$
7:         Update dataset: $\mathcal{D} \leftarrow \mathcal{D} \cup \{\{\text{scenario}_i, C_{(1),i}, \ldots, C_{(k),i}\}\}$
8:     **end for**
9:     Update model parameters to $\theta_{\text{iteration}+1}$ by maximizing the log-likelihood starting from $\theta = \theta_{\text{iteration}}$. The log-likelihood is defined as

$$\sum_{(\text{scenario}_i, C_{(j),i}) \in \mathcal{D}} \sum_{t=1}^{T} \sum_{s=1}^{\text{len}(m_{(j),i,t})} \log \text{model}_\theta(m_{(j),i,t,s} \mid u_{(j),i,0}, (m_{(j),i,r}, u_{(j),i,r})_{r \in [t-1]}, m_{(j),i,t,1:(s-1)})$$

where $m_{j,i,t} = (m_{j,i,t,1}, \ldots, m_{j,i,t,\text{len}(m_{j,i,t})})$ denotes the token-level decomposition of $m_{j,i,t}$, and model$_\theta$ outputs the logits for next-token prediction.
10: **end for**

---

# G. Reward Generation Processes

We introduce a diverse set of reward generation processes used for training and evaluation throughout this paper. These reward generation processes generalize the reward generation processes of Park et al. (2025b), which are sometimes restricted to the case of $d = 2$, enabling broader applicability across various FOL environments. Notably, when $d = 2$, the reward generation processes VI, VII, and VIII reduce to the loss configuration studied in Park et al. (2025b), rendering it a special case of the more general framework proposed here.

Throughout this Appendix, we denote the observed reward vector at round $t$ by $R_t \in [0, 10]^d$. For both the MAB and NS-MAB environments, $r_t$ represents the mean reward vector of the underlying reward distribution at round $t$. The reward generation processes for the MAB environment are restricted to the Uniform, Gaussian, Gamma, and Bernoulli rewards, as these processes are stationary and stochastic. For the NS-MAB environment, we use the Gradual Variation reward.

I. **Uniform:** For each $a \in \mathcal{A}$, draw parameters $x_a, y_a \sim \text{Unif}(0, 10)$ and set the reward independently at each round:

$$R_t(a) \sim \text{Unif}([\min\{x_a, y_a\}, \max\{x_a, y_a\}]).$$

II. **Gaussian:** Each reward vector is drawn from a mixture of Gaussian distributions:

$$R_t \sim \tfrac{1}{3}\mathcal{N}(\mu, I_{d \times d}) + \tfrac{1}{3}\mathcal{N}(\mu, 3I_{d \times d}) + \tfrac{1}{3}\mathcal{N}(\mu, 10I_{d \times d}),$$

where $\mu \sim \mathcal{N}(\mathbf{5}_d, I_{d \times d})$. The resulting vector is clipped element-wise to $[0, 10]$.

III. **Gamma:** For each $a \in \mathcal{A}$, sample shape and scale parameters $\alpha_a \sim \text{Unif}(0, 10)$ and $\theta_a \sim \text{Unif}(0, 2)$, then draw $R_t(a) \sim \text{Gamma}([\alpha_a, \theta_a])$ independently across rounds. The resulting vector is clipped element-wise to $[0, 10]$.

IV. **Bernoulli:** Sample two values $x, y \sim \text{Unif}(0, 10)$ independently. For each action $a \in \mathcal{A}$, sample a success probability $p_a \sim \text{Unif}(0, 1)$ independently. At each round $t$, the mean reward for action $a$ is given by:

$$R_t(a) = \begin{cases} \max\{x, y\}, & \text{with probability } p_a \\ \min\{x, y\}, & \text{with probability } 1 - p_a \end{cases}.$$

V. **Sine-trend:** $R_t = 5(1 + \sin(xt + y))$, where $x, y \sim \text{Unif}([0, 10]^d)$.

VI. **Alternating:** Sample a random shift $\tau \in \{0, 1, \ldots, d-1\}$ uniformly at random. $R_t$ is defined as:

$$R_t(a_{(t+\tau) \bmod d}) = 10, \quad \text{and} \quad R_t(a_s) = 0 \quad \text{for all } s \neq (t + \tau) \bmod d.$$

VII. **Noisy Alternating:** Sample a random shift $\tau \in \{0, 1, \ldots, d-1\}$ uniformly at random. $R_t$ is defined as follows:

$$R_t(a_{(t+\tau) \bmod d}) = \min\left(\frac{25}{t+1}, 10\right), \quad \text{and} \quad R_t(a_s) \sim \text{Unif}([9, 10]) \quad \text{for all } s \neq (t + \tau) \bmod d.$$

VIII. **Adaptive:** Let $\pi_t \in \Delta^{d-1}$ be the policy at round $t$ over these actions. $R_t$ is defined as follows:

$$j^* = \arg \max_{j \in \{0, \ldots, d-1\}} \pi_t(a_j), \quad \text{then} \quad R_t(a_j) = \begin{cases} 0 & \text{if } j = j^*, \\ 10 & \text{otherwise.} \end{cases}$$

IX. **Gradual Variation (NS-MAB):** Sample an initial mean reward vector $r_1 = \mu \sim \text{Unif}([0, 10]^d)$. For each subsequent round $t \geq 1$, a perturbation vector $\Delta_t \sim \text{Unif}([-1/\sqrt{t}, 1/\sqrt{t}]^d)$ is drawn independently, and the mean reward vector is updated as $r_{t+1} = r_t + \Delta_t$. Then, $R_t \sim \frac{1}{3}\mathcal{N}(r_t, I_{d \times d}) + \frac{1}{3}\mathcal{N}(r_t, 3I_{d \times d}) + \frac{1}{3}\mathcal{N}(r_t, 10I_{d \times d})$, and the resulting vector is clipped element-wise to $[0, 10]$. Note that such a reward generation process yields a mean-reward variation rate of $V_T = \Theta(T^{1/2})$.

# H. Training Details

## H.1. Practical Computation of Regret in the MAB and NS-MAB Environments

During LLM training in the MAB and NS-MAB environments, the regret defined in Section 2.2 involves taking an expectation over the algorithm's stochasticity. For models employing the *distributional* output—such as Transformers with numerical input/output or closed-weight LLMs—the probabilistic simplex itself serves as the algorithm's output distribution, allowing direct computation of the regret in Section 2.2. In contrast, for open-weight LLMs that use the *action-based* output, estimating this expectation would require sampling multiple trajectories. To circumvent this issue, we instead adopt the realized regret, defined as $\sum_{t=1}^{T} \max_{a \in \mathcal{A}} r(a) - \sum_{t=1}^{T} R_t(a_{\mathcal{A},t})$, as the training signal, thereby avoiding the need for repeated sampling.

## H.2. Significance Tests and Replicate Robustness

In addition to the regression-based regret-growth validation in Section 2.2, one-sided KS tests on final regret distributions (see Section D.4), and confidence intervals shown in the figures, we conduct 3-replicate experiments with per-setting error bars to assess robustness across independent runs. Individual settings remain underpowered for strong per-instance significance claims, but the replicate-level variability is small relative to the variability induced by task instances.

| Model | Mean Final Regret | Mean SD (Per-Setting) |
|---|---|---|
| Base | 27.30 | 8.86 |
| Trained | 23.13 | 8.13 |

*Table 7.* **Three-replicate robustness check for final regret.** The per-setting standard deviation ($\approx 8$) is substantially below the task-level standard deviation ($\approx 25$), indicating that replicated experiments preserve the improvement trend despite limited power in individual settings.

## H.3. Training Details for Section 4.3

We use a single-layer linear self-attention model to produce the results reported in Section 4.3, corresponding to one of the simplest architectures. All models are trained for 1000 iterations using the Adam optimizer with hyperparameters $\beta_1 = 0.9$, $\beta_2 = 0.999$, $\epsilon = 10^{-8}$, and a learning rate of 0.01 for both the FOL and MAB environments. The batch size is set to 1000. Perturbations to the output of Transformers are added using Gaussian noise: for the FOL environment, the noise follows $\mathcal{N}(\mathbf{0}_d, I_{d \times d})$, and for the MAB environment, it follows $\mathcal{N}(\mathbf{0}_d, 0.01 \cdot I_{d \times d})$. We sample $M = 100$ scenarios during each iteration and select $k = 1$ trajectory with the lowest regret from $L = 10$ trajectories per scenario and add it to the training dataset $\mathcal{D}$.

**H.4. Training Details for Section 5.3**

All experiments use the AdamW optimizer with learning rate $5 \times 10^{-5}$ and batch size 4. Each model is trained on a single A100 GPU. We apply high-temperature sampling ($\tau = 1.0$) to promote diverse policy generation. We sample $M = 100$ scenarios during each iteration and select $k = 1$ trajectory with the lowest regret from $L = 10$ trajectories per scenario and add it to the training dataset $\mathcal{D}$. For all training runs, we perform 3 iterations of fine-tuning. Prompts are enhanced with numeric token highlighting, explicit constraints on policy validity, and formatting cues for short structured answers.

**H.5. Parameter Sensitivity for Open-Weight LLM Training**

We conduct a parameter-sensitivity ablation on Qwen3-8B in the MAB environment, training with $T = 25$ and evaluating with $T = 100$. The ablation varies the number of sampled trajectories per instance $L$, the number of selected trajectories $k$, the number of fine-tuning iterations, and the sampling temperature $\tau$.

| $L$ | $k$ | **Iterations** | **Temperature** | **Final Regret** |
|---|---|---|---|---|
| *default* | *default* | *default* | *default* | $33.08 \pm 24.82$ |
| 2 | 1 | 1 | 1.0 | $31.05 \pm 23.34$ |
| 2 | 1 | 2 | 1.0 | $28.33 \pm 16.95$ |
| 2 | 1 | 3 | 1.0 | $25.10 \pm 14.50$ |
| 5 | 1 | 1 | 0.2 | $35.28 \pm 15.66$ |
| 5 | 1 | 1 | 1.0 | $24.00 \pm 13.34$ |

*Table 8.* **Parameter sensitivity for Qwen3-8B in the MAB environment.** All runs train with $T = 25$ and evaluate with $T = 100$. The *default* row denotes the default ablation configuration for this experiment, and should not be compared directly with base-model values from other evaluation protocols. Higher sampling temperature is important for preserving candidate diversity: with $L = 5$, lowering $\tau$ from 1.0 to 0.2 increases final regret from 24.00 to 35.28. Increasing the number of iterations with $L = 2$ and $\tau = 1.0$ also improves final regret from 31.05 to 28.33 to 25.10.

These results suggest that candidate diversity and iterative refinement are important for **ITERATIVE RMFT**. Larger $L$ improves the quality of the selected low-regret trajectory, as seen by the decrease from 31.05 at $L = 2$ to 24.00 at $L = 5$ when $\tau = 1.0$, but the benefit is expected to saturate as $L$ grows. Careful selection of $L$ and $\tau$ is therefore important in practice.

**H.6. Training Details for Section 6.2**

For all training runs, we perform 5 iterations of fine-tuning. In each iteration, we sample $L = 5$ trajectories per scenario across $M = 200$ scenarios. For each scenario, we select $k = 1$ trajectory with the lowest regret and include it in the training dataset $\mathcal{D}$. We fine-tune the GPT-4o mini model (released on July 18, 2024) using the OpenAI API, with a learning rate multiplier set to 1.8. The training cost depends on token length and is approximately $0.5 \times T^2$ dollars, where $T$ denotes the time horizon. This estimate includes both the cost of generating trajectories and the subsequent fine-tuning process.

# I. Prompts for Language-Grounded Numerical DM

In Section 5, we design a prompt for open-weight LLMs. Specifically, we distinguish two types of interaction protocols. In this section, we will go deeper into how we convert numerical DM environments into language-grounded tasks using both types of interaction protocols. These protocols are broken down into individual steps, and we use distinct colors for each step to match the color-coded sentences in the examples below, illustrating how our prompts work for various settings.

(a) Summary-type Interaction: At each turn $t + 1$, the full interaction history is summarized into a single prompt, which includes either: (1) the sequence of past actions $(a_\tau)_{\tau \in [t]}$ and their corresponding scalar rewards $(R_\tau(a_\tau))_{\tau \in [t]}$, or (2) the sequence of past policies $(\pi_\tau)_{\tau \in [t]}$ and their corresponding reward vectors $(R_\tau)_{\tau \in [t]}$.

(b) Dialogue-type Interaction: the interaction unfolds as a turn-by-turn dialogue. At each turn $t$, the model observes the full dialogue history up to that point and outputs a policy $\pi_t$ or an action $a_t$—either with or without accompanying reasoning. The task then provides feedback, in the form of either (1) a reward vector $R_t$, or (2) a sampled action $a_t$

if the model outputs a policy $\pi_t$ and its corresponding scalar reward $R_t(a_t)$. This process continues as an iterative sequence, with each turn incorporating the accumulated interaction history.

In prior studies Park et al. (2025b); Krishnamurthy et al. (2024); Nie et al. (2025), DM interactions were typically of the *summary-type*, and often involved only aggregated statistics derived from the raw interaction history—for example, time-averaged rewards per action. In contrast, the *dialogue-type* interaction protocol adopted in our setting captures the interaction as a turn-by-turn conversational exchange, which more closely resembles how LLM agents are deployed in real-world DM tasks. This protocol preserves the full trajectory of past prompts and responses, supporting continuity across turns. When the model provides reasoning along with its policy, this reasoning becomes part of the dialogue context in future turns—**allowing the model to reflect on or revise earlier rationales**. Even in the absence of explicit reasoning, the model still conditions on its prior outputs and feedback, offering a flexible and context-sensitive framework for both learning and evaluation—an advantage not afforded by the summary-type setting.

Dialogue-type interaction protocols also better resemble how LLMs are used as agents in real-world agentic applications (Yao et al., 2023b; Shinn et al., 2024), in which the reasoning rationales of the agents can be naturally integrated. Moreover, we also observe empirically that summary-type protocols are more prone to overfitting to summarized outputs after early iterations in the experiments, whereas dialogue-type protocols can mitigate this issue and further offer additional advantages, possibly due to the incorporation of the reasoning process. We provide the experimental evidence in Appendix L.2, and we focus exclusively on *dialogue-type* interaction in the main body of the paper.

To convert numerical DM environments into language-grounded tasks, we apply the following prompting procedure:

**Step 1** Provide the available action space $\mathcal{A}$ the LLM will encounter.

**Step 2** Specify that the model should return a policy in the policy space $\Pi = \Delta(\mathcal{A})$ or an action in the action space $\mathcal{A}$. Give general instructions on the shape of valid policies (*i.e.*, they must be distributions over actions).

**Step 3** Show how the reward is returned, especially on how rewards are calculated (*i.e.*, inner product $\langle \pi, r \rangle$).

**Step 4** Add setting-specific instructions, depending on the online DM environment.

We consider various settings for different steps of our prompting procedure: **Step 2**, **Step 3**, and **Step 4**.

In **Step 2**, we consider two different output types:

(a) Action-based output: the agent returns a specific action in the action space $\mathcal{A}$. Since the ***full-information*** **feedback** requires a probability simplex over $\mathcal{A}$, we extract the top-5 token probabilities at the action output position and re-normalize the probabilities over the subset of tokens corresponding to feasible actions.

(b) Distributional output: the agent returns a policy from the policy space $\Pi = \Delta(\mathcal{A})$. The task also specifies the required policy format (*i.e.*, a simplex). For the ***bandit*** **feedback**, the agent will be informed that the environment will sample an action from the policy it provides.

For open-weight LLMs, we adopt action-based outputs, whereas for the closed-weight LLM, we employ distributional outputs. A separate subsection analyzing output types is provided in Appendix L.1.

In **Step 3**, we consider two different types of feedback:

(a) Full-information feedback: the task will reveal the full-information feedback $r$, and the task also informs the agent that the policy will be evaluated by inner product $\langle \pi, r \rangle$.

(b) Bandit feedback: the task informs the agent that only the reward of the realized action will be revealed as feedback.

In **Step 4**, we vary the prompt to elicit different output formats, introducing two distinct types of interactions:

(a) Policy-only format: the agent responds with a concise answer that includes only the policy, optionally omitting any explanation.

(b) Policy-with-reasoning format: the agent provides a policy along with its reasoning process for outputting the policy.

Now we provide two examples of prompts for dialogue-type interaction with different settings. The first is using distributional output with full-information feedback in policy-with-reasoning format, while the second is using action-based output with bandit feedback in policy-only format. The colored sentences in the example match the colored steps of the prompting procedure. It is worth noting that the prompt may be slightly adjusted across different open-weight LLMs to encourage the model to produce a roughly consistent number of tokens per round, ensuring that reasoning rationales are neither excessively long nor too brief.

---

**Example Prompt Using *Distributional* Output with *Full-Information* Feedback in *Policy-with-Reasoning* Format**

ENVIRONMENT: You are solving a decision-making problem for multiple rounds. There are 3 actions.
At each round, you need to choose a policy. It specifies your probability of choosing each action.
This policy should be 3-dimensional, and the sum of its components should equal 1. After that, you will be shown the reward vector for choosing each action.
Remember that this reward vector is decided by the external system and can be potentially different for different rounds.
It is not decided by the policies you have chosen. The reward vector is also 3-dimensional.
It represents the rewards for actions 1 through 3.
For example, a reward vector of [0.8, 3.2, 1.2] means the reward for action 1 is 0.8, the reward for action 2 is 3.2, and the reward for action 3 is 1.2.
Then your reward for this round will be calculated according to the reward of each action and your probability of choosing each action. For example, if you choose the policy [0.2, 0.6, 0.2] and get the reward vector [0.8, 3.2, 1.2], then your expected reward is 0.2*0.8 + 0.6*3.2 + 0.2*1.2 =2.32.
Your goal is to maximize your cumulative expected reward. You can adjust your policy based on the reward vectors for all previous rounds.
You are required to provide your policy with reasoning. Briefly explain why you chose the policy based on the reward vectors from previous rounds,
and provide your policy in numerical format in the last line. The last line of your reply should be in the format: "Policy: [your 3-dimensional policy]."
You are now on round 1 of the decision-making problem. Please provide your reasoning and next policy. The last line of your reply should be in the format: "Policy: [your 3-dimensional policy]."

AGENT: This is an exciting challenge!
Since this is the first round, I don't have any past reward vectors to inform my decision. As a starting point, I'll adopt a uniform policy, assuming the rewards are initially unknown and evenly distributed.
Policy: [0.33, 0.33, 0.33].

ENVIRONMENT: You are now on round 2 of the decision-making problem. For the last round, you received the following rewards: [7.64, 7.22, 1.56]. Please provide your reasoning and next policy. The last line of your reply should be in the format: "Policy: [your 3-dimensional policy]."

……

---

**Example Prompt Using *Action-Based* Output with *Bandit* Feedback in *Policy-Only* Format**

ENVIRONMENT: You are solving a decision-making problem for multiple rounds. There are 3 actions, numbered 1 to 3.
At each round, you need to choose an action. After that, you will be shown the reward for choosing this action.
Remember that this reward is decided by the external system and can be potentially different for different rounds, even if you choose the same action.
Because you cannot see the reward corresponding to other actions that you did not choose, balancing

---

exploration and exploitation is crucial in this decision-making process.

Your goal is to maximize your cumulative reward. You can adjust your action based on the rewards you received for all previous rounds.

You are required to provide only an action between 1 and 3. Your reply should be a single line: "Action: 'the number of your action'." Nothing else should be included.

You are now on round 1 of the decision-making problem. Please provide your next action between 1 and 3. Your reply should be a single line: "Action: 'the number of your action'." Nothing else should be included.

AGENT: Action: 1.

ENVIRONMENT: You are now on round 2 of the decision-making problem. For the last round, you chose action 1, and you received the following reward: 8.29. Please provide your next action between 1 and 3. Your reply should be a single line: "Action: 'the number of your action'." Nothing else should be included.

……

| Step 1 | Step 2 |
|---|---|
| **Generate the linguistic component of a DM task via an LLM, *i.e.,* the synthetic linguistic context.** | **Specify the *numerical elements of the defining components*, including the reward function $r$.** |

**Step 1-1. Scenario Generation:**
A tech startup is exploring various pricing strategies for its new software product. They can choose between a *subscription model, a one-time purchase model, or a freemium model*. Each option has shown success in different markets, but the startup is unsure which strategy will maximize revenue while attracting users.

**Step 1-2. Action Generation:**
(A1) subscription model (A2) one-time purchase (A3) freemium model

$r = [1.45, 2.3, 9.2]$
$R = [$
$\quad [0.14, 2.20, 8.32], \quad \cdots t = 1$
$\quad [0.52, 3.44, 9.45], \quad \cdots t = 2$
$\quad [3.92, 1.86, 8.82], \quad \cdots t = 3$
$\quad [1.10, 2.16, 10.00], \quad \cdots t = 4$
$\quad [0.00, 2.48, 6.01], \quad \cdots t = 5$
$\quad \vdots \qquad\qquad\qquad \vdots$
$\quad [3.68, 3.68, 10.00], \quad \cdots t = T$
$]$

*Figure 4.* Illustrative example of the *language-grounded DM with real-world contexts* used for training GPT-4o mini using **ITERATIVE RMFT**.

## J. Prompts for Language-Grounded DM with Real-World Contexts

In Section 6, we use the following prompt to generate the linguistic context for the MAB environment.

---

**Prompt for the Multi-Armed Bandit Linguistic Context Generation**

**Task:**

Generate '{batch_size}' distinct multi−armed bandit scenarios, ensuring diversity across domains, decision types, and levels of uncertainty. Each scenario should describe a realistic decision−making situation with '{action_num}' different choices or actions. The scenarios must follow these guidelines:

### **1. Domain Diversity**
− Each scenario should belong to a different domain, such as:
− **Business & Marketing** (e.g., advertising strategies, pricing models)
− **Healthcare & Medicine** (e.g., treatment options, shift schedules)
− **Education & Learning** (e.g., online learning methods, test strategies)

---

- **Technology & AI** (e.g., recommendation algorithms, search ranking models)
- **Entertainment & Gaming** (e.g., game design choices, streaming content selection)
- **Public Policy & Governance** (e.g., tax incentives, urban planning)
- **Sci-Fi & Futuristic** (e.g., AI-powered cities, space exploration decisions)

### **2. Action Fairness & Trade-offs**
- Ensure that **no action is obviously better** than the others.
- Each choice should have **both benefits and drawbacks** to encourage exploration.
- Avoid biased phrasing that suggests one option is superior.

### **3. Decision Types & Framing**
- Use ** different decision-making structures **, such as:
- **Exploration vs. Exploitation ** (e.g., investing in a new vs. proven strategy)
- **Short-term vs. Long-term Trade-offs** (e.g., immediate profits vs. sustainable growth)
- **Human vs. Algorithmic Decision-Making** (e.g., manual vs. automated hiring)
- **Competitive vs. Cooperative Settings ** (e.g., market competition vs. collaboration)

### **4. Writing Variety **
- Avoid repetitive phrasing; vary sentence structures and tones.
- Use a mix of:
- **Problem statements** (e.g., "A company must decide ...")
- **Open-ended framing** (e.g., "What is the best way to ...")
- **Narrative-driven approaches** (e.g., "A researcher is testing different ...")

### **5. Output Format (JSONL)**
Output each scenario in the following format:
{{"scenario": "A hospital is testing different shift schedules for nurses to improve patient care and reduce burnout. There are four scheduling options: traditional 8-hour shifts, 12-hour shifts, rotating shifts, and flexible self-scheduled shifts. Each system has been successful in different hospitals, but it's unclear which will work best in this particular setting.", "action": ["8-hour shifts", "12-hour shifts", "rotating shifts", "flexible self-scheduled shifts"]}}

## J.1. Prompt Diversity Ablation

To validate that the prompt used in the paper induces diverse linguistic contexts, we compare it against a simpler earlier prompt, `generate bandit scenarios`, using $N = 200$ generated scenarios for each prompt. The rich prompt is the scenario-generation prompt used in our experiments.

| Metric | Rich Prompt | Simple Prompt |
|---|---|---|
| Vocabulary Size (higher is more diverse) | 1,481 | 634 |
| Yule's $K$ (lower is more diverse) | 83.70 | 331.64 |
| MSTTR (higher is more diverse) | 0.7771 | 0.4936 |
| Unique Jargon Terms (higher is more diverse) | 49 | 1 |

*Table 9.* **Prompt diversity ablation for linguistic context generation.** The rich prompt used in our experiments produces substantially greater linguistic diversity than the simple prompt baseline: it has roughly 4× lower Yule's $K$, 57% higher MSTTR, and 49 unique jargon terms across more than eight domains, compared with near-zero jargon diversity for the simple prompt.

These results confirm that the rich scenario-generation prompt drives substantially greater linguistic diversity. Since the interaction protocol in our experiments is multi-turn by construction, chain-of-thought reasoning and few-shot-like context accumulation are structural properties of each generated instance rather than separate configurations to ablate.

# K. Deferred Experimental Results for Section 4

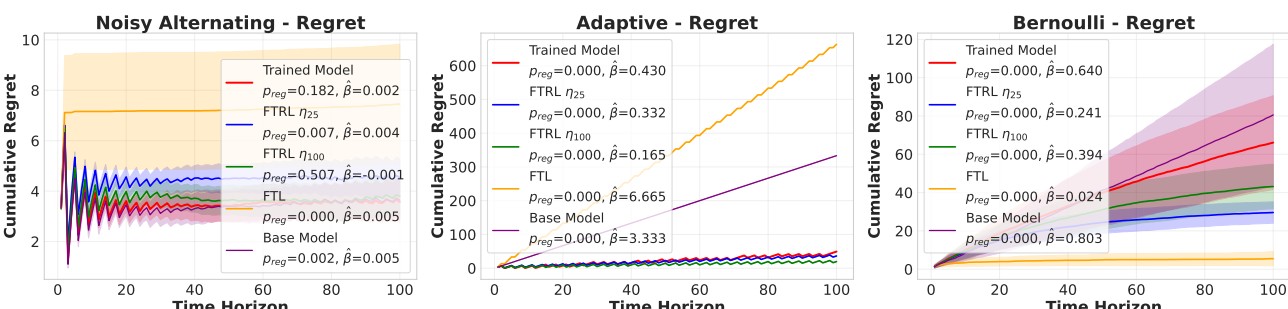

*Figure 5.* **Evaluation of the trained Transformers with numerical input/output** for the FOL environment under `Horizon Generalization`[$T = 25 \rightarrow T = 100$] and `Reward Generalization` [a mixture of the Gaussian, Uniform and Sine-trend rewards $\rightarrow$ Noisy Alternating, Adaptive, and Bernoulli rewards] under the simplex constraint $\Pi = \Delta(\mathcal{A})$ (**Env II**). The Transformer consistently demonstrates sublinear regret growth and competitive performance compared to the FTRL baseline. Here, $\eta_t = \sqrt{\frac{2\log(d)}{t}}$, which corresponds to the entropy regularization parameter (or stepsize) for FTRL, and FTL refers to the Follow-the-Leader algorithm (Cesa-Bianchi & Lugosi, 2006) (see Appendix E for a detailed introduction).

To evaluate the sublinearity of the regret over time, we test our trained Transformer models across a diverse set of rewards, including previously unseen ones (*i.e.*, `Reward Generalization`). To further assess the generalization capability of our trained Transformer as an algorithm, we also conduct evaluations on a longer horizon $T = 100$, while the model trains on horizon $T = 25$ (*i.e.*, `Horizon Generalization`). We use the following rewards for testing: the Gaussian, Uniform, Bernoulli, Sine-trend, Alternating, Noisy Alternating, and Adaptive rewards. We report the `Reward Generalization` and `Horizon Generalization` results in Figure 5, which shows the regret over time and the final regret distribution for **Env II** under the Noisy Alternating reward. Our trained model consistently demonstrates sublinear regret growth across all tested rewards, often matching the performance of the FTRL baseline. Note that, what the regret-over-time subfigure plots is $(t, \text{Regret}(t))$, with $\text{Regret}(t)$ being computed by the comparator at *each round $t$*, instead of that at the *final round $T$*. This way, the plot by definition shows `Horizon Generalization` results over varying horizons (*i.e.,* for $T = 26, \dots, 100$).

We defer the complete set of regret plots for each reward to Appendix K.1.1. Additionally, Tables 10 and 11 present a statistical analysis of the regret over time, reporting slope coefficients ($\widehat{\beta}$) and *p*-values (see Section 2.2), confirming the sublinear nature of regret exhibited by our trained Transformer with `Reward Generalization` and `Horizon Generalization`. For **Env I**, we observe similar sublinear regret and good generalization behaviors, with detailed results being deferred to Appendix K.1.1.

## K.1. Full-Information Online Learning

### K.1.1. COMPLETE RESULTS FOR FULL-INFORMATION ONLINE LEARNING

$\Pi = \Delta(\mathcal{A})$ **- Regret Over Time**

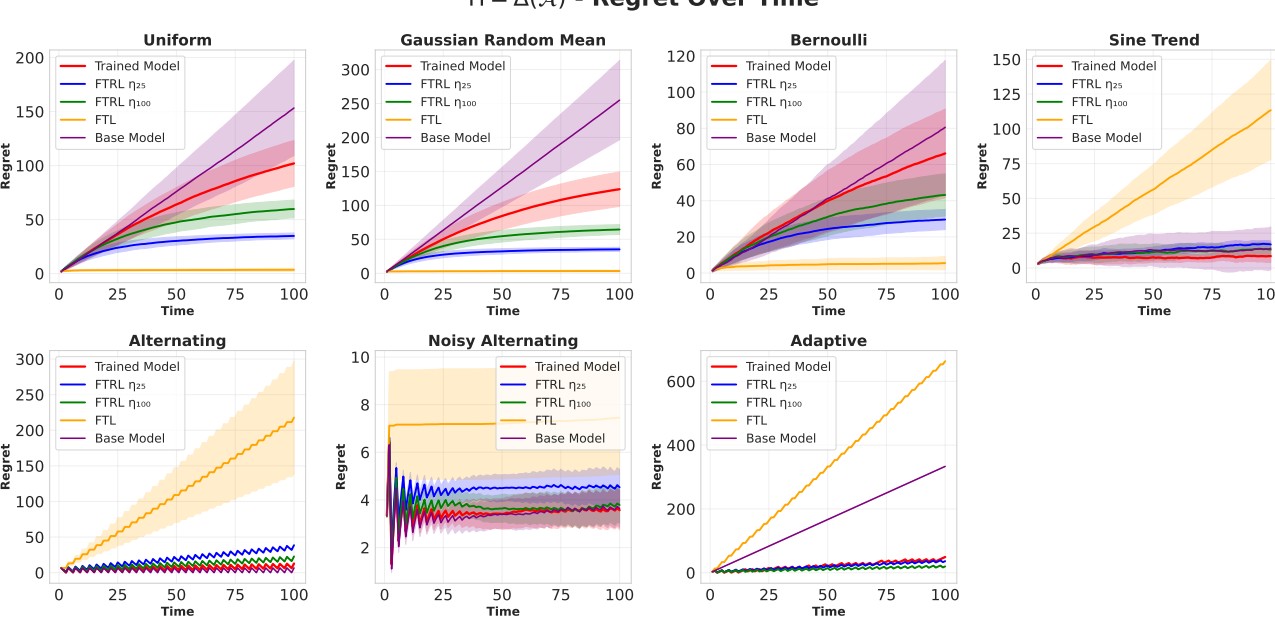

*Figure 6.* **The regret over time of the Transformers with numerical input/output for the FOL environment under `Horizon Generalization` [$T = 25 \rightarrow T = 100$] with $\Pi = \Delta(\mathcal{A})$.** The model is trained on the Gaussian reward. The evaluation considers multiple reward generation processes, including the training reward generation process and Reward Generalization[Gaussian → Uniform, Bernoulli, Sine-trend, Alternating, Noisy Alternating, Adaptive]. We compare the performance of our trained model (red) against classical baselines: FTRL with stepsize $\eta_{25}$ (blue) and $\eta_{100}$ (green), and FTL (orange). The shaded regions represent the standard deviation over 100 random instances. The results demonstrate that our model consistently achieves sublinear regret across diverse reward generation processes with $T = 100$ despite being trained only for the Gaussian reward with $T = 25$.

$\Pi = B(\mathbf{0}_d, 1, \|\cdot\|_2)$ **- Regret Over Time**

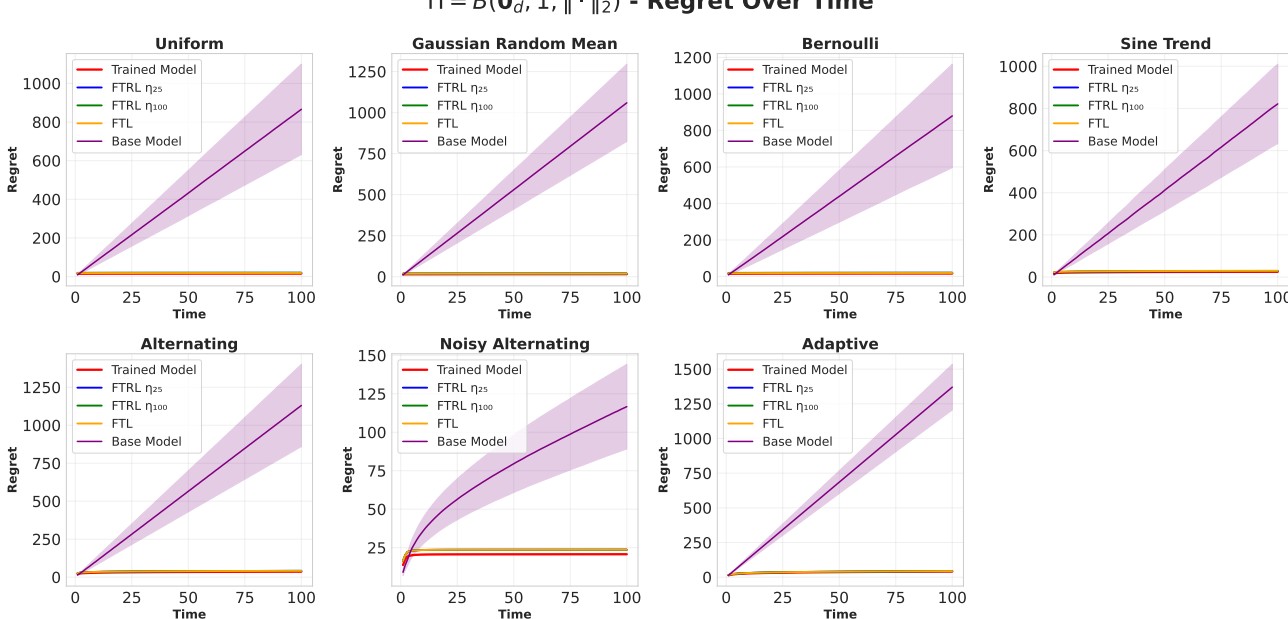

*Figure 7.* **The regret over time of the Transformers with numerical input/output for the FOL environment under `Horizon Generalization`[$T = 25 \rightarrow T = 100$] with $\Pi = B(\mathbf{0}_d, 1, \|\cdot\|_2)$.** The model is trained on the Gaussian reward, and the evaluation considers multiple reward generation processes, including the training reward generation process and multiple kinds of `Reward Generalization`[Gaussian $\rightarrow$ Uniform, Bernoulli, Sine-trend, Alternating, Noisy Alternating, Adaptive]. We compare the performance of our trained model (red) against classical baselines: FTRL and FTL. The shaded areas indicate the standard deviation across 100 random instances.

We report both $p_{\text{trend}}$ and $\widehat{\beta}$ for each reward generation process to assess the generalization capability of our Transformer-based agent in Table 10 and 11. Across all reward generation processes, we observe that $\widehat{\beta} < 1$ with consistently low $p_{\text{trend}}$, indicating that the cumulative regret grows sublinearly with high statistical confidence. These findings are further supported by the regret over time presented in the accompanying figures. Notably, this generalization holds despite the model being trained under a fixed time horizon ($T = 25$) and a single distributional family—namely, the Gaussian reward. These results suggest that our Transformer is capable of extrapolating to unseen temporal scales and diverse reward structures, highlighting its robustness in the FOL environment in both $\Pi = \Delta(\mathcal{A})$ and $\Pi = B(\mathbf{0}_d, 1, \|\cdot\|_2)$. We also provide the regret over time for each reward generation process that we evaluate in Figure 6 and Figure 7.

*Table 10.* **Summary of $p_{\mathbf{reg}}$ and fitted regret exponent $(\widehat{\beta})$ with Horizon Generalization[$T = 25 \to T = 100$] for the FOL environment** with $\Pi = \Delta(\mathcal{A})$. We provide results on both the training distribution and multiple Reward Generalization.

| Reward | Algorithm | $\widehat{\beta}$ | $p_{\mathrm{reg}}$ |
|---|---|---|---|
| Adaptive | Trained | 0.430 | < 0.001 |
| | FTRL $\eta_{25}$ | 0.332 | < 0.001 |
| | FTRL $\eta_{100}$ | 0.165 | < 0.001 |
| | FTL | 6.665 | < 0.001 |
| Alternating | Trained | 0.062 | < 0.001 |
| | FTRL $\eta_{25}$ | 0.322 | < 0.001 |
| | FTRL $\eta_{100}$ | 0.163 | < 0.001 |
| | FTL | 2.300 | < 0.001 |
| Bernoulli | Trained | 0.580 | < 0.001 |
| | FTRL $\eta_{25}$ | 0.223 | < 0.001 |
| | FTRL $\eta_{100}$ | 0.410 | < 0.001 |
| | FTL | 0.026 | < 0.001 |
| Gaussian | Trained | 0.665 | < 0.001 |
| | FTRL $\eta_{25}$ | 0.212 | < 0.001 |
| | FTRL $\eta_{100}$ | 0.495 | < 0.001 |
| | FTL | 0.003 | < 0.001 |
| Noisy Alternating | Trained | 0.003 | 0.051 |
| | FTRL $\eta_{25}$ | 0.003 | 0.042 |
| | FTRL $\eta_{100}$ | 0.001 | 0.659 |
| | FTL | 0.007 | < 0.001 |
| Sine-trend | Trained | 0.085 | < 0.001 |
| | FTRL $\eta_{25}$ | 0.123 | < 0.001 |
| | FTRL $\eta_{100}$ | 0.076 | < 0.001 |
| | FTL | 1.092 | < 0.001 |
| Uniform | Trained | 0.995 | < 0.001 |
| | FTRL $\eta_{25}$ | 0.247 | < 0.001 |
| | FTRL $\eta_{100}$ | 0.514 | < 0.001 |
| | FTL | 0.006 | < 0.001 |

*Table 11.* **Summary of $p_{\mathbf{reg}}$ and $\widehat{\beta}$ with Horizon Generalization[$T = 25 \to T = 100$] for the FOL environment** with $\Pi = B(\mathbf{0}_d, 1, \|\cdot\|_2)$. We provide results on both the training distribution and multiple Reward Generalization.

| Reward | Algorithm | $\widehat{\beta}$ | $p_{\mathrm{reg}}$ |
|---|---|---|---|
| Adaptive | Trained | 0.166 | < 0.001 |
| | FTRL $\eta_{25}$ | 0.170 | < 0.001 |
| | FTRL $\eta_{100}$ | 0.170 | < 0.001 |
| | FTL | 0.170 | < 0.001 |
| Alternating | Trained | 0.084 | < 0.001 |
| | FTRL $\eta_{25}$ | 0.086 | < 0.001 |
| | FTRL $\eta_{100}$ | 0.086 | < 0.001 |
| | FTL | 0.086 | < 0.001 |
| Bernoulli | Trained | 0.010 | < 0.001 |
| | FTRL $\eta_{25}$ | 0.010 | < 0.001 |
| | FTRL $\eta_{100}$ | 0.010 | < 0.001 |
| | FTL | 0.013 | < 0.001 |
| Gaussian | Trained | 0.007 | < 0.001 |
| | FTRL $\eta_{25}$ | 0.003 | < 0.001 |
| | FTRL $\eta_{100}$ | 0.003 | < 0.001 |
| | FTL | 0.003 | < 0.001 |
| Noisy Alternating | Trained | 0.009 | < 0.001 |
| | FTRL $\eta_{25}$ | 0.009 | 0.001 |
| | FTRL $\eta_{100}$ | 0.010 | 0.001 |
| | FTL | 0.009 | 0.001 |
| Sine-trend | Trained | 0.041 | < 0.001 |
| | FTRL $\eta_{25}$ | 0.042 | < 0.001 |
| | FTRL $\eta_{100}$ | 0.042 | < 0.001 |
| | FTL | 0.042 | < 0.001 |
| Uniform | Trained | 0.006 | < 0.001 |
| | FTRL $\eta_{25}$ | 0.004 | < 0.001 |
| | FTRL $\eta_{100}$ | 0.005 | < 0.001 |
| | FTL | 0.005 | < 0.001 |

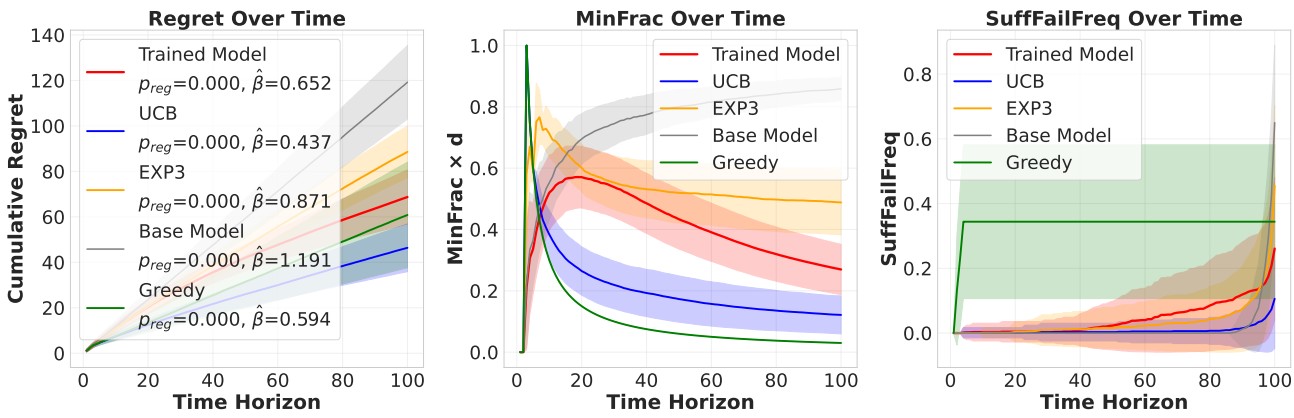

*Figure 8.* **Evaluation of the trained Transformers with numerical input/output for the MAB environment under `Horizon Generalization`[$T = 25 \rightarrow T = 100$]**, trained and evaluated with the Gaussian reward. The trained model exhibits sublinear regret growth, as validated by our validation framework and comparison with baseline algorithms. `MinFrac`($t$) also exhibits a trend that reflects a proper *E-E tradeoff*: the metric first increases (active exploration in early rounds) and later decreases (progressive exploitation of optimal actions). Meanwhile, `SuffFailFreq`($t$) maintains a consistently lower value than the Greedy and the Base Model near the end of the horizon, indicating convergence toward more optimal action choices. Overall, these patterns demonstrate an improved *E-E tradeoff* compared to baselines.

### K.2. Multi-Armed Bandits

For the MAB environment, using Algorithm 2, we train a single-layer linear attention model. As shown in Figure 8, our trained model exhibits sublinear regret in longer horizons (*i.e.,* with `Horizon Generalization`). Notably, the `MinFrac`($t$) metric shows an *E-E* trend: it first increases, reflecting active exploration in early rounds, and later decreases as the model progressively exploits the best actions. This improves upon the base model before training and aligns with the behavior of other baseline algorithms. The consistently lower `SuffFailFreq`($t$) of our model compared to the greedy policy near the end of the horizon indicates convergence toward more optimal action choices while preserving adequate exploration. We also provide the regret over time and the final regret distribution for the Gamma reward in Figure 9, where the same patterns can also be observed. It is worth noting that although we train our model with the Gaussian reward and horizon $T = 25$, it generalizes to *different horizon lengths* and *reward types*, demonstrating good `Reward Generalization` and `Horizon Generalization` performance in such MAB environments.

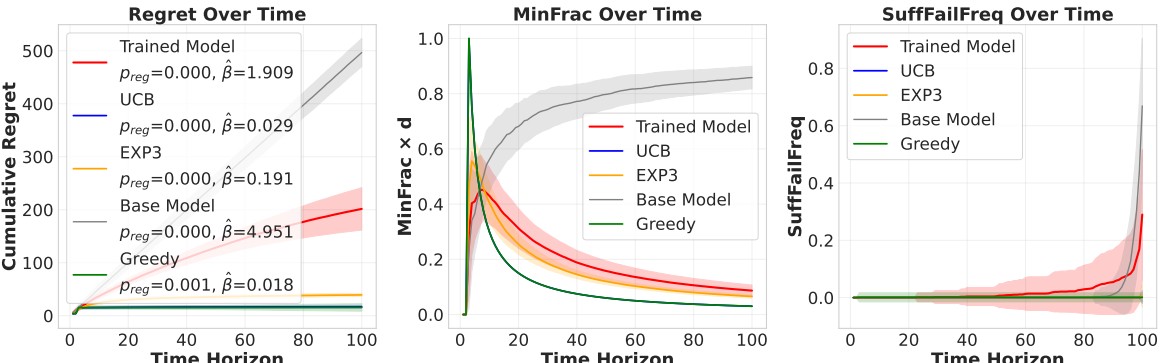

*Figure 9.* **The regret over time and the final regret distribution for the Gamma reward under `Horizon Generalization`[$T = 25 \rightarrow T = 100$].** `MinFrac`($t$) also exhibits a trend that reflects a proper *E-E tradeoff*: the metric first increases (active exploration in early rounds) and later decreases (progressive exploitation of optimal actions). Meanwhile, `SuffFailFreq`($t$) maintains a consistently lower value than the Greedy and the Base Model near the end of the horizon, indicating convergence toward more optimal action choices.

# L. Deferred Experimental Results for Section 5

In this section, we provide more results for open-weight LLMs in Section 5. We mainly consider two parts of the experimental results deferred from Section 5: 1) the experimental results of *distributional* output, where we identify a major problem for *distributional* output for the open-weight LLM, Gemma-2-9b-it – *high-entropy simplex bias*; 2) the experimental results of a smaller open-weight LLM, Phi-3.5-mini-instruct. We also provide an example that demonstrates improved reasoning rationales and additional figures deferred from Section 5.3.

## L.1. Experimental Results of *Distributional* Output

In Section 5, we provide results for only *action-based* output, while in this part of the Appendix, we provide results for *distributional* output, and explore the reason why *distributional* output cannot achieve a similar improvement as we have seen in the *action-based* output in open-weight LLMs like Gemma-2-9b-it. We identify that for Gemma-2-9b-it, the simplex it generates usually concentrates around the uniform simplex, and cannot assign a very high probability mass to one action, and a very low probability mass to other actions. We refer to this problem as *high-entropy simplex bias*, where **ITERATIVE RMFT** cannot mitigate this problem in Gemma-2-9b-it. In this Appendix and Section 6, however, the experimental results for the stronger closed-weight LLM like GPT-4o mini demonstrate that *high-entropy simplex bias* is not a universal problem for any LLMs.

In this part, we will mainly focus on the FOL environment to understand which type of output can be used for **ITERATIVE RMFT**.

### L.1.1. EXPERIMENTAL RESULTS OF $d = 3$ ACTIONS

We first provide the results of $d = 3$ actions using *distributional* output. Similar to the experimental results in Section 5.3, the model is trained with $d = 3$ actions and a time horizon of $T = 25$ using a mixture of the Gaussian, Uniform, and Sine-trend rewards, with policy space $\Pi = \Delta(\mathcal{A})$. However, the experiments in this section use *distributional* output and *policy-with-reasoning* format. As in Section 5.3, we also evaluate it with time horizon $T = 50$ and a mixture of the Gaussian, Uniform, and Sine-trend rewards. From Figure 10, we can conclude that **ITERATIVE RMFT** cannot minimize the regret for $d = 3$ actions. However, in Section 6, we consistently succeed in training with distributional output. We therefore hypothesize that this is due to the *high-entropy simplex bias* of the open-weight (or smaller) models, which we explain in the following subsection.

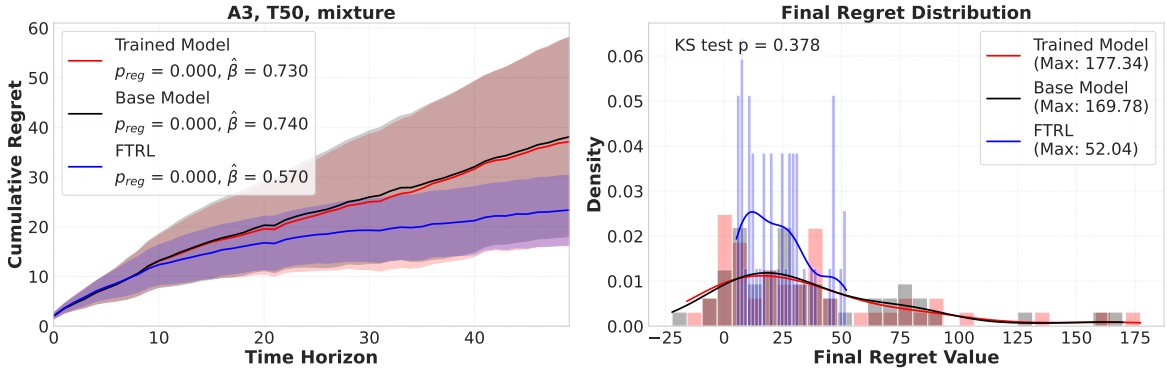

*Figure 10.* **The regret over time and the final regret distribution for the FOL environment with `Horizon Generalization`[$T = 25 \to T = 50$] on Gemma-2-9b-it**, trained and tested on a mixture of the Gaussian, Uniform and Sine-trend rewards. Our trained model shows a similar regret value and regret growth rate to the base model, indicating that **ITERATIVE RMFT** with *distributional* outputs is ineffective on Gemma-2-9b-it.

### L.1.2. EVIDENCE OF *High-Entropy Simplex Bias*

Assume that the LLM never observes information about numbers in the context of distributional output, and thus never encounters probabilities smaller than 0.1 (*i.e.*, no extreme outputs). Since the LLM does not operate through a number but instead treats each digit as a token, the absence of such tokens in training means it cannot generate them. Therefore, if we expect the LLM to produce a versatile policy (or probability distribution), it must at least be exposed to those specific

tokens during training. We hypothesize that smaller open-weight LLMs are unable to produce high-entropy policies. We illustrate this with experiments by examining the distribution of the probabilistic simplex generated by Gemma-2-9b-it.

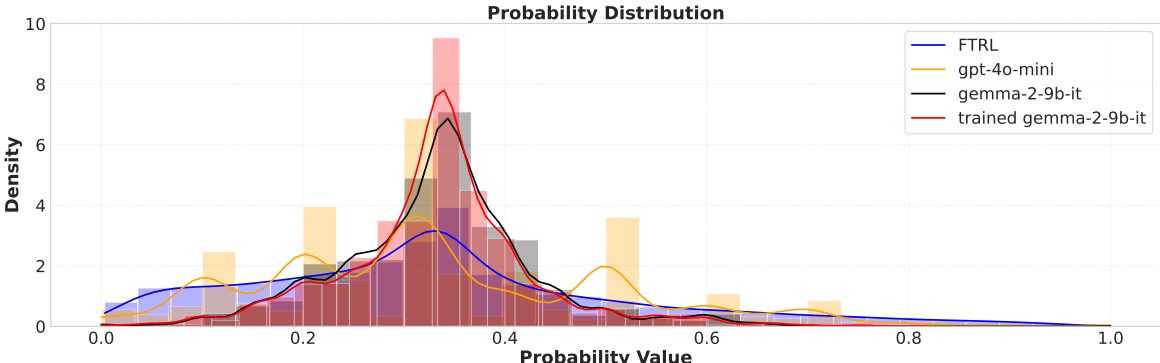

*Figure 11.* **Distribution of the probabilistic simplex with** $d = 3$ **actions.** We collect the probability values of multiple probabilistic simplices generated by Gemma-2-9b-it, and plot the distribution. The simplex of Gemma-2-9b-it has significantly higher entropy than baseline algorithms and the more capable closed-weight LLM, GPT-4o mini. The simplex of the trained model has a similar high entropy.

Figure 11 indicates that FTRL and the capable closed-weight LLM like OpenAI's GPT-4o mini, although they may have some specific preference for certain values (like 0.10), generally include both simplices with high entropy (values concentrated near uniform policy) and low entropy (values which are relatively far from near uniform policy). However, the simplex of the open-weight LLMs like Gemma-2-9b-it concentrates near a uniform policy with a low variance. The trained model does not show an improvement in the *high-entropy simplex bias* problem, which is a potential limitation for **ITERA-TIVE RMFT**, especially in small LLMs, where its original distribution for probabilistic simplices has certain preferences or limitations.

L.1.3. EXPERIMENTAL RESULTS OF SIMPLER PROBABILISTIC SIMPLICES

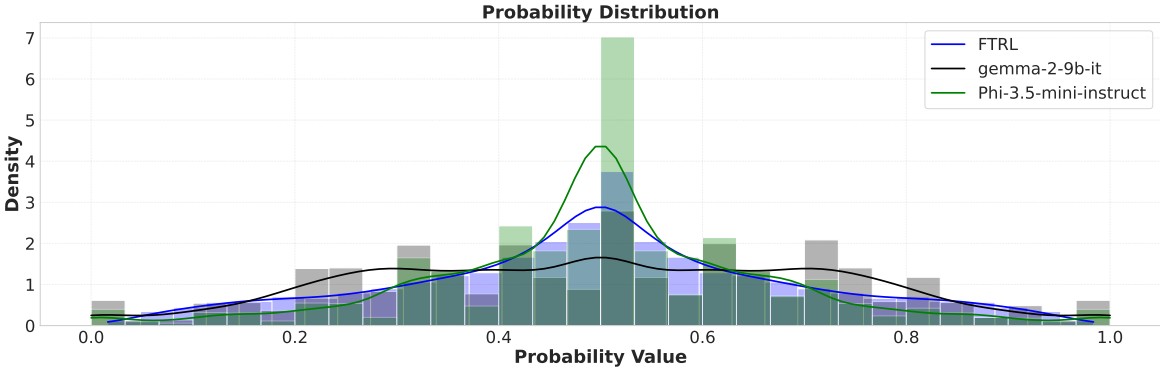

*Figure 12.* **Distribution of the probabilistic simplex with** $d = 2$ **actions.** We gather the probability values from multiple probabilistic simplices generated by Gemma-2-9b-it and Phi-3.5-mini-instruct and visualize their distributions. The results indicate that the simplices produced by both models are not severely affected by the *high-entropy simplex bias*.

From the discussion above, we suspect that the problem for *distributional* output is that the model faces a serious *high-entropy simplex bias* problem. In this part, we examine simpler probabilistic simplices of action $d = 2$ for the FOL environment. The experimental results show that if the probabilistic simplices of action $d = 2$ are simple enough, Gemma-2-9b-it or even smaller open-weight LLMs like Phi-3.5-mini-instruct can have improved regret behavior when applying **ITERATIVE RMFT**.

We first show results of the distribution of the probabilistic simplex with action $d = 2$ in Figure 12, which demonstrates that for a simpler simplex (action $d = 2$), the *high-entropy simplex bias* is not that severe.

We also provide results on Gemma-2-9b-it. We train the model with $d = 2$ actions and a time horizon of $T = 25$ using

a mixture of the Gaussian, Uniform, and Sine-trend rewards, and evaluate it with the same distribution and a longer time horizon $T = 50$ in Table 12 and Figure 13.

| | Gaussian | | | Uniform | | | Sine-trend | | |
|---|---|---|---|---|---|---|---|---|---|
| | max(LR) | avg(LR) | $\widehat{\beta}$ | max(LR) | avg(LR) | $\widehat{\beta}$ | max(LR) | avg(LR) | $\widehat{\beta}$ |
| FTRL | 36.60 | 20.22 | 0.87 | 43.25 | 29.44 | 0.73 | 9.72 | 4.53 | 0.2 |
| Gemma-2-9b-it | 47.44 | 21.18 | 0.91 | 71.28 | 28.93 | 0.84 | 56.04 | 9.41 | 0.34 |
| Trained Gemma-2-9b-it | 51.28 | 21.14 | 0.92 | 59.85 | 25.80 | 0.82 | 52.88 | 8.61 | 0.24 |

*Table 12.* **Summary of the regret value for the FOL environment with `Horizon Generalization`[$T = 25 \rightarrow T = 50$] on Gemma-2-9b-it**, trained and tested on a mixture of the Gaussian, Uniform, and Sine-trend rewards with $d = 2$ actions. The trained model achieves a lower regret value and sublinear regret growth than the base model on the Uniform and Sine-trend rewards.

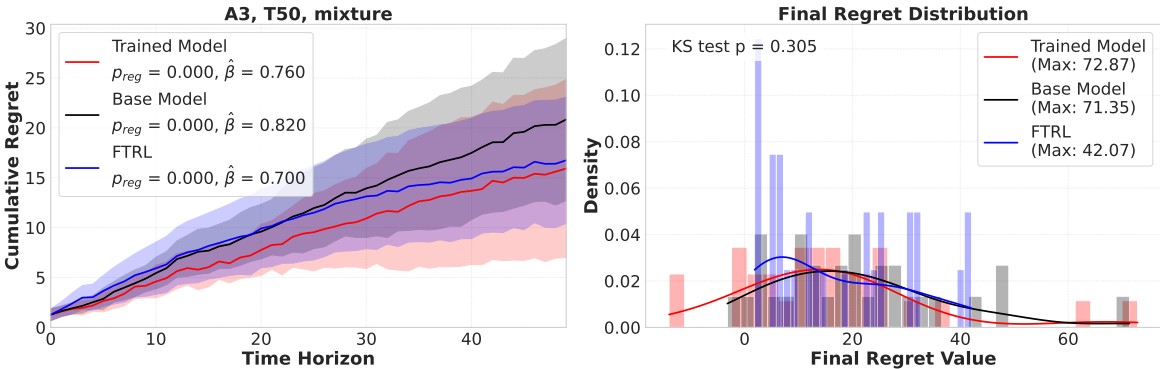

*Figure 13.* **The regret over time and the final regret distribution for the FOL environment under `Horizon Generalization`[$T = 25 \rightarrow T = 50$] on Gemma-2-9b-it,** trained and evaluated on a mixture of the Gaussian, Uniform, and Sine-trend rewards with $d = 2$ actions. The trained model attains **lower regret** than both the baseline algorithm FTRL and the base model Gemma-2-9b-it, demonstrating **sublinear regret behavior**.

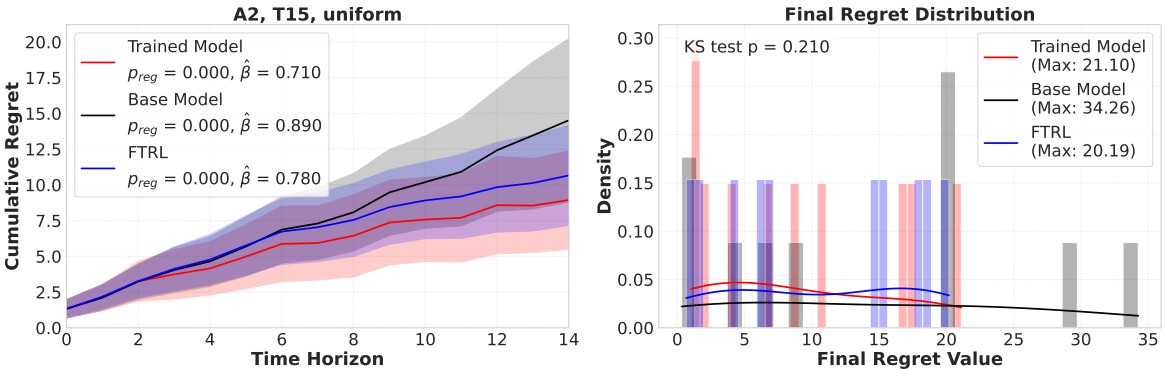

*Figure 14.* **The regret over time and the final regret distribution for the FOL environment under `In-Distribution` on Phi-3.5-mini-instruct**, trained and tested with the Uniform reward ($T = 15$, $d = 2$). The trained model shows consistently lower cumulative regret and a slower rate of regret increase compared to both the baseline algorithm (FTRL) and the untrained base model, confirming sublinear regret performance.

We then provide results on Phi-3.5-mini-instruct. We train the model with $d = 2$ actions and a time horizon of $T = 15$ using the Uniform reward, and evaluate it with the same distribution and the same time horizon $T = 15$ in Figure 14.

This result shows that for $d = 2$, **ITERATIVE RMFT** can reduce the regret value for Gemma-2-9b-it and even smaller open-weight LLMs like Phi-3.5-mini-instruct. Since $d = 2$ requires an easier simplex than $d = 3$, this result probably indicates that if Gemma-2-9b-it (or even Phi-3.5-mini-instruct) is capable enough to generate a "good" probabilistic simplex, **ITERATIVE RMFT** will still work for the *distributional* output.

### L.2. Comparing *Dialogue-Type* and *Summary-Type* Interactions in ITERATIVE RMFT

In this section, we compare *dialogue-type* and *summary-type* interactions in **ITERATIVE RMFT**. The advantages of *dialogue-type* interactions are as follows: (1) they retain the complete history of the interaction directly in the prompt, whereas *summary-type* interactions must extract information from previous outputs and integrate a manually or automatically generated summary into the current prompt—an approach that may be unnatural for real-world use cases; and (2) when applied to **ITERATIVE RMFT**, *summary-type* interactions are prone to overfitting.

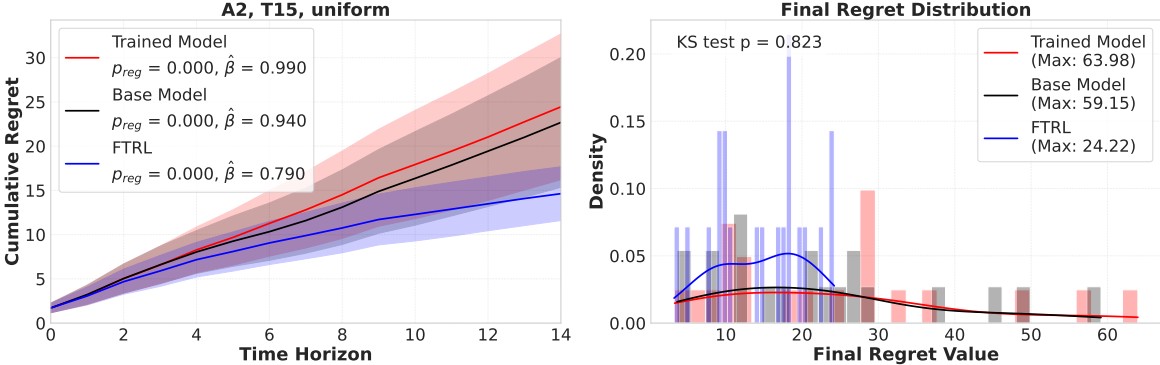

*Figure 15.* **The regret over time and the final regret distribution for the FOL environment under `In-Distribution` on Phi-3.5-mini-instruct**, trained and tested with the Uniform reward ($T = 15$, $d = 2$). Unlike the other experiments, this one is based on *summary-type* **interaction**. In this case, the trained model actually performs worse, showing even higher regret than the base model (Phi-3.5-mini-instruct).

We compare the results of *summary-type* interaction with *dialogue-type* interaction (see Figure 14 for *dialogue-type* interaction) on Phi-3.5-mini-instruct. For *summary-type* interaction, we also train the model with $d = 2$ actions and a time horizon of $T = 15$ using the Uniform reward, and evaluate it with the same distribution and the same time horizon $T = 15$. Figure 15 shows the regret behavior of *summary-type* interaction, where we find that the regret behavior of the trained model is worse than that of the base model. We identify this as a case of overfitting, possibly because the Phi-3.5-mini-instruct is not capable enough to do the training using *summary-type* interaction. The loss plot over iterative training in Figure 16 further supports this claim.

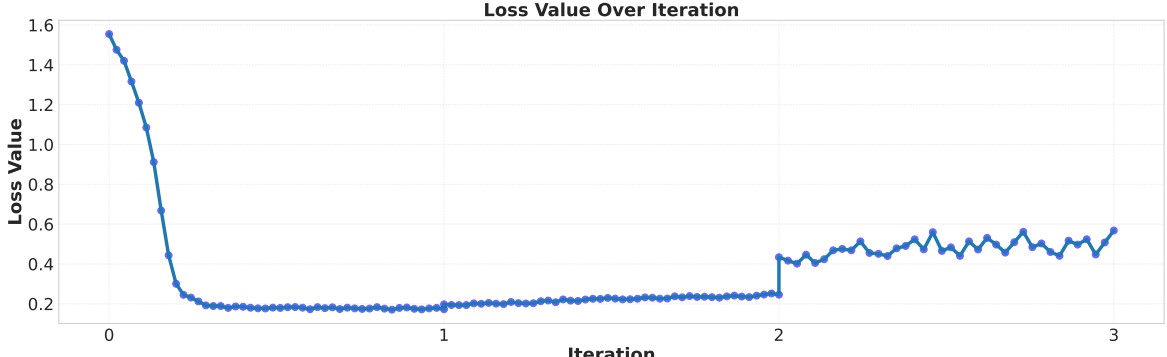

*Figure 16.* **Loss over iteration for the Uniform reward ($T = 15$) with *summary-type* interaction.** Although the loss drops significantly during iteration 1, the loss increases during iterations 2 and 3, showing that training is overfitting.

## L.3. Deferred Figures of the FOL Environment

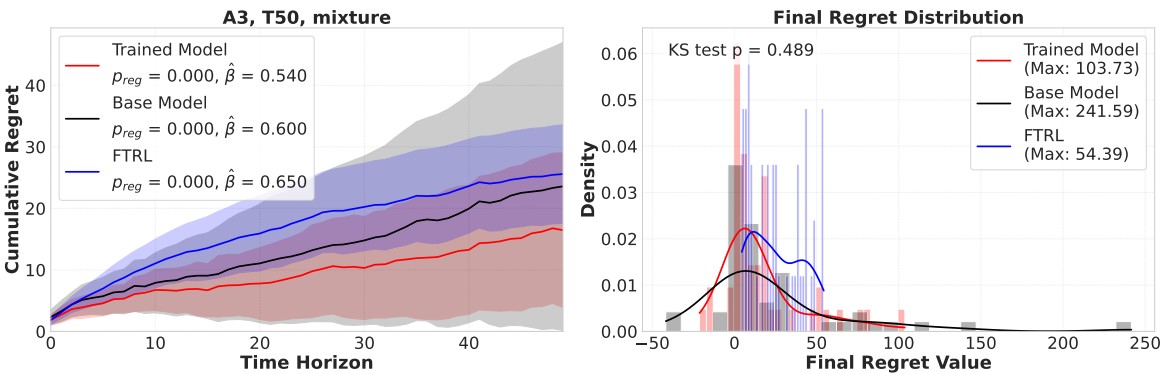

*Figure 17.* **The regret over time and final regret distribution for the FOL environment under `Horizon Generalization`[$T = 25 \rightarrow T = 50$] on Gemma-2-9b-it**, trained and evaluated with a mixture of the Gaussian, Uniform, and Sine-trend rewards, which shows a lower regret value and sublinear regret growth after ITERATIVE RMFT.

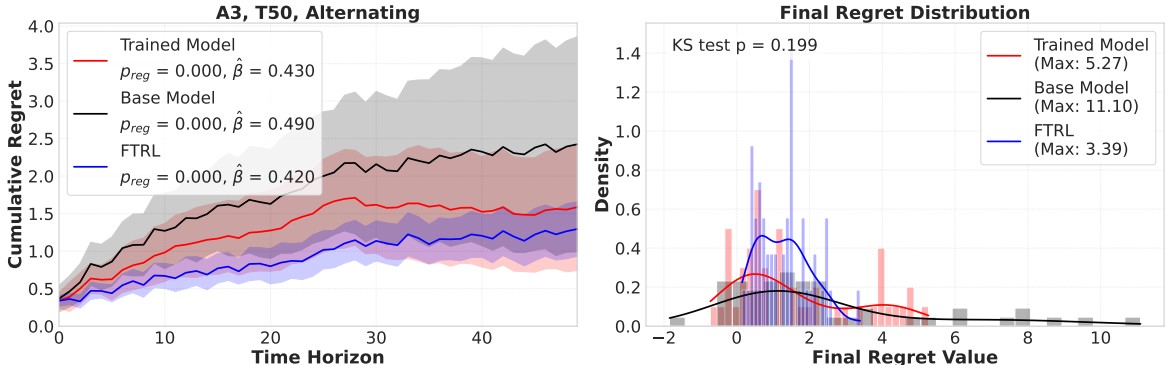

*Figure 18.* **The regret over time and the final regret distribution for the FOL environment under both `Horizon Generalization`[$T = 25 \rightarrow T = 50$] and `Reward Generalization`[a mixture of the Gaussian, Uniform, and Sine-trend rewards $\rightarrow$ Alternating] on Gemma-2-9b-it**, which shows lower empirical regret and sublinear regret growth after ITERATIVE RMFT.

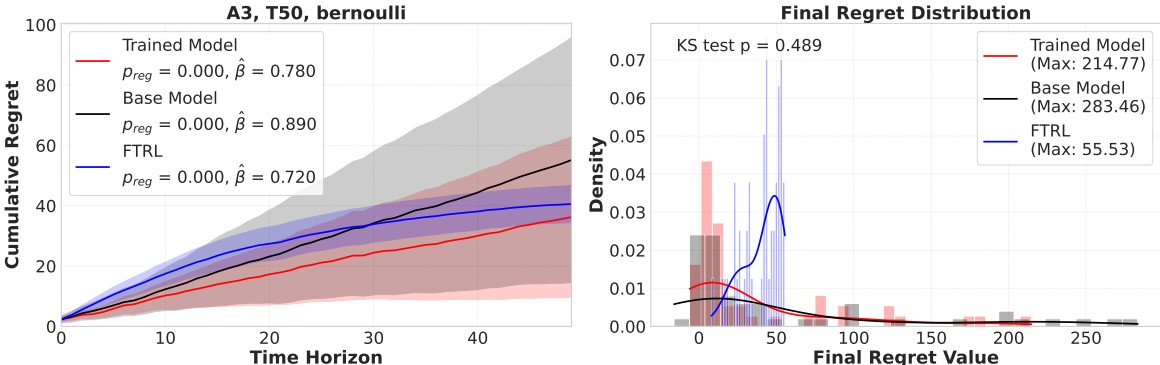

*Figure 19.* **The regret over time and the final regret distribution for the FOL environment under both `Horizon Generalization`[$T = 25 \rightarrow T = 50$] and `Reward Generalization`[a mixture of the Gaussian, Uniform, and Sine-trend rewards $\rightarrow$ Bernoulli] on Gemma-2-9b-it**, which shows lower empirical regret and sublinear regret growth after ITERATIVE RMFT.

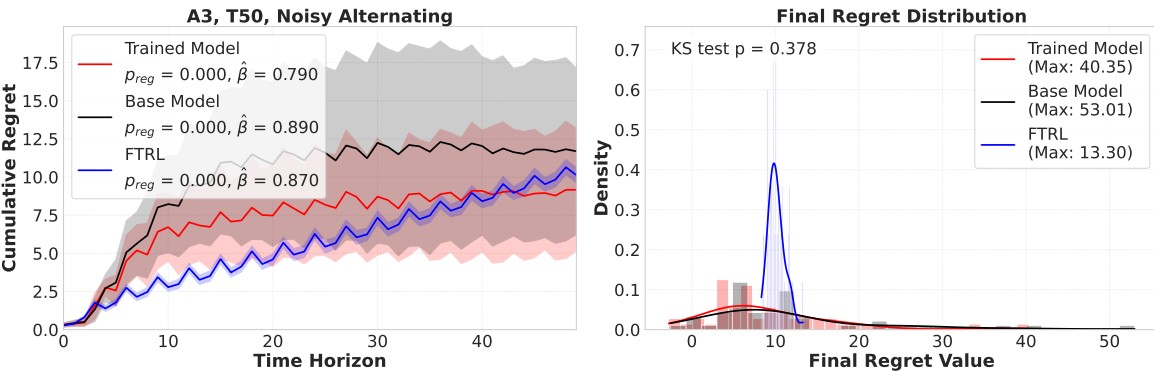

*Figure 20.* **The regret over time and the final regret distribution for the FOL environment under both `Horizon Generalization`[$T = 25 \rightarrow T = 50$] and `Reward Generalization`[a mixture of the Gaussian, Uniform, and Sine-trend rewards $\rightarrow$ Noisy Alternating] on Gemma-2-9b-it**, which shows lower empirical regret and sublinear regret growth after ITERATIVE RMFT.

### L.4. Deferred Figures of the MAB Environment

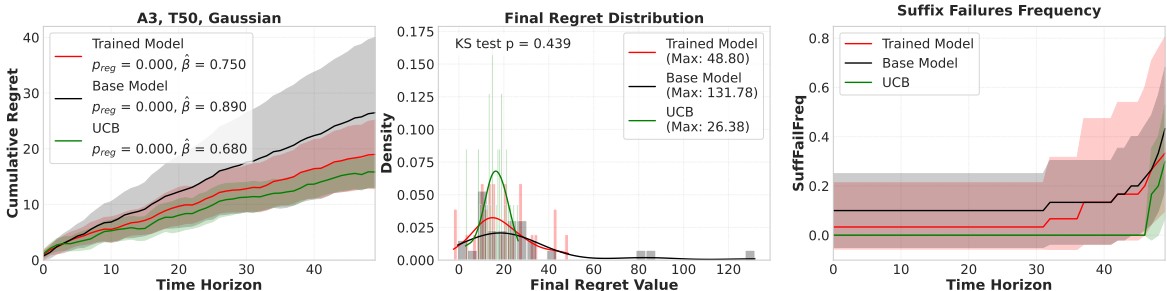

*Figure 21.* **The regret over time, the final regret distribution, and the exploration metric using `SuffFailFreq`($t$) for the MAB environment under `Horizon Generalization`[$T = 25 \rightarrow T = 50$] on Gemma-2-9b-it**, which demonstrates a lower regret value, sublinear growth rate, and improved *E-E tradeoff* after ITERATIVE RMFT.

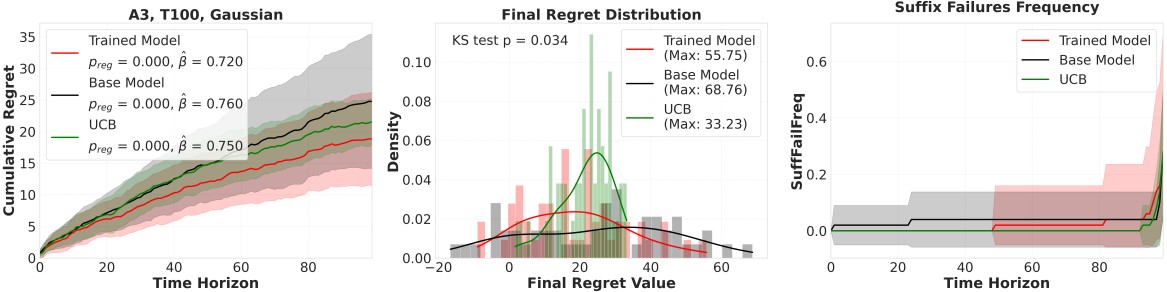

*Figure 22.* **The regret over time, the final regret distribution, and the exploration metric using `SuffFailFreq`($t$) for the MAB environment under `Horizon Generalization`[$T = 25 \rightarrow T = 100$] on Qwen3-8B**, which demonstrates a lower regret value, sublinear growth rate, and improved *E-E tradeoff* after ITERATIVE RMFT. The exploitation metric `MinFrac`($t$) for Qwen3-8B is reported separately in Figure 23.

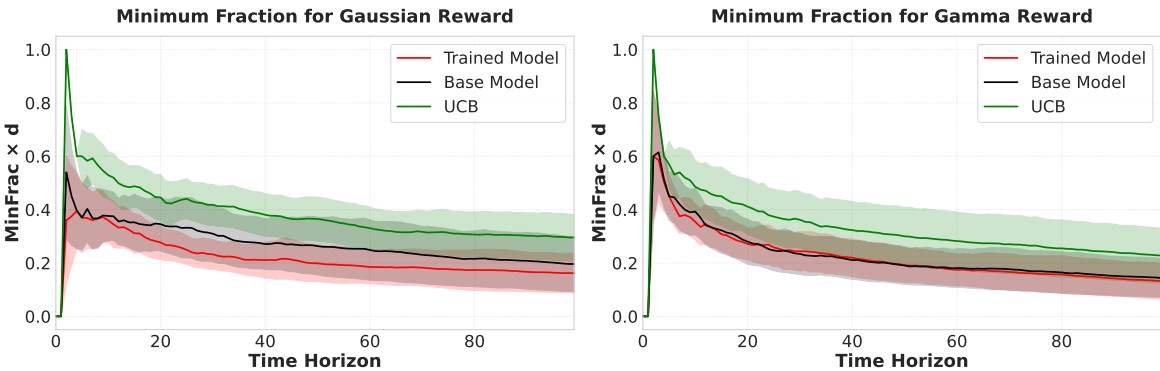

*Figure 23.* **The exploitation metric `MinFrac`(t) for the MAB environment under `Horizon Generalization`[T = 25 →** T = 100] **on Qwen3-8B** using both the Gaussian and Gamma rewards. The result shows that the base model already exhibits strong exploitation behavior.

### L.5. Assessing How ITERATIVE RMFT Improves the Reasoning Rationales and Enhances Robustness to Reward Fluctuations on the Open-Weight LLM Qwen3-8B

**Improved Reasoning Rationales.**   In this section, we visualize representative reasoning rationales produced by the base and trained models of Qwen3-8B in Figure 24. The color-coded examples illustrate how our post-training paradigm improves the reasoning of the models in terms of both the *semantic–numerical alignment* and the *E–E tradeoff*. Here, *semantic–numerical alignment* reflects the model's capability to accurately interpret numerical concepts from language descriptions, while the *E–E tradeoff* captures the model's decision-making ability in online DM tasks.

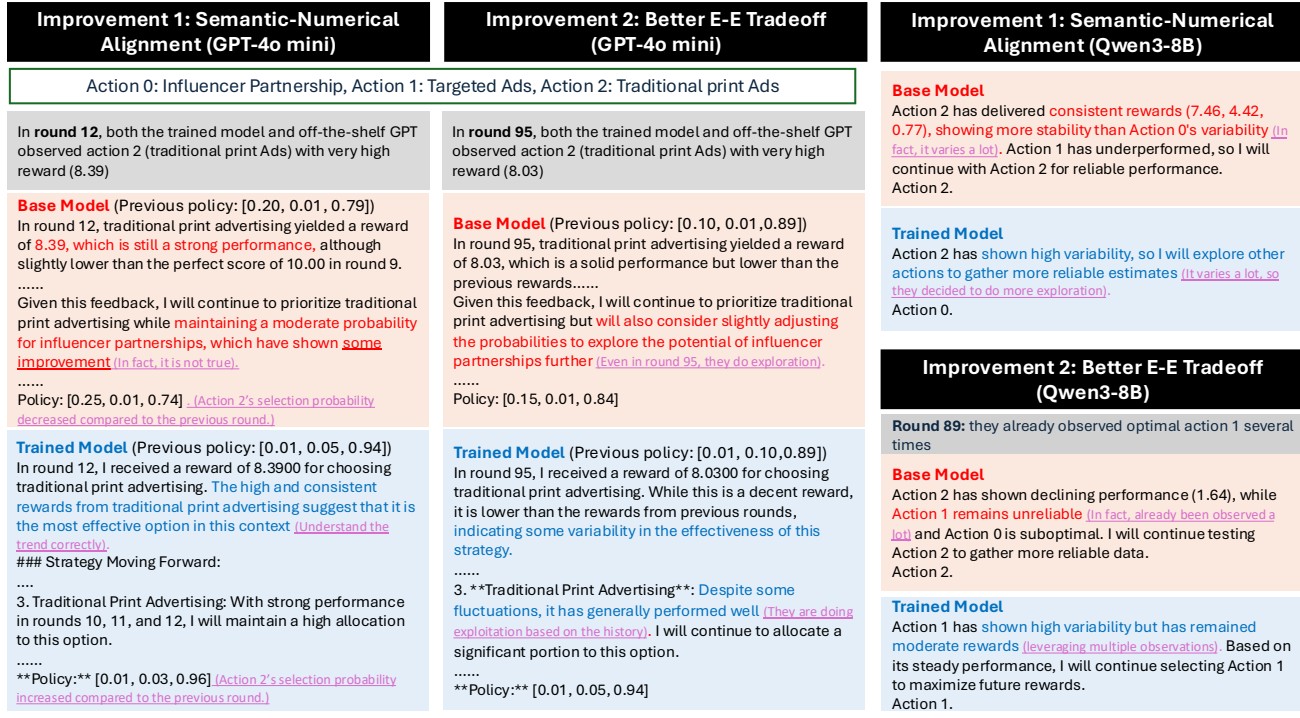

*Figure 24.* **Illustration of the reasoning rationales generated by the base model and the trained model in the MAB environment for both GPT-4o mini and Qwen3-8B.** The figure highlights two major improvements observed after regret-based post-training: (1) enhanced *semantic–numerical alignment*, and (2) improved *E–E tradeoff*. Red text indicates incorrect or inconsistent reasoning by the base model, blue text denotes correct and reward-aligned reasoning by the trained model, and purple text provides explanatory annotations clarifying why each response is good or bad.

*Figure 25.* **The Instability-Induced Optimal Action Rejection Fraction Over Time.** The trained model has a lower fraction than the base model, indicating its robustness to unstable reward behavior.

**Quantitative Analysis: Enhanced Robustness to Reward Fluctuations**   To gain deeper insight into how **ITERATIVE RMFT** enhances the decision-making capability of LLMs on Qwen3-8B, we perform a quantitative analysis on the robustness to reward fluctuations comparing the outputs of the base model and the trained model for the MAB environment, inspired by a qualitative observation that the base model exhibits high sensitivity to reward fluctuations and struggles to handle optimal actions with both high mean rewards and high variance.

Using GPT-4o mini as a zero-shot classifier, we categorize the model outputs into two groups depending on whether they explicitly mention that the optimal action is avoided due to unstable reward behavior and report the resulting fraction in Figure 25.

We observe that after **ITERATIVE RMFT**, the instability-induced optimal action rejection fraction drops, indicating that the trained model exhibits an improvement in robustness to unstable reward behavior.

## M. Deferred Experimental Results for Section 6

We include here the tables that we deferred from the main paper due to page constraints.

### M.1. Full-Information Online Learning

M.1.1. IN-DISTRIBUTION EVALUATION

In the In-Distribution evaluation, which is the same configuration as training, we observe improvements in both the max value and average value of the final regret and the empirical growth of regret in Table 13.

| | Gaussian | | | Uniform | | | Sine-trend | | |
|---|---|---|---|---|---|---|---|---|---|
| | max(LR) | avg(LR) | $\widehat{\beta}$ | max(LR) | avg(LR) | $\widehat{\beta}$ | max(LR) | avg(LR) | $\widehat{\beta}$ |
| FTRL | 29.72 | 18.52 | 0.69 | 33.24 | 19.32 | 0.81 | 37.32 | 29.22 | 0.36 |
| GPT-4o mini | 40.16 | 16.35 | 0.64 | 36.75 | 15.69 | 0.74 | 37.24 | 8.98 | 0.43 |
| Trained GPT-4o mini | 29.89 | 15.03 | 0.61 | 25.76 | 13.12 | 0.66 | 25.39 | 8.35 | 0.39 |

*Table 13.* **Summary of the regret value for the FOL environment under In-Distribution on GPT-4o mini,** trained and evaluated on a mixture of the Gaussian, Uniform, and Sine-trend rewards with time horizon $T = 15$. We report the maximum and average values of the cumulative regret at round $T$, denoted as max(LR) and avg(LR), respectively. The exponent $\widehat{\beta}$ is estimated from the empirical growth of regret, where regret scales approximately as $\Theta(t^{\widehat{\beta}})$ (see Section 2.2). Highlighted (yellow) cells indicate the lower regret value between those corresponding to *GPT-4o mini* and *Trained GPT-4o mini*, emphasizing which performs better in terms of regret at round $T$ and growth rate exponent ($\widehat{\beta}$) for each reward generation process. This model consistently shows improved regret over time for In-Distribution tasks.

M.1.2. EVALUATION FOR GENERALIZATION

|  | **Gaussian** | | | **Uniform** | | | **Sine-trend** | | |
|---|---|---|---|---|---|---|---|---|---|
|  | max(LR) | avg(LR) | $\widehat{\beta}$ | max(LR) | avg(LR) | $\widehat{\beta}$ | max(LR) | avg(LR) | $\widehat{\beta}$ |
| FTRL | 30.36 | 18.44 | 0.63 | 29.30 | 17.55 | 0.76 | 25.54 | 10.65 | 0.47 |
| GPT-4o mini | 31.07 | 17.10 | 0.61 | 34.91 | 14.67 | 0.72 | 33.85 | 9.47 | 0.42 |
| Trained GPT-4o mini | 36.26 | 16.56 | 0.60 | 24.21 | 12.76 | 0.64 | 33.65 | 8.57 | 0.39 |

*Table 14.* **Summary of the regret value for the FOL environment under `Linguistic Context Generalization`[GPT-4o → Gemini-2.0-Flash] on GPT-4o mini,** trained and evaluated on a mixture of the Gaussian, Uniform, and Sine-trend rewards with time horizon $T = 15$. Highlighted (yellow) cells indicate the lower value between the base model and the trained model, highlighting which model performs better in terms of cumulative regret at round $T$ and the growth rate exponent $\widehat{\beta}$ for each reward generation process. These results demonstrate consistent improvement of the trained model under `Linguistic Context Generalization`.

|  | **Alternating** | | | **Bernoulli** | | |
|---|---|---|---|---|---|---|
|  | max(LR) | avg(LR) | $\widehat{\beta}$ | max(LR) | avg(LR) | $\widehat{\beta}$ |
| FTRL | 6.19 | 6.19 | 0.20 | 28.33 | 10.82 | 0.72 |
| GPT-4o mini | 35.70 | 28.21 | 0.71 | 32.14 | 10.10 | 0.71 |
| Trained GPT-4o mini | 37.30 | 28.20 | 0.70 | 30.85 | 9.66 | 0.73 |

*Table 15.* **Summary of the regret value for the FOL environment under `Reward Generalization`[a mixture of the Gaussian, Uniform, and Sine-trend rewards → Alternating, Bernoulli] on GPT-4o mini,** trained on a mixture of the Gaussian, Uniform, and Sine-trend rewards. Highlighted (yellow) cells indicate the lower value between the base model and the trained model, highlighting which model performs better in terms of cumulative regret at round $T$ and the growth rate exponent $\widehat{\beta}$ for each reward generation process. The trained model achieves comparable or better performance under `Reward Generalization`, demonstrating that its regret behavior does not significantly degrade under distribution shift of the reward.

|  | **Gaussian** | | | **Uniform** | | | **Sine-trend** | | |
|---|---|---|---|---|---|---|---|---|---|
|  | max(LR) | avg(LR) | $\widehat{\beta}$ | max(LR) | avg(LR) | $\widehat{\beta}$ | max(LR) | avg(LR) | $\widehat{\beta}$ |
| FTRL | 34.09 | 23.46 | 0.68 | 31.41 | 21.14 | 0.77 | 29.06 | 10.72 | 0.42 |
| GPT-4o mini | 49.54 | 22.49 | 0.65 | 42.92 | 18.39 | 0.72 | 28.60 | 11.56 | 0.46 |
| Trained GPT-4o mini | 38.86 | 20.93 | 0.62 | 39.68 | 16.90 | 0.69 | 28.42 | 12.45 | 0.47 |

*Table 16.* **Summary of the regret value for the FOL environment under `Action Space Size Generalization`[$d = 3 \rightarrow d = 4$] on GPT-4o mini**, trained and evaluated on a mixture of the Gaussian, Uniform, and Sine-trend rewards. Highlighted (yellow) cells indicate the lower value between the base model and the trained model, highlighting which model performs better in terms of cumulative regret at round $T$ and growth rate exponent $\widehat{\beta}$ for each reward generation process. The trained model demonstrates comparable or improved performance, indicating robustness to increased action dimensionality.

## M.2. Multi-Armed Bandits

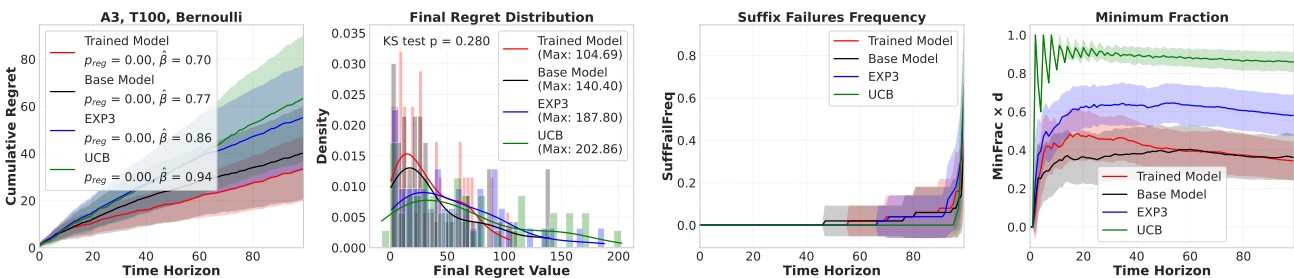

*Figure 26.* **The regret over time, the final regret distribution, and the exploration and exploitation metrics for the MAB environment under both `Horizon Generalization`[$T = 25 \rightarrow T = 100$] and `Reward Generalization`[Gaussian → Bernoulli] on GPT-4o mini**, which shows a lower regret and sublinear regret behavior for the trained model. The trained GPT-4o mini model sustains effective exploration and adapts its exploitation strategy over time.

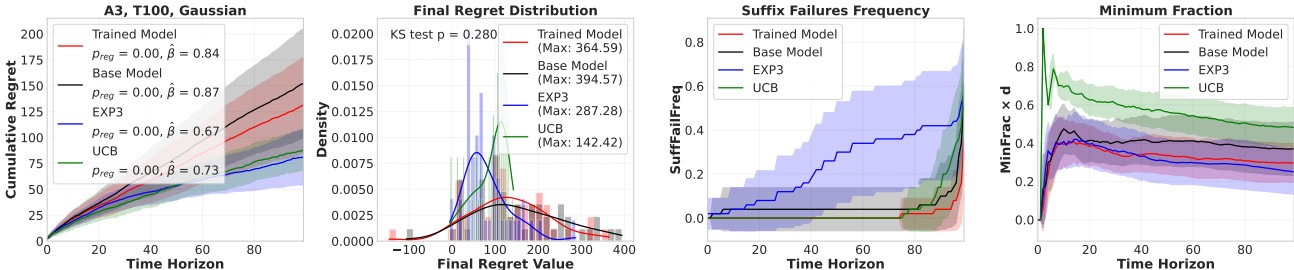

*Figure 27.* **The regret over time, the final regret distribution, and the exploration and exploitation metrics for the MAB environment under `Linguistic Context Generalization`[GPT-4o → Gemini-2.0-Flash] on GPT-4o mini using Gemini-2.0-Flash-generated contexts**, which shows a lower regret and sublinear regret behavior for the trained model. The trained GPT-4o mini model sustains effective exploration and adapts its exploitation strategy over time.

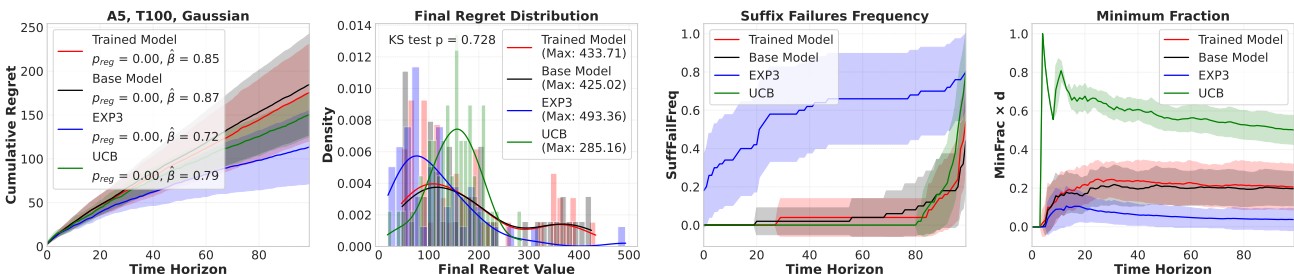

*Figure 28.* **The regret over time, the final regret distribution, and the exploration and exploitation metrics for the MAB environment under `Action Space Size Generalization`[$d = 3 \rightarrow d = 5$] on GPT-4o mini**. In this setting, the trained model has a slightly lower regret-growth exponent, but not lower maximum regret or final-regret distribution, and the trained GPT-4o mini model sustains effective exploration and adapts its exploitation strategy over time.

## M.3. Non-Stationary Multi-Armed Bandits

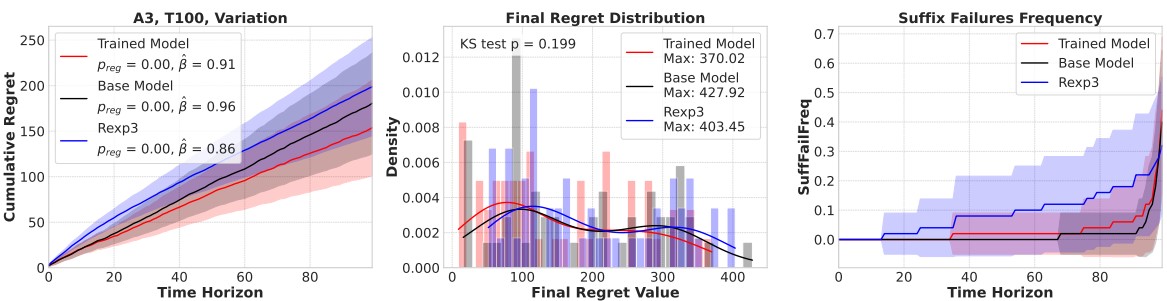

*Figure 29.* **The regret over time, the final regret distribution, and the exploration metric for the NS-MAB environment under Horizon Generalization**[$T = 25 \rightarrow T = 100$]**,** trained and evaluated with the Gradual Variation reward, which shows a lower regret and sublinear regret behavior for the trained model.

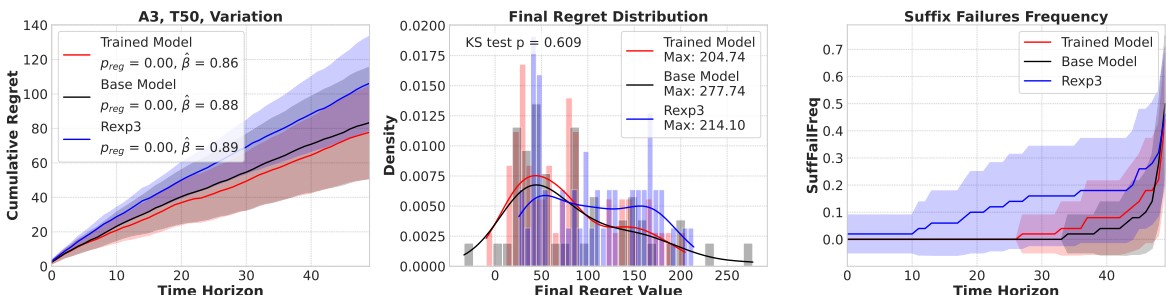

*Figure 30.* **The regret over time, the final regret distribution, and the exploration metric for the NS-MAB environment under Horizon Generalization**[$T = 25 \rightarrow T = 50$] **and Linguistic Context Generalization**[**GPT-4o mini** $\rightarrow$ **Gemini 2.0-Flash**]**,** trained and evaluated with the Gradual Variation reward, which shows a lower regret and sublinear regret behavior for the trained model.

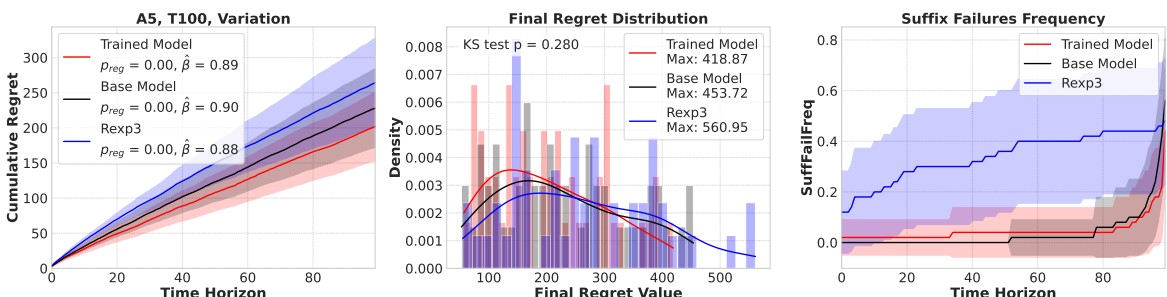

*Figure 31.* **The regret over time, the final regret distribution, and the exploration metric for the NS-MAB environment under Horizon Generalization**[$T = 25 \rightarrow T = 100$] **and Action Space Size Generalization**[$d = 3 \rightarrow d = 5$]**,** trained and evaluated with the Gradual Variation reward, which shows a lower regret and sublinear regret behavior for the trained model.

## M.4. Improved Reasoning Rationales

Lastly, we visualize representative reasoning rationales produced by the base and trained models of GPT-4o mini in Figure 24. The color-coded examples highlight how our training framework enhances both *semantic–numerical alignment* and the *E-E tradeoff*. They illustrate that our post-training can indeed enhance the reasoning rationales of the models in these examples, via both a better understanding of the numerical values and the online DM principle of properly balancing exploration and exploitation.

## N. Omitted Proof of Theorem 1

For any history length $t \geq 1$, we define

$$g_t\big((R_1,\ldots,R_{t-1},\mathbf{1}_d); V,K,Q,v_c,k_c,q_c\big) := \sum_{\tau=1}^{t-1} \big(VR_\tau + v_c\big)\big((KR_\tau + k_c)^\top (Q\mathbf{1}_d + q_c)\big), \tag{7}$$

where each $R_\tau \in \mathbb{R}^d$ denotes a reward vector in the FOL environment. We adopt the empty-sum convention, so $g_1((\mathbf{1}_d); V,K,Q,v_c,k_c,q_c) = 0$. Throughout the analysis, we assume $\{R_\tau\}_{\tau=1}^T$ are i.i.d. draws from $\mathcal{N}(\mathbf{0}_d, I_{d\times d})$.

**Theorem 1.** *Consider the policy space* $\Pi = B(\mathbf{0}_d, R_\Pi, \|\cdot\|_2)$ *for some* $R_\Pi > 0$, *and consider the minimization of Equation* (4)*, which corresponds to Algorithm 1 with infinite data and* $k = 1$*. Then, plugging in any global minimizer of Equation* (4) *within the model class parameterized by Equation* (7) *and projecting the resulting output using* $\mathrm{Proj}_{\Pi,\|\cdot\|}$ *yield an output from running FTRL with an* $\ell_2$*-regularizer and a stepsize of order* $\Theta\left(\frac{1}{\sqrt{Td}}\right)$.

*Proof.*

$$g(Z_t; V,K,Q,v_c,k_c,q_c)$$
$$= \sum_{i=1}^{t} \Big(VR_iR_i^\top(K^\top(Qc+q_c)) + \big(Vk_c^\top(Qc+q_c) + v_c(Qc+q_c)^\top K\big)R_i + v_c k_c^\top(Qc+q_c)\Big), \tag{8}$$

which can be expressed with a larger class

$$g(Z_t, \mathbb{A}, \beta, \mathbb{C}, \delta) := \sum_{i=1}^{t}(\mathbb{A}R_iR_i^\top\beta + \mathbb{C}R_i + \delta), \tag{9}$$

where $\mathbb{A} \in \mathbb{R}^{d\times d}$, $\beta, \mathbb{C}, \delta \in \mathbb{R}^d$, and $Z_t = (R_1,\ldots,R_t)$. Then, if a minimizer of

$$f(\mathbb{A}, \beta, \mathbb{C}, \delta) := \mathbb{E}\left[\sum_{t=1}^{T}\left\|\sum_{i=1}^{t-1}(\mathbb{A}R_iR_i^\top\beta + \mathbb{C}R_i + \delta) - \pi^\star(R_1,\ldots,R_T)\right\|_2^2\right]$$

can be expressed as $\mathbb{A} = V, \beta = K^\top(Qc+q_c), \mathbb{C} = Vk_c^\top(Qc+q_c) + v_c(Qc+q_c)^\top K, \beta = v_c k_c^\top(Qc+q_c)$, then we can conclude that the corresponding $V, Q, K, v_c, q_c, k_c$ are also a minimizer of

$$\mathbb{E}\left[\sum_{t=1}^{T}\left\|g(Z_{t-1}) - \pi^\star(R_1,\ldots,R_T)\right\|_2^2\right],$$

since the corresponding $V, Q, K, v_c, q_c, k_c$ constitute a minimizer among a larger class. Now, since $\Pi = B(\mathbf{0}_d, R_\Pi, \|\cdot\|)$, we can rewrite $f$ as

$$f(\mathbb{A}, \beta, \mathbb{C}, \delta) := \mathbb{E}\left[\sum_{t=1}^{T}\left\|\sum_{i=1}^{t-1}(\mathbb{A}R_iR_i^\top\beta + \mathbb{C}R_i + \delta) - R_\Pi\left(\frac{\sum_{i=1}^{T}R_i}{\left\|\sum_{i=1}^{T}R_i\right\|_2}\right)\right\|_2^2\right] \tag{10}$$

Here, Expectation is calculated with $R_i \sim \mathcal{N}(\mathbf{0}_d, I_{d\times d})$ and all $R_i$ are independent.

**Step 1. Finding the best** $\delta$. We have

$$\mathbb{E}\left[\sum_{t=1}^{T}\left\|\sum_{i=1}^{t-1}(\mathbb{A}R_iR_i^\mathsf{T}\beta + \mathbb{C}R_i + \delta) - R_\Pi\left(\frac{\sum_{i=1}^{T}R_i}{\left\|\sum_{i=1}^{T}R_i\right\|_2}\right)\right\|_2^2\right]$$

$$= \mathbb{E}\left[\sum_{t=1}^{T}\left\|\sum_{i=1}^{t-1}(\mathbb{A}R_iR_i^\mathsf{T}\beta + \mathbb{C}R_i - A\beta) - R_\Pi\left(\frac{\sum_{i=1}^{T}R_i}{\left\|\sum_{i=1}^{T}R_i\right\|_2}\right)\right\|_2^2\right] + \frac{T(T-1)}{2}\left\|A\beta + \delta\right\|_2^2$$

since $\mathbb{E}[\|X\|_2^2] = \mathbb{E}[\|X - \mathbb{E}[X]\|_2^2] + \|\mathbb{E}[X]\|_2^2$, $\delta = A\beta$ holds for any random vector $X$. Therefore, $\delta = -A\beta$ is the condition to minimize $f$.

**Step 2. Plugging the optimality condition for** $\delta$ **into** $f$ .

Plugging $\delta = -A\beta$ to $f$ provides

$$f(\mathbb{A},\beta,\mathbb{C},-\mathbb{A}\beta) = \mathbb{E}\left[\sum_{t=1}^{T}\left\|\sum_{i=1}^{t-1}(\mathbb{A}(R_iR_i^\mathsf{T} - I_{d\times d})\beta + \mathbb{C}R_i) - R_\Pi\left(\frac{\sum_{i=1}^{T}R_i}{\left\|\sum_{i=1}^{T}R_i\right\|_2}\right)\right\|_2^2\right]$$

$$= \sum_{t=1}^{T}\left(\underbrace{\mathbb{E}\left[\left\|\sum_{i=1}^{t-1}\mathbb{A}(R_iR_i^\mathsf{T} - I_{d\times d})\beta\right\|_2^2\right]}_{(i)} + \mathbb{E}\left[\left\|\sum_{i=1}^{t-1}\mathbb{C}R_i\right\|_2^2\right] + \mathbb{E}\left[\left\|R_\Pi\frac{\sum_{i=1}^{T}R_i}{\left\|\sum_{i=1}^{T}R_i\right\|_2}\right\|_2^2\right]\right)$$

$$+ 2\underbrace{\sum_{t=1}^{T}\mathbb{E}\left[\left(\sum_{i=1}^{t-1}\mathbb{A}(R_iR_i^\mathsf{T} - I_{d\times d})\beta\right)^\mathsf{T}\left(\sum_{i=1}^{t-1}\mathbb{C}R_i\right)\right]}_{(ii)}$$

$$-2\underbrace{\sum_{t=1}^{T}\mathbb{E}\left[\left(\sum_{i=1}^{t-1}\mathbb{A}(R_iR_i^\mathsf{T} - I_{d\times d})\beta\right)^\mathsf{T}\left(R_\Pi\frac{\sum_{i=1}^{T}R_i}{\left\|\sum_{i=1}^{T}R_i\right\|_2}\right)\right]}_{(iii)}$$

$$-2\sum_{t=1}^{T}\mathbb{E}\left[\left(\sum_{i=1}^{t-1}\mathbb{C}R_i\right)^\mathsf{T}\left(R_\Pi\frac{\sum_{i=1}^{T}R_i}{\left\|\sum_{i=1}^{T}R_i\right\|_2}\right)\right]$$

We can easily check that $(ii)$ and $(iii)$ are 0 as they are polynomials of odd degrees and we have $Z \overset{d}{=} -Z$. For the part $(i)$, we have

$$\sum_{t=1}^{T}\mathbb{E}\left[\left\|\sum_{i=1}^{t-1}\mathbb{A}(R_iR_i^\mathsf{T} - I_{d\times d})\beta\right\|_2^2\right] = \sum_{t=1}^{T}\mathbb{E}\left[\sum_{i_1=1}^{t-1}\sum_{i=1}^{t-1}\beta^\mathsf{T}(R_{i_1}R_{i_1}^\mathsf{T} - I_{d\times d})\mathbb{A}^\mathsf{T}\mathbb{A}(R_iR_i^\mathsf{T} - I_{d\times d})\beta\right]$$

$$\overset{=}{\underset{(1)}{=}} \sum_{t=1}^{T}\mathbb{E}\left[\sum_{i=1}^{t-1}\beta^\mathsf{T}(R_iR_i^\mathsf{T} - I_{d\times d})\mathbb{A}^\mathsf{T}\mathbb{A}(R_iR_i^\mathsf{T} - I_{d\times d})\beta\right] \tag{11}$$

$$= \frac{(T-1)T}{2}\beta^\mathsf{T}\mathbb{E}\left[(A(R_iR_i^\mathsf{T} - I_{d\times d}))^\mathsf{T}(A(R_iR_i^\mathsf{T} - I_{d\times d}))\right]\beta.$$

Here, (1) holds because if $i_1 \neq i$, we can calculate $\mathbb{E}(R_{i_1}R_{i_1}^\mathsf{T} - I_{d\times d}) = O_{d\times d}$. Note that Equation (11) is minimized when $\mathbb{P}(\mathbb{A}(R_iR_i^\mathsf{T} - I_{d\times d})\beta = \mathbf{0}_d) = 1$.

If $\mathbb{A} \neq O_{d\times d}$, suppose that the singular value decomposition of $A = U\Lambda V$ yields that $\Lambda$ is a diagonal matrix whose first diagonal element is non-zero, and $U, V$ are orthogonal matrices. Then, we want to find $\beta$ such that $U\Lambda V(R_iR_i^\mathsf{T} - I_{d\times d})\beta = \mathbf{0}_d$ for any $R_i$ such that $p(R_i) \neq 0$, where $p$ corresponds to the probability density function of reward vectors. Since $U$ is invertible, we only need to consider $\Lambda V(R_iR_i^\mathsf{T} - I_{d\times d})\beta = \mathbf{0}_d$. Since $\Lambda$'s first diagonal component is non-zero, we will

consider the equation $e_1^\mathsf{T}\Lambda V(R_i R_i^\mathsf{T} - I_{d\times d})\beta = 0$. This is equivalent to $V_1(R_i R_i^\mathsf{T} - I_{d\times d})\beta = 0$, where $V_1$ is the first row of $V$, and is a non-zero vector.

**Claim 1.** *Define $\mathcal{L} = B(\mathbf{0}_d, 1, \|\cdot\|_2)$ and $\mathcal{S} := \left\{ V_1\left(R_i R_i^\mathsf{T} - I_{d\times d}\right) : R_i \in \mathcal{L} \right\} \subseteq \mathbb{R}^d$. Then, $\mathcal{S}$ has positive volume. As a consequence, this set is full $d$-dimensional and contains $d$ linearly independent vectors.*

Therefore, we can find $d$ reward vectors $\{R_i\}_{i\in[d]}$ such that the vectors $\{V_1(R_i R_i^\mathsf{T} - I_{d\times d})\}_{i\in[d]}$ are linearly independent. Hence, if we want to minimize Equation (11), either $A = O_{d\times d}$ or $\beta = \mathbf{0}_d$ should hold. In both cases, Equation (9) can be re-written as

$$g(Z_t; \mathbb{A}, \beta, \mathbb{C}, \delta) := \sum_{i=1}^{t} \mathbb{C}R_i,$$

and this is covered by the original parametrization (Equation (7)) with $K^\mathsf{T}(Qc + q_c) = v_c = \mathbf{0}_d$.

**Step 3. Finding the best $\mathbb{C}$.**

Now, we optimize over $\mathbb{C}$, by minimizing the following objective:

$$
\begin{aligned}
f(\mathbb{C}) := & \mathbb{E}\left[\sum_{t=1}^{T} \left\| \sum_{i=1}^{t-1} \mathbb{C}R_i - R_\Pi\left( \frac{\sum_{i=1}^{T} R_i}{\left\|\sum_{i=1}^{T} R_i\right\|_2} \right) \right\|_2^2 \right] \\
= & \sum_{t=1}^{T} \mathbb{E}\left[ \left\| \sum_{i=1}^{t-1} \mathbb{C}R_i \right\|_2^2 \right] - 2R_\Pi \sum_{t=1}^{T} \mathbb{E}\left[ \left( \sum_{i=1}^{t-1} \mathbb{C}R_i \right)^\mathsf{T} \left( \frac{\sum_{i=1}^{T} R_i}{\left\|\sum_{i=1}^{T} R_i\right\|_2} \right) \right] \\
& + R_\Pi^2 \sum_{t=1}^{T} \mathbb{E}\left[ \frac{(\sum_{i=1}^{T} R_i)^\mathsf{T}(\sum_{i=1}^{T} R_i)}{\left\|\sum_{i=1}^{T} R_i\right\|_2^2} \right] \\
= & \frac{T(T-1)}{2} d\mathbb{C}^\mathsf{T}\mathbb{C} - 2R_\Pi \mathbb{C}^\mathsf{T} \sum_{t=1}^{T} \mathbb{E}\left[ \frac{\left(\sum_{i=1}^{t-1} R_i\right)^\mathsf{T} \left(\sum_{i=1}^{T} R_i\right)}{\left\|\sum_{i=1}^{T} R_i\right\|_2} \right] \\
& + \sum_{t=1}^{T} \mathbb{E}\left[ \frac{(\sum_{i=1}^{T} R_i)^\mathsf{T}(\sum_{i=1}^{T} R_i)}{\left\|\sum_{i=1}^{T} R_i\right\|_2^2} \right],
\end{aligned}
$$

since $\mathbb{E}\left[\left\|\sum_{i=1}^{t-1} R_i\right\|_2^2\right] = (t-1)d$.

Therefore, the optimal $\mathbb{C}$ can be calculated as

$$\mathbb{C} = \frac{2R_\Pi}{T(T-1)d} \sum_{t=1}^{T} \mathbb{E}\left[ \frac{\left(\sum_{i=1}^{t-1} R_i\right)\left(\sum_{i=1}^{T} R_i\right)^\mathsf{T}}{\left\|\sum_{i=1}^{T} R_i\right\|_2} \right],$$

or we can rewrite it as

$$\mathbb{C} = \frac{2R_\Pi}{T(T-1)d} \sum_{t=1}^{T} \mathbb{E}\left[ \frac{S_{t-1} S_T^\mathsf{T}}{\|S_T\|_2} \right],$$

where $S_T := \sum_{i=1}^{T} R_i$, $S_{t-1} := \sum_{i=1}^{t-1} R_i$. Because $(S_{t-1}, S_T)$ is jointly Gaussian, $\mathbb{E}[S_{t-1} \mid S_T] = \frac{t-1}{T} S_T$, so

$$\mathbb{E}\left[ \frac{S_{t-1} S_T^\mathsf{T}}{\|S_T\|_2} \right] = \frac{t-1}{T} \mathbb{E}\left[ \frac{S_T S_T^\mathsf{T}}{\|S_T\|_2} \right]$$

and

$$\mathbb{C} = \frac{2R_\Pi}{T(T-1)d} \frac{T-1}{2} \mathbb{E}\left[ \frac{S_T S_T^\mathsf{T}}{\|S_T\|_2} \right] = \frac{R_\Pi}{Td} \mathbb{E}\left[ \frac{S_T S_T^\mathsf{T}}{\|S_T\|_2} \right].$$

**Claim 2.** *If $QX \overset{d}{=} X$ for all $Q \in \mathbb{O}(d)$, then*

$$\mathbb{E}\left[\frac{XX^\mathsf{T}}{\|X\|_2}\right] = \frac{\mathbb{E}[\|X\|_2]}{d} I_{d \times d}.$$

Therefore, we have

$$\mathbb{C} = \frac{R_\Pi}{Td} \frac{\mathbb{E}\|S_T\|_2}{d} I_{d \times d} \overset{(i)}{=} \frac{R_\Pi}{Td} \sqrt{2T} \frac{\Gamma\left(\frac{d+1}{2}\right)}{\Gamma\left(\frac{d}{2}\right)} I_d = \frac{\sqrt{2}R_\Pi}{\sqrt{T}d} \frac{\Gamma\left(\frac{d+1}{2}\right)}{\Gamma\left(\frac{d}{2}\right)} I_d.$$

Here, (i) holds since $S_t \sim \mathcal{N}(\mathbf{0}_d, tI)$. Actually, this can be calculated even in closed-form as

$$\mathbb{C} = \frac{\sqrt{2}R_\Pi}{\sqrt{T}} \frac{(d-1)!!}{2^{d/2}} \frac{\sqrt{\pi}}{(\frac{d}{2}-1)!d} I_d,$$

if $d$ is even, and as

$$\mathbb{C} = \frac{\sqrt{2}R_\Pi}{\sqrt{T}} \frac{\left(\frac{d-1}{2}-1\right)!2^{\frac{d-1}{2}}}{(d-2)!!} \frac{1}{\sqrt{\pi}} I_d,$$

if $d$ is odd. If $d$ goes to infinity, we have

$$\lim_{d \to \infty} \sqrt{d}\mathbb{C} = \lim_{d \to \infty} \frac{\sqrt{2}R_\Pi}{\sqrt{T}d} \frac{d}{\sqrt{2}} I_d = \frac{R_\Pi}{\sqrt{T}} I_{d \times d}. \tag{12}$$

$\square$

**Claim 1.** *Define $\mathcal{L} = B(\mathbf{0}_d, 1, \|\cdot\|_2)$ and $\mathcal{S} := \left\{V_1\left(R_i R_i^\mathsf{T} - I_{d \times d}\right) : R_i \in \mathcal{L}\right\} \subseteq \mathbb{R}^d$. Then, $\mathcal{S}$ has positive volume. As a consequence, this set is full $d$-dimensional and contains $d$ linearly independent vectors.*

*Proof of Claim 1.* Let $v := V_1^\mathsf{T} \in \mathbb{R}^d$, and define

$$s_v(R) := (RR^\mathsf{T} - I_d)v, \qquad R \in \mathcal{L}.$$

Since $RR^\mathsf{T} - I_d$ is symmetric, the image of this map is the transpose of the set

$$\{V_1(RR^\mathsf{T} - I_d) : R \in \mathcal{L}\}.$$

Thus it suffices to show that $s_v(\mathcal{L}) \subset \mathbb{R}^d$ has positive $d$-dimensional volume.

The derivative of $s_v$ with respect to $R$ is

$$D_R s_v(R) = (R^\mathsf{T}v)I_d + Rv^\mathsf{T}.$$

Indeed, for any perturbation $h \in \mathbb{R}^d$,

$$D_R s_v(R)[h] = h(R^\mathsf{T}v) + R(h^\mathsf{T}v).$$

Therefore the Jacobian matrix is

$$J_v(R) := D_R s_v(R) = (R^\mathsf{T}v)I_d + Rv^\mathsf{T}.$$

Using the matrix determinant lemma, whenever $R^\mathsf{T}v \neq 0$,

$$\det J_v(R) = \det\left((R^\mathsf{T}v)I_d + Rv^\mathsf{T}\right) = 2(R^\mathsf{T}v)^d.$$

Because $v \neq 0$, we can choose $R_0 \in \mathcal{L}$ such that $R_0^\mathsf{T}v \neq 0$; for example,

$$R_0 = \frac{1}{2} \frac{v}{\|v\|_2}.$$

Then
$$\det J_v(R_0) = 2(R_0^\mathsf{T} v)^d \neq 0.$$

By continuity, there exists an open ball $\mathcal{B} \subset \mathcal{L}$ around $R_0$ such that
$$\det J_v(R) \neq 0 \qquad \text{for all } R \in \mathcal{B}.$$

Hence $s_v$ is locally nondegenerate on $\mathcal{B}$. By the inverse function theorem, $s_v(\mathcal{B})$ contains an open subset of $\mathbb{R}^d$. Therefore $s_v(\mathcal{L})$ has positive $d$-dimensional volume.

Since transposition preserves $d$-dimensional volume, the set
$$\{V_1 (RR^\mathsf{T} - I_d) : R \in \mathcal{L}\}$$

also has positive volume. Consequently, this set is full-dimensional and contains $d$ linearly independent vectors. □

**Claim 2.** *If $QX \stackrel{d}{=} X$ for all $Q \in \mathbb{O}(d)$, then*
$$\mathbb{E}\left[\frac{XX^\mathsf{T}}{\|X\|_2}\right] = \frac{\mathbb{E}[\|X\|_2]}{d} I_{d\times d}.$$

*Proof of Claim 2.* By the rotational invariance of $X$, we have that for any $Q \in \mathbb{O}(d)$,
$$\mathbb{E}\left[\frac{XX^\mathsf{T}}{\|X\|}\right] = \mathbb{E}\left[\frac{QX(QX)^\mathsf{T}}{\|QX\|}\right] = Q\mathbb{E}\left[\frac{XX^\mathsf{T}}{\|X\|}\right]Q^\mathsf{T}.$$

This shows that $\mathbb{E}\left[\frac{XX^\mathsf{T}}{\|X\|}\right]$ is invariant under conjugation by orthogonal matrices, and hence must be a scalar multiple of the identity matrix:
$$\mathbb{E}\left[\frac{XX^\mathsf{T}}{\|X\|}\right] = \lambda I_{d\times d},$$

for some $\lambda \in \mathbb{R}$. To determine $\lambda$, we take the trace of both sides:
$$\text{Tr}\left(\mathbb{E}\left[\frac{XX^\mathsf{T}}{\|X\|}\right]\right) = \mathbb{E}\left[\frac{\text{Tr}(XX^\mathsf{T})}{\|X\|}\right] = \mathbb{E}\left[\frac{\|X\|^2}{\|X\|}\right] = \mathbb{E}[\|X\|],$$

and
$$\text{Tr}(\lambda I_d) = \lambda d.$$

Equating both traces yields $\lambda = \frac{\mathbb{E}[\|X\|]}{d}$. Therefore, we have
$$\mathbb{E}\left[\frac{XX^\mathsf{T}}{\|X\|}\right] = \frac{\mathbb{E}[\|X\|]}{d} \cdot I_{d\times d},$$

which completes the proof. □

