# OpenReview forum: "Post-Training LLMs as Better Decision-Making Agents: A Regret-Minimization Approach"
_ICML.cc/2026/Conference — ICML 2026 regular_

### Official Review · Reviewer_Kiri · 2026-03-02

**Soundness:** 3
**Presentation:** 3
**Significance:** 3
**Originality:** 3
**Overall Recommendation:** 4
**Confidence:** 4

**Summary:**

This paper studies how to post-train LLMs to become better sequential decision-making agents in online learning and bandit settings. The authors propose ITERATIVE RMFT, an iterative self-training procedure that samples multiple decision trajectories, scores them by regret, and fine-tunes the model on the lowest-regret trajectories. Experiments across several model types show lower regret and better exploration behavior, with some generalization to new horizons and reward processes.

**Compliance With Llm Reviewing Policy:**

Affirmed.

**Final Justification:**

I appreciate the authors’ rebuttal, which addressed my main concerns. As a result, I maintain my original score and overall assessment.

**Key Questions For Authors:**

1. How sensitive is performance to the number of sampled trajectories, the top-k selection, and the number of iterations?
2. How does the method perform when regret is only approximately available or is noisy?
3. For the closed-weight model experiments, can you provide more details on data size and reproducibility?

**Limitations:**

Same as the weaknesses above.

**Strengths And Weaknesses:**

Strengths:
1. The method is simple and clearly described. The sample, score, select, and fine-tune loop is easy to follow.
2. Regret provides a clean and unified training signal across different online decision-making environments.
3. The empirical evaluation is fairly broad, covering numerical Transformers, open-weight LLMs, and a closed-weight model. The analysis connecting learned behavior to classic no-regret algorithms is helpful for interpretation.

Weaknesses:
1. Originality feels limited because the core recipe is close to best-of-N self-training, with regret mainly acting as the scoring function.
2. Training relies on access to regret computed with hindsight, which may be unrealistic in many real deployments. It would help to test estimated or noisy regret.
3. Most evaluations remain synthetic and relatively short-horizon, so it is unclear how well this transfers to richer interactive tasks.

---

> ### Author Rebuttal · Authors · 2026-03-31
>
> Thank you for your review. Please find our responses below.
>
> ## W1. Originality — Close to Best-of-N with Regret as Scoring Function
>
> We respectfully disagree that the contribution reduces to "best-of-N with a different scoring function." The novelty spans five dimensions.
>
> **1. New task formulation.** Iterative RMFT is the first framework systematically post-training LLMs as *no-regret online learners* in language-grounded sequential DM — spanning FOL, MAB, NS-MAB, and (new) contextual bandits — with heterogeneous horizons, action spaces, and natural-language context.
>
> **2. Theoretically grounded signal.** Regret is not an arbitrary quality score. Our results formally establish convergence: (i) **Result A** — in the single-layer setting, the procedure converges a.s. to FTRL ($c_n \to c^\star$) via monotonicity from $k$ fallback copies and a uniform lower bound $q_\varepsilon>0$ ensuring finite hitting time; (ii) **Result B** — for arbitrary neural architectures, excess regret contracts geometrically (see Reviewer gyG3 W1). These guarantees do not hold for arbitrary reward-based filtering.
>
> **3. 2× sample efficiency over AD; outperforms GRPO** (Qwen-3-8B, UCB teacher, train $T=25$, eval $T=100$): Iterative RMFT significantly outperforms $AD_{IMk} $  and $AD_{IML}$; $AD_{2IML}$ ($2\times$ budget) and both GRPO variants; $\mathrm{GRPO}_{\text{regret}}$ run at IMk only due to time constraints. (see Reviewer gyG3 W3).
>
> **4. Model-generated CoT reasoning.** Action+CoT RMFT consistently outperforms action-only RMFT — a dimension absent from Nie et al. (2024) and manually designed in Schmied et al. (2025).
>
> **5. Novel operationalization of regret for SFT.** Park et al.'s differentiable regret-loss requires backpropagation through regret — infeasible for autoregressive LLMs. Our trajectory-level selection criterion applies to any SFT pipeline, including proprietary APIs (GPT-4o mini), without modifying the loss.
>
> ## W2. Hindsight Regret Requires Privileged Information
>
> Regret is used **solely as a training-time supervision signal** in settings where dynamics are known or replayable (e.g., sandboxed simulators). No privileged information is required at inference time. **This use of privileged information has also been standard in the literature**: AD (Laskin et al., ICLR 2023), Park et al. (ICLR 2025), and Nie et al. (2024) all require environment access during training.
>
> In practice: in stochastic MABs, ranking by regret reduces to ranking by accumulated reward (the optimal action cancels out). For open-weight LLMs, we use *realized regret* (Appendix H.1) — a noisy single-sample estimate — and consistent improvements confirm robustness. We will clarify the training/inference distinction in the revision.
>
> ## W3. Synthetic and Short-Horizon Evaluations
>
> **Horizon length is not short.** A horizon of $T=100$ exceeds standard web-agent benchmarks (WebShop, WebArena: 30–50 steps). Our models are trained at $T=25$ and evaluated at $T=100$ (**horizon generalization**), demonstrating that strategies transfer without retraining.
>
> **Contextual bandits** ($A=3$, $T=25$, $C=2$, 5 epochs, 200 scenarios; GPT-4o mini): non-linear reward causes LinUCB to fail catastrophically (max regret 219.64); Iterative RMFT (135.07) outperforms base model (142.63) — full results in Reviewer XSDJ W3.
>
> ## Q1. Sensitivity to $L$, $k$, and Number of Iterations
>
> As per your feedback, **we have added new controlled ablation** on Qwen-3-8B (MAB, train $T=25$, eval $T=100$), varying $L$, $k$, number of iterations, and temperature one at a time (see Reviewer XSDJ W2 for the full table).
>
> **Key finding:** Multiple rounds of $L$ samples consistently outperform one large round of $I\times L$ samples at the same total budget — each round generates from the *current best model*, producing progressively higher-quality candidates.
>
> ## Q2. Performance with Approximate/Noisy Regret
>
> Our training **already operates with approximate regret**:
>
> 1. **Realized regret is inherently noisy.** For open-weight LLMs in MAB/NS-MAB (Appendix H.1), we use a single-sample Monte Carlo estimate. Consistent improvements confirm robustness.
> 2. **Noisy reward processes.** The Gaussian mixture reward (Distribution II) has variance components 1, 3, and 10; Gamma reward (Distribution III) has heavy-tailed parameters. $\hat{\beta}$ improves 0.760 → 0.680 on Gemma-2-9b-it (Figure 2).
> 3. **Quantitative robustness (Appendix L.5, Figure 25):** Iterative RMFT significantly reduces instability-induced optimal action rejection on Qwen3-8B.
>
> ## Q3. Closed-Weight Model — Data Size and Reproducibility
>
> Full GPT-4o mini training details (data size, hyperparameters, API configuration) are already documented in Appendix H.4. We will make this section more prominent in the revision.

---

> > ### Author Rebuttal · Reviewer_Kiri · 2026-03-31
> >
> > The rebuttal has addressed my concerns, and I will maintain my original score.

---

> > > ### Author Response · Authors · 2026-04-01
> > >
> > > Thank you very much for your thoughtful response—we really appreciate you taking the time to review our rebuttal and are glad to hear that your concerns have been fully addressed.
> > >
> > > Since you selected "(a) Fully resolved - My concerns have been adequately addressed. If you select this option, please consider adjusting your score accordingly." we would sincerely appreciate it if you could consider adjusting the score accordingly, if possible, as we have put significant effort into obtaining the new experiment results and preparing the rebuttal. If there are any other questions we can help further address, please do not hesitate to let us know.
> > >
> > > Thank you very much again for your consideration.

---

### Official Review · Reviewer_4icZ · 2026-03-08

**Soundness:** 2
**Presentation:** 3
**Significance:** 3
**Originality:** 2
**Overall Recommendation:** 4
**Confidence:** 3

**Summary:**

This paper proposes ITERATIVE RMFT, a post-training framework that leverages regret as a training signal to improve the decision-making (DM) capabilities of large language models. The approach iteratively distills low-regret decision trajectories into base models via supervised fine-tuning, without relying on pre-defined expert algorithms or manually designed reasoning formats. The authors evaluate the framework across numerical Transformers, open-weight LLMs (e.g., Gemma-2-9b-it, Qwen3-8B), and a closed-weight model (GPT-4o mini) on online DM tasks (FOL, MAB, NS-MAB), claiming improvements in regret reduction, exploration-exploitation tradeoff, and generalization across reward distributions, horizons, linguistic contexts, and action spaces.

**Compliance With Llm Reviewing Policy:**

Affirmed.

**Final Justification:**

I will maintain my score.

**Key Questions For Authors:**

Your results focus on positive outcomes, but provide no analysis of failure cases. (i) When does ITERATIVE RMFT fail to reduce regret or improve exploration-exploitation? (ii) How robust is your framework to adversarial environments (e.g., maliciously designed reward signals) or noisy/biased feedback?

**Limitations:**

yes

**Strengths And Weaknesses:**

## Strengths
1. Theoretical Grounding in Decision Theory: The use of regret as the core training signal provides a more principled foundation for LLM DM post-training, compared to heuristic-based or reward-only optimization methods. This aligns the work with rigorous decision-theoretic principles, a strength in ICML’s focus on theoretically motivated methods.
2. The framework’s compatibility with both open-weight and closed-weight LLMs (via API-accessible SFT) addresses a critical gap in existing DM-focused LLM post-training, which often requires full model access.
3. Comprehensive Experimental Coverage: The authors evaluate across multiple model classes (numerical Transformers, open-source LLMs, closed-source LLMs) and DM environments (FOL, MAB, NS-MAB), with a focus on generalization (reward, horizon, linguistic context, action space)—a key criterion for assessing the robustness of DM methods.
4. The empirical finding that the framework elicits exploration-exploitation and converges to classical DM algorithms without explicit distillation is noteworthy, providing evidence that LLMs can autonomously learn principled DM behaviors.

## Weaknesses

1. Insufficient Novelty and Methodological Limitations
* ITERATIVE RMFT is essentially an SFT pipeline with trajectory rollouts and filtering, and I think its novelty relative to existing work is overstated. Selecting high-performance trajectories for SFT is not new (e.g., data distillation, preference-based SFT), but the only distinction is the use of regret as the filtering metric.
* The paper fails to clearly differentiate itself from prior work on LLM DM post-training (e.g., Nie et al., 2024; Schmied et al., 2025) beyond the choice of training signal (I saw line 1142, and maybe more experiments are needed to compare).
2. Critical Flaws in Experimental Design
* the "iterative" component is poorly justified: the authors do not demonstrate that multiple iterations provide meaningful improvements over a single round of low-regret trajectory SFT. For example, is it better to have more roll-out trajectories in the same iteration, or more iterations with fewer trajectories?
* Synthetic Environments Only: All experiments are conducted in synthetic or procedurally generated environments, with no evaluation in real-world human-in-the-loop systems. The authors acknowledge this but fail to address how the results translate to practical DM scenarios.
* Weak Generalization Validation: The "linguistic context generalization" (Section 6) only tests generalization to contexts generated by a different LLM (Gemini 2.0-Flash), not to diverse, naturally occurring real-world language contexts (e.g., domain-specific jargon, ambiguous instructions). No comparison of different prompts, few-shot, CoT settings, etc.
* Authors compare their method to classical DM algorithms (FTRL, UCB) but frame these as "reference points" rather than proper baselines. They fail to compare ITERATIVE RMFT to state-of-the-art LLM DM post-training methods (e.g., RLHF for DM, algorithm distillation, GRPO/PPO using regret as rewards) to demonstrate superiority. The untrained model is a weak baseline, and the lack of comparison to competing methods makes it impossible to assess the framework’s added value.
*  noise parameters (Gaussian perturbation) and their impact on performance, and computational cost (e.g., number of trajectories sampled per iteration, training time) relative to baseline methods need to be reported and ansysis

---

> ### Author Rebuttal · Authors · 2026-03-31
>
> Thank you for your review. Please find our responses below.
>
> ## W1/W2. Insufficient Novelty — Iterative Component Not Justified
>
> We respectfully disagree. The novelty spans five dimensions:
>
> **1. Theoretically grounded signal.** Regret directly measures the gap between cumulative performance and the hindsight-optimal policy — enabling exploration–exploitation balance that reward-based signals cannot. Our results formally establish this:
>
> - **Result A (Convergence to FTRL):** In the single-layer linear-attention setting, the procedure converges a.s. to $c^\star$ (FTRL), via $k$ fallback copies (monotonicity), uniform $q_\varepsilon > 0$ (finite hitting time), and re-entry prevention. These guarantees do not hold for arbitrary reward filtering.
> - **Result B (General Model Classes):** For arbitrary neural architectures, excess regret contracts geometrically toward zero when residuals vanish — covering single- and multi-layer transformers (see Reviewer gyG3 W1 for the full bound).
>
> **2. Iterative design is justified empirically.** Multiple rounds of $L$ samples outperform one round of $I\times L$ samples at the same total budget — each round generates from the *current best model* (see Reviewer XSDJ W2 for ablation).
>
> **3. 2× sample efficiency over AD; outperforms GRPO** (Qwen-3-8B, UCB teacher, train $T=25$, eval $T=100$): Iterative RMFT significantly outperforms $AD_{IMk} $  and $AD_{IML}$; $AD_{2IML}$ ($2\times$ budget) and both GRPO variants; $\mathrm{GRPO}_{\text{regret}}$ run at IMk only due to time constraints (see Reviewer gyG3 W3).
>
> **4. CoT ablation** confirms model-generated reasoning matters: action+CoT RMFT (avg regret 19.68) outperforms action-only (22.23) at eval $T=50$ — see Reviewer XSDJ Q1.
>
> **5. New task formulation:** the first framework post-training LLMs as *no-regret online learners* across FOL, MAB, NS-MAB, and contextual bandits, with heterogeneous horizons and natural-language context.
>
> ## W2. Synthetic Environments Only
>
> Synthetic environments are the appropriate testbed for validating a training *methodology* — they allow exact regret computation, isolated ablations, and controlled scaling. In fact, **the use of similar synthetic environments has been standard**: AD (Laskin et al., ICLR 2023), Park et al. (ICLR 2025), and Nie et al. (2024) all evaluate exclusively in synthetic settings.
>
> As per your feedback, we added **contextual bandit** and **MDP** experiments (see Reviewer XSDJ W3 for results). We are also pursuing web-based agentic settings as a follow-up.
>
> ## W3. Weak Generalization Validation
>
> Our multi-turn structure means CoT and few-shot are structural properties of every instance — not separate configurations to ablate.
>
> We conducted a **rich vs. simple prompt** ablation (N=200 each). *The rich prompt is what we use in the paper; the simple prompt ("generate bandit scenarios") was an earlier version tried before submission*:
>
>
> | Metric | Rich Prompt | Simple Prompt |
> |---|---|---|
> | Vocabulary Size (higher = more diverse) | 1,481 | 634 |
> | Yule's K (lower = more diverse) | 83.70 | 331.64 |
> | MSTTR (higher = more diverse) | 0.7771 | 0.4936 |
> | Unique Jargon Terms (higher = more diverse) | 49 | 1 |
>
>
> Rich prompt: **4× greater vocabulary diversity** (Yule's K), 57% higher MSTTR, 49 jargon terms across 8+ domains vs. near-zero in the simple prompt — confirming the rich prompt drives substantially greater linguistic diversity.
>
> ## W4. Gaussian Noise Parameters and Computational Cost
>
> We conducted ablation studies comparing three noise configurations (GPT-4o mini):
>
>
> | Training | Mix test | Sine-Trend OOD | $\mathcal{N}(\mu,1)$ matched |
> |---|---|---|---|
> | Mixture | **47.36 / 23.11** | **35.12 / 10.21** | 46.05 / 22.67 |
> | Narrow $\mathcal{N}(\mu,1)$ | 55.82 / 27.94 | 51.95 / 12.84 | **35.93 / 17.12** |
> | Wide $\mathcal{N}(\mu,10)$ | 52.67 / 26.08 | 37.21 / 11.35 | 52.88 / 25.54 |
>
>
> *(Max Regret / Avg Regret)* Mixture achieves best balance in-distribution and OOD; narrow-trained models excel only on matched distributions. Cost details: Appendix H.1–H.4.
>
> ## W5. Failure Cases and Adversarial Robustness
>
> **Failure modes (which are already in Appendix) (3):**
>
> 1. **Low-entropy simplex bias (Gemma-2-9b-it, $d=3$):** Iterative RMFT fails to reduce regret (KS $p=0.378$) — Gemma's tokenization-induced bias prevents assigning high probability to a single action, and training does not resolve this.
> 2. **Interaction-type overfitting (Phi-3.5-mini):** Summary-type prompts cause higher regret ($\hat{\beta}=0.990$ vs. base $0.940$); loss rises in iterations 2–3, motivating dialogue-type interaction.
> 3. **Diminishing gains ($d=5$ MAB):** KS test $p=0.728$ — not statistically significant. Performance boundary as action space grows.
>
> **Adversarial robustness:** FOL is adversarial by construction. Across Adaptive (Dist. VIII, $\hat{\beta}=0.430$ vs. base $3.333$), Alternating (VI, $\hat{\beta}=0.062$), and Noisy Alternating (VII, $\hat{\beta}=0.790$), Iterative RMFT maintains sublinear regret.

---

> > ### Author Rebuttal · Reviewer_4icZ · 2026-04-01
> >
> > The response resolves my previous issues. I am happy to raise my rating to 4

---

> > > ### Author Response · Authors · 2026-04-01
> > >
> > > Thank you very much for your kind and encouraging response. We truly appreciate the time and effort you put into reviewing our rebuttal.
> > >
> > > We are grateful that our response was able to address your concerns, and we sincerely thank you for raising your rating, which means a lot to us.
> > >
> > > Your thoughtful feedback has been invaluable in improving our work. Thank you very much again for your support.

---

### Official Review · Reviewer_XSDJ · 2026-03-13

**Soundness:** 3
**Presentation:** 2
**Significance:** 3
**Originality:** 3
**Overall Recommendation:** 4
**Confidence:** 3

**Summary:**

This paper investigates a systematic approach to enhancing the online DM capabilities of LLMs through post-training. While LLMs are increasingly deployed as agents, they often struggle with fundamental decision-making principles, such as the exploration-exploitation tradeoff. The authors propose ITERATIVE RMFT, which leverages Regret, defined as the difference between an agent’s cumulative reward and the hindsight optimal reward, as a selection signal. Specifically, the model generates multiple trajectories (including CoT reasoning), and only those with low regret are utilized for SFT.

**Compliance With Llm Reviewing Policy:**

Affirmed.

**Final Justification:**

My concerns have been addressed and I maintain my positive score.

**Key Questions For Authors:**

- **Regret Estimation:** In real-world scenarios where the trainer does not know the true reward distribution (e.g., web browsing tasks), how do you propose estimating regret values to provide a viable training signal?
- **Sample Efficiency:** How sensitive is the model’s performance to the number of sampled trajectories $L$ and the $top-k$ selection? If $L$ is small, is there a risk of the model falling into sub-optimal behavior (i.e., greedy collapse)?
- **CoT Quality:** During the iterative process, did you observe hallucinations within the reasoning paths? Specifically, does the model ever provide incorrect logic to justify a low-regret action? If so, how does this affect the efficacy of the fine-tuning process?

**Limitations:**

yes

**Strengths And Weaknesses:**

**Strengths**

1. Unlike prior work using standard RL (typically focused on reward maximization for fixed tasks), this paper adopts Regret Minimization as a general principle to stimulate the model’s DM proficiency. This enables the model to learn how to learn within dynamic environments.
2. Through controlled experiments on single-layer linear attention Transformers, the authors demonstrate that ITERATIVE RMFT induces the emergence of classical no-regret algorithms, such as FTRL.
3. The framework is compatible with both open-source models (via SFT) and closed-source models (via fine-tuning APIs), and it naturally integrates model-generated CoT reasoning without requiring human-engineered templates.

**Weaknesses**

1. **Requirement of Privileged Information during Training：** Calculating regret requires knowing the hindsight optimal action, implying the trainer must have access to the underlying reward distribution or state transitions. While the paper notes this information is available during training, it limits the method’s applicability to environments where dynamics are fully known or perfectly simulatable.
2. **Computational Cost of Iteration:** The iterative nature of the algorithm requires sampling $L$ trajectories for $M$ scenarios in every iteration. For large-scale LLMs, the computational overhead of this sampling and subsequent fine-tuning could be prohibitive. The paper provides limited discussion on convergence speed or the optimal number of iterations.
3. **Scalability to Complex MDPs:** The experiments primarily focus on MAB, NS-MAB, and FOL. While these are foundational, it remains unclear how ITERATIVE RMFT handles credit assignment in deep, multi-step MDPs where regret is difficult to define or compute.
4. **Table Caption Formatting:** Table captions must be placed **above** the tables (not below) to comply with ICML guidelines.

---

> ### Author Rebuttal · Authors · 2026-03-31
>
> Thank you for your detailed review. Please find our responses below.
>
> ## W1/Q1. Privileged Information During Training
>
> Our method does **not** require privileged information **at inference time**. Regret is used solely as a **training-time signal** in simulatable environments — not too different from access to "reward samples" in RL. Three clarifications on what is actually required:
>
> - **MABs:** Regret = accumulated reward comparison. The **optimal-action comparator is identical across all $L$ trajectories** and cancels in selection — no privileged info needed.
> - **FOL:** Best-in-hindsight = argmax of accumulated **revealed** reward vectors — computed **for evaluation** anyway, and found on the fly.
> - **In practice (Appendix H.1):** We use *realized regret* (noisy single-sample Monte Carlo estimate) to approximate expected regret. Consistent improvements confirm robustness.
>
> This requirement of optimal labels as "privileged information" is a **common practice** in post-training LLMs: algorithm distillation (AD) (Laskin et al. 2023), Park et al. (2025), Nie et al. (2024), and meta-RL methods (Lee et al. 2023, Sinii et al. 2024, Beck et al. 2025) all require environment access during training. Song et al. (2026) further validate this paradigm.
>
> **Ongoing work:** We are removing the need for the privileged comparator in a follow-up paper via regret over a compact abstract representation space, with promising preliminary results in web-agentic tasks.
>
> ## W2/Q2. Computational Cost and Parameter Sensitivity
>
> **Existing documentation:** Appendix H.1–H.4 already reports exact GPU/API costs. We will make this more explicit in the revision.
>
> **Sample efficiency vs. baselines (→ full table: Reviewer gyG3 W3).** Iterative RMFT outperforms $\mathrm{AD}_{2IML}$ (2× budget) at matched cost — 2× better sample efficiency.
>
> **Parameter ablation** (Qwen-3-8B, MAB, train $T=25$, eval $T=100$; one parameter varied at a time; base config defined in Appendix H):
>
> | $L$ | $k$ | iterations | temperature | final regret |
> |---|---|---|---|---|
> | base | base | base | base | $33.08\pm 24.82$ |
> | 2 | 1 | 1 | 1.0 | $31.05\pm 23.34$ |
> | 2 | 1 | 2 | 1.0 | $28.33\pm 16.95$ |
> | 2 | 1 | 3 | 1.0 | $25.10\pm 14.50$ |
> | 5 | 1 | 1 | 0.2 | $35.28\pm 15.66$ |
> | 5 | 1 | 1 | 1.0 | $24.00\pm 13.34$ |
>
> More iterations at fixed total budget outperform fewer: 31.05 → 28.33 → 25.10 as iterations increase 1→3 (at $L=2$).
>
> **Key findings:** (1) *Temperature matters*: $\tau=0.2$ with $L=5$ yields 35.28 — *worse* than base (33.08) — while $\tau=1.0$ yields 24.00. Low temperature collapses candidate diversity, negating the benefit of larger $L$. (2) *Higher $L$ improves results but likely saturates*: $L=2\to5$ at $\tau=1.0$ reduces regret from 31.05 to 24.00, but further gains are expected to diminish — careful $L$ selection is important in practice.
>
> ## W3. Scalability to Complex MDP Problems
>
> **1. Contextual Bandits (CB).** $A=3$, $T=25$, $C=2$, GPT-4o mini, 200 scenarios, 5 epochs. Reward is a weighted nonlinear combination of two LLM-as-a-judge scores — making LinUCB inapplicable. Scenarios span healthcare, finance, education, and public policy.
>
> Example:
> ```json
> {"scenario": "A hospital testing treatment protocols for chronic pain patients where optimal treatment varies by age, pain severity, and medical history.", "action": ["opioid medication", "nerve blocks", "physical therapy"]}
> ```
>
> | Method | Final Max Regret ($\downarrow$) | Final Avg Regret ($\downarrow$) |
> |---|---|---|
> | LinUCB | 219.64 | 41.35 |
> | Base Model | 142.63 | 52.31 |
> | **Iterative RMFT** | **135.07** | **49.67** |
>
> LinUCB's max regret (219.64) reveals clear failure on nonlinear-reward scenarios; Iterative RMFT's lower max regret (135.07) confirms robustness on these hard cases.
>
> **2. MDPs.** $S=3$, $A=2$, $T=25$, 200 scenarios, 5 epochs. States labeled abstractly ($S_1,S_2,S_3$).
>
> | Method | Final Max Regret ($\downarrow$) | Final Avg Regret ($\downarrow$) |
> |---|---|---|
> | Base Model | 202.19 | 154.32 |
> | **Iterative RMFT** | **195.32** | **153.15** |
>
> Maximum regret continued decreasing, indicating further improvement with more epochs. Iterative RMFT shows promise beyond MAB/NS-MAB/FOL.
>
> ## W4. Table Caption Formatting
>
> We will place all table captions above the tables per ICML guidelines. Thank you.
>
> ## Q1. CoT Quality and Hallucinations
>
> CoT hallucination is well-documented across STaR, ReST, and self-play methods, yet those approaches still demonstrate consistent gains. We compare action-only vs. action+CoT RMFT (FOL, Gaussian mixture reward, train $T=25$, eval $T=50$):
>
> | Setting | Avg Regret |
> |---|---|
> | Action-only RMFT | 22.23 |
> | Action+CoT RMFT | **19.68** |
>
> Action+CoT RMFT outperforms action-only; rationale improvements shown in Figure 6 and Appendix J.5. We will add failure case analysis and mitigations in the revision.

---

> > ### Author Rebuttal · Reviewer_XSDJ · 2026-04-02
> >
> > My concerns have been adequately addressed. I will keep my postive score.

---

> > > ### Author Response · Authors · 2026-04-02
> > >
> > > Thank you very much for your thoughtful response—we really appreciate you taking the time to review our rebuttal and are glad to hear that your concerns have been adequately addressed.
> > >
> > > Since you selected "(a) Fully resolved - My concerns have been adequately addressed. If you select this option, please consider adjusting your score accordingly." we would sincerely appreciate it if you could consider adjusting the score accordingly, if possible, as we have put significant effort into obtaining the new experiment results and preparing the rebuttal. If there are any other questions we can help further address, please do not hesitate to let us know.
> > >
> > > Thank you very much again for your consideration.

---

### Official Review · Reviewer_gyG3 · 2026-03-13

**Soundness:** 3
**Presentation:** 3
**Significance:** 3
**Originality:** 3
**Overall Recommendation:** 4
**Confidence:** 4

**Summary:**

This paper introduces ITERATIVE RMFT (Iterative Regret-Minimization Fine-Tuning), a post-training method that repeatedly samples decision trajectories from an LLM, selects those with low regret, and fine-tunes the model on them. On the theory side, the paper proves that single-layer linear self-attention Transformers can implement FTRL (Follow-the-Regularized-Leader) for full-information online learning. Empirically, the approach is validated across three settings: (1) symbolic multi-armed bandit tasks with Transformers trained from scratch, (2) open-weight LLMs fine-tuned for language-grounded bandits, and (3) GPT-4o mini fine-tuned via the API. Results show sublinear regret and generalization across reward distributions, horizons, action spaces, and linguistic contexts.

**Compliance With Llm Reviewing Policy:**

Affirmed.

**Key Questions For Authors:**

How does ITERATIVE RMFT compare to Park et al.'s (ICLR 2025) regret-loss approach in a direct experiment? Both use regret as a training signal. If the iterative trajectory selection is the key differentiator, an ablation comparing single-pass regret-loss training vs. iterative selection would clarify this. This directly affects my originality assessment-if the difference is marginal, the novelty over Park et al. is limited.

Can you provide a direct comparison with Algorithm Distillation (Laskin et al., ICLR 2023) trained on the same environments? AD is the most natural baseline for "training Transformers to do sequential decision-making." Showing that regret-based selection outperforms algorithm-specific distillation would be the strongest evidence for this approach.

For results where improvements are small (e.g., Gemma d=2 Gaussian in Section 5.3, Table 9 GPT-4o mini Gaussian), are the differences statistically significant across random seeds? Standard deviations or confidence intervals would help separate signal from noise.

**Limitations:**

The paper discusses the theory-practice gap and the low-entropy simplex bias thoughtfully. I'd encourage the authors to additionally acknowledge: (a) the overlap with Park et al. (ICLR 2025) and the missing direct comparison; (b) that non-stationary bandit results underperform Rexp3; (c) the absence of an Algorithm Distillation baseline.

**Strengths And Weaknesses:**

Strengths
S1. Clean Theoretical Result: Transformers Implement FTRL. The present paper pushes this into online learning and regret minimization specifically, connecting to FTRL rather than generic gradient descent. It's a clean, verifiable contribution.

S2. Thorough Experimental Coverage. The evaluation is comprehensive. It tests across 4 generalization axes: horizon generalization (T=25 to T=100), reward distribution generalization (Gaussian, Uniform, Bernoulli, Sine-trend, Alternating, Adaptive adversarial-9 distributions total), action space generalization (d=3 to d=5), and linguistic context generalization (GPT-4o to Gemini evaluation).

S3. Practical and Broadly Applicable. ITERATIVE RMFT works across Transformers trained from scratch, open-weight LLMs (via LoRA fine-tuning), and closed-weight APIs (GPT-4o mini via the fine-tuning endpoint).

Weaknesses
W1. Theory-Practice Gap. Theorems 1-2 apply to a single-layer linear self-attention mechanism with i.i.d. Gaussian inputs. The actual algorithm uses multi-layer Transformers with softmax attention on natural language. The paper acknowledges this gap (Section 4 discussion) but doesn't bridge it. It's unclear whether the practical improvements stem from something related to FTRL implementation or from other effects of regret-based training-like simply selecting high-quality trajectories as training data. The theory motivates the approach but doesn't explain the empirical results. This concern is common in this literature, but it deserves more explicit discussion.

W2. Differentiation from Park et al. (ICLR 2025) "Do LLM Agents Have Regret?" Park et al. also study regret in LLMs for online learning and propose a regret-loss training objective with theoretical guarantees. The current paper cites them but the differentiation isn't fully precise. Both use regret as a training signal for LLMs on bandit/online learning tasks, and Park et al. also establish a connection to FTRL. The key differences seem to be: (a) ITERATIVE RMFT uses iterative trajectory selection rather than a single-pass loss; (b) the current paper tests on language-grounded settings with GPT-4o mini; (c) the theoretical result here is about Transformer weights implementing FTRL, while Park et al. focus on regret-loss optimization guarantees. These distinctions need to be made precise-ideally with a direct empirical comparison.

W3. No Direct Comparison with Algorithm Distillation. Laskin et al. (ICLR 2023) proposed Algorithm Distillation (AD), which trains Transformers on RL training histories via autoregressive prediction, enabling in-context learning of RL algorithms. AD is the most direct prior method for training Transformers to do sequential decision-making.

W4. Some Improvements Are Modest or Within Noise. Several results show marginal improvements that may not clear the bar for statistical significance. In Table 9 (GPT-4o mini, Gaussian, T=15), avg(LR) drops from 16.35 to 15.03-an 8% improvement. For the Gemma d=2 results (Section 5.3), improvements appear within noise (e.g., avg(LR) 21.14 trained vs. 21.18 base for Gaussian). The paper doesn't report confidence intervals or significance tests for individual settings. Given known variance in LLM evaluations, error bars across random seeds would help distinguish signal from noise.

---

> ### Author Rebuttal · Authors · 2026-03-31
>
> Thank you for your positive feedback. Please find our responses below.
>
> ## W1. Theory-Practice Gap
>
> We add **two new theoretical results** that **substantially bridge this gap**.
>
> **Result A — Finite-*L*, Finite-*k* Convergence to FTRL (single-layer linear attention).** Additional assumption: $k$ copies of the original model's output as fallbacks plus $L$ independent Gaussian proposals.
>
> **Theorem:** iterates converge a.s., $c_n \to c^\star$. Three proof ingredients: (i) $k$ fallback copies guarantee monotonicity on the loss; (ii) for any $\varepsilon$-ball around $c^\star$, there is a uniform lower bound $q_\varepsilon > 0$ on the probability that $\geq k$ proposals land inside, so the hitting time is a.s. finite; (iii) monotonicity prevents escape once entered. This shows the iterative training loop is **not a heuristic** — it **provably recovers** the FTRL solution.
>
> **Result B — General Model Classes (beyond single-layer linear attention).** Let $e_n = R_T(g_n) - R^\star_{T,\mathcal{G}}$ be the excess regret at iteration $n$. Then:
>
> $$e_{n+1} \leq \rho_n(\varepsilon)e_n + \varepsilon + B_r\sqrt{T\left(a_n(\mathcal{G}) + 2\Gamma_n + \varepsilon^{\mathrm{opt}}_n\right)}$$
>
> where $\rho_n(\varepsilon) = \mathbb{P}(\mathrm{Bin}(L,q_n(\varepsilon)) < k) < 1$, $a_n$ = teacher-approximation error, $\Gamma_n$ = generalization bound, $\varepsilon^{\mathrm{opt}}_n$ = optimization error. When residuals vanish: $e_n \to 0$ (geometric contraction). Gaussian perturbations satisfy this for any continuously parameterized, bounded-output model class with compact parameter set — covering both **single-** and **multi-layer** residual linear transformers.
>
> ## W2/Q1. Differentiation from Park et al. (ICLR 2025)
>
> Park et al. optimize a **differentiable** regret-loss on **numeric-input/output Transformers**, requiring backpropagation through the regret signal. Neither holds for **language-grounded** settings or proprietary LLM APIs (GPT-4o mini). Our contribution is a new **operationalization** of regret as a trajectory-level SFT selection criterion for natural-language outputs, applicable to autoregressive LLMs without modifying existing post-training paradigms.
>
> | Dimension | Park et al. (ICLR 2025) | Iterative RMFT |
> |---|---|---|
> | Core question | Do LLMs already behave as no-regret learners? | Can post-training instill no-regret behavior? |
> | Output format | Numerical policy vectors | Natural-language + numerical |
> | Tasks | Abstract online learning, matrix games | Language-grounded bandits, DM tasks |
> | Training loop | Single-phase regret-loss | Iterative rollout → filter → finetune |
> | Theory | Regret-loss → FTRL (optimization guarantee) | Single-layer Transformer → FTRL (architecture guarantee) |
> | Reasoning | None | Model-generated CoT |
>
> **Q1 follow-up:** Park et al.'s loss requires **differentiable** discrete sampling, which is applicable to Transformer models, but is not directly applicable to the **autoregressive-sampling-process** of LLMs, which is inherently non-differentiable. Moreover, see W3 for a head-to-head comparison against AD and RL.
>
> ## W3. Comparison with AD and RL
>
> Setup: Qwen-3-8B, UCB teacher, train $T=25$, eval $T=100$. Let $I$=iterations, $L$=samples/instance, $M$=instances/iter, $k$=selected. Total *used*: $IMk$; *generated*: $IML$.
>
> | Method | Base | $\mathrm{AD}_{IMk}$ | $\mathrm{AD}_{IML}$ | $\mathrm{AD}_{2IML}$ | $\mathrm{GRPO}_\text{step}$ | $\mathrm{GRPO}_\text{regret}$ | **Ours** |
> |---|---|---|---|---|---|---|---|
> | Avg Regret ($T=100$) | 52.10 | 58.72 | 33.44 | 32.97 | 35.88 | 40.09 | **22.37** |
>
> Iterative RMFT **significantly outperforms** $AD_{IMk} $  and $AD_{IML}$; $AD_{2IML}$ ($2\times$ budget) and both GRPO variants; $\mathrm{GRPO}_{\text{regret}}$ run at IMk only due to time constraints. **Two reasons for efficiency:** (1) *Minimal policy drift* — selective SFT avoids drastic behavioral changes. (2) *UCB horizon mismatch* — AD trained at $T=25$ fails to generalize at $T=100$ because UCB's exploration is coupled with history length.
>
> ## W4. Significance Tests
>
> The current draft includes regression-based regret-growth validation ($\hat{\beta}$), one-sided KS tests on final regret distributions, and confidence intervals in all figures. Individual settings remain underpowered, so **we conduct 3-replicate experiments with per-setting error bars**:
>
> | | Mean final regret | Mean SD (per-setting) |
> |---|---|---|
> | Base | 27.30 | 8.86 |
> | Trained | 23.13 | 8.13 |
>
> Per-setting SD ($\sim$ 8) is far below task-based SD ($\sim$ 25) — replicated experiments confirm per-setting robustness.
>
> ## Limitation 2. Non-Stationary Bandit vs. Rexp3
>
> Rexp3 is a **near-optimal** algorithm with **numerical input/output**, while LLMs were not designed for such a sequential decision-making problem, and importantly, **have language as input/output**. All numerical baselines should be viewed as **reference points** (gold-standard upper bounds), not targets claimed to be beaten.

---

### Decision · Program_Chairs · 2026-04-30

**Decision:**

Accept (regular)

**Comment:**

This paper introduces Iterative Regret-Minimization Fine-Tuning (Iterative RMFT), a novel post-training procedure that improves LLM decision-making by distilling low-regret trajectories. During the discussion phase, the authors successfully addressed key concerns regarding baseline comparisons (AD, GRPO), statistical significance, and theoretical grounding. Although minor gaps persist in extending to highly complex MDPs and discrete sampling, the methodology's originality and strong empirical results provide significant insights for the community.